# Premature commitment to uncertain decisions during human NMDA receptor hypofunction

Alexandre Salvador[1,2,3,4], Luc H. Arnal [5], Fabien Vinckier[3,4,6], Philippe Domenech [7,8], Raphaël Gaillard[3,4,9,10] & Valentin Wyart [1,2,10 ✉]

Making accurate decisions based on unreliable sensory evidence requires cognitive inference. Dysfunction of n-methyl-d-aspartate (NMDA) receptors impairs the integration of noisy input in theoretical models of neural circuits, but whether and how this synaptic alteration impairs human inference and confidence during uncertain decisions remains unknown. Here we use placebo-controlled infusions of ketamine to characterize the causal effect of human NMDA receptor hypofunction on cognitive inference and its neural correlates. At the behavioral level, ketamine triggers inference errors and elevated decision uncertainty. At the neural level, ketamine is associated with imbalanced coding of evidence and premature response preparation in electroencephalographic (EEG) activity. Through computational modeling of inference and confidence, we propose that this specific pattern of behavioral and neural impairments reflects an early commitment to inaccurate decisions, which aims at resolving the abnormal uncertainty generated by NMDA receptor hypofunction.

[1] Laboratoire de Neurosciences Cognitives et Computationnelles, Institut National de la Santé et de la Recherche Médicale, Paris, France. [2] Département d'Études Cognitives, École Normale Supérieure, Université PSL, Paris, France. [3] Université de Paris, Paris, France. [4] Département de Psychiatrie, Service Hospitalo-Universitaire, GHU Paris Psychiatrie et Neurosciences, Paris, France. [5] Institut de l'Audition, Inserm unit 1120, Institut Pasteur, Paris, France. [6] Équipe Motivation, Cerveau et Comportement, Institut du Cerveau, Sorbonne Université, Paris, France. [7] Équipe Neurophysiologie des Comportements Répétitifs, Institut du Cerveau, Sorbonne Université, Paris, France. [8] Département Médico-Universitaire de Psychiatrie et d'Addictologie, CHU AP-HP Henri Mondor, Université Paris-Est Créteil, Créteil, France. [9] Unité de Neuropathologie Expérimentale, Département de Santé Globale, Institut Pasteur, Paris, France. [10] These authors jointly supervised this work: Raphaël Gaillard, Valentin Wyart. ✉email: valentin.wyart@inserm.fr

In uncertain environments where sensory observations are unreliable, making decisions requires the combination of multiple pieces of ambiguous or even conflicting sensory information to form accurate beliefs about their generative cause or their consequences[1]. This form of "cognitive inference" can be described in terms of probabilistic (Bayesian) reasoning, where beliefs correspond to posterior distributions of hidden states of the environment given the available evidence[2]. In practice, this inference process has been extensively modeled in terms of a gradual evidence accumulation process[3] that implements—or approximates—normative Bayesian inference[4,5]. Previous research in humans has shown that the accuracy of this accumulation process is not bounded by the ability to maintain accumulated evidence over time[6,7], but by a limited computational precision[8] —i.e., random variability (noise) during the accumulation of evidence itself. These findings set the precision of inference as an important cognitive "bottleneck" on decision-making under uncertainty[9,10].

Theoretical models of neural circuits have identified n-methyl-d-aspartate (NMDA) synaptic receptors as necessary for the accurate integration of noisy input[11,12]. Indeed, hypofunction of NMDA receptors has been proposed to destabilize the attractor-like dynamics observed in these circuits, by altering the strength of recurrent synaptic connectivity. At the cognitive level, this synaptic alteration is thought to impair inference in a way that can trigger decision biases[13], "jumping to conclusions"[14,15], but also deficits in confidence[16]. Together, this theoretical work confers a central role to NMDA receptors in the computational precision of neural circuits implementing cognitive inference. However, and despite the breadth of this work, direct experimental characterization of the effects of NMDA receptor hypofunction on inference and confidence during uncertain decisions is still missing.

To address this issue, we administered sub-anesthetic infusions of ketamine, a non-competitive NMDA receptor antagonist[17], to healthy adult volunteers performing a visual cue combination task in a placebo-controlled, double-blind randomized crossover trial. The task was designed to provide specific estimates of the precision of cognitive inference, by measuring participants' ability to integrate the evidence provided by successive stimuli over several seconds[7,8]. We relied on a validated model of decision-making to decompose the sources of human decision errors in this task across sensory, inference and response selection stages of processing[8]. We further recorded electroencephalographic (EEG) activity from tested participants to identify which neural computations are altered under ketamine, from the visual processing of each cue up to the accumulation of the evidence provided by the same cues. Finally, by offering participants the opportunity to waive each of their decisions when judged as insufficiently accurate, we measured the effect of ketamine on decision uncertainty.

In this work, we present converging evidence that NMDA receptor hypofunction increases decision uncertainty but simultaneously drives a premature commitment to inaccurate decisions. Using computational modeling of inference and confidence, we show how this mechanism compensates for the elevated decision uncertainty generated by NMDA receptor hypofunction.

## Results

**Pharmacological protocol**. Healthy adult volunteers ($N = 20$) performed a visual cue combination task while being administered either ketamine or sodium chloride (placebo) intravenously, using a three-stage infusion protocol (Fig. 1). Each tested participant performed the task under ketamine (K+) and placebo (K−), during two experimental sessions taking place on separate days in a counterbalanced, double-blind order across participants. Our pharmacological protocol was developed using simulations of the pharmacokinetic parameters of a three-compartment model, and aimed at achieving constant plasma concentration of ketamine of 150 ng/mL during task execution, which started 30 min after infusion onset (Fig. 1a, see "Methods"). Two participants were excluded because they did not complete the task under ketamine, leaving $N = 18$ participants in all analyses.

Blood samples taken moments before and after task execution (at 30 min and ~90 min after infusion start) confirmed stable plasma concentration of ketamine close to target (Supplementary Fig. 1a; 30 min: $167.8 \pm 11.4$ ng/mL, mean $\pm$ s.e.m.; 90 min: $167.2 \pm 9.5$ ng/mL; paired $t$ test, $t_{17} = 0.1$, $p = 0.945$). Furthermore, ratings on psychiatric symptom scales showed known "dissociative" effects of ketamine[18] (Fig. 1b and Supplementary Fig. 1b, see "Methods"), accompanied by a mild elevation of blood pressure (Supplementary Fig. 1c).

**Cue combination task**. In every trial, participants observed a sequence of 4 to 12 visual stimuli, after which they were asked to indicate which category (among two possible ones) they judged the sequence to have been drawn from (Fig. 1c). Stimuli corresponded to oriented bars drawn from one of two overlapping probability distributions (categories) centered on orthogonal orientations, each associated with a color. The positions of the two colors on the circle varied randomly across trials, a task feature which decorrelated the orientation of stimuli from the evidence they provide to the decision process (see "Methods"). The difficulty of the task was adapted online using a titration procedure to reach similar decision accuracies under ketamine and placebo (Fig. 1d, e; ketamine: $76.1 \pm 0.9$%; placebo: $77.9 \pm 0.8$%; $t_{17} = -1.7$, $p = 0.106$).

At the end of every trial, participants were offered the possibility to opt out of their decision, in which case they were rewarded not based on the accuracy of their decision, but based on the outcome of a lottery of known probability of success which varied from trial to trial (Fig. 1c, see "Methods"). Because decision accuracy did not differ between conditions, we could use opt-out decisions to measure the effect of ketamine on decision uncertainty. Like decision accuracy, mean response times did not differ between ketamine and placebo (Fig. 1d; ketamine: $757 \pm 34$ ms; placebo: $761 \pm 30$ ms; $t_{17} = -0.2$, $p = 0.845$). Nevertheless, participants opted out of their decisions substantially more often under ketamine than placebo (Fig. 1d, e; ketamine: $42.9 \pm 2.5$%; placebo: $34.4 \pm 2.0$%; $t_{17} = 4.3$, $p < 0.001$). This finding suggests elevated decision uncertainty under ketamine, despite matched decision accuracy.

**Increased decision uncertainty under ketamine**. The decision of opting out of a decision should be made by comparing the expected accuracy of the decision based on the accumulated evidence to the probability of success of the presented lottery. The higher fraction of opt-out decisions observed under ketamine despite matched decision accuracy could thus reflect different effects: 1. weaker metacognitive sensitivity to the expected accuracy of the decision, 2. stronger reliance on the presented lottery than the expected accuracy, or 3. a lower decision criterion for opting out. To arbitrate between these different accounts, we constructed psychometric curves of opt-out rate as a function of the expected accuracy and the probability of success of the presented lottery (Fig. 2a, see "Methods").

Comparing these curves between conditions revealed a clear upward shift under ketamine (expected accuracy: $F_{1,17} = 13.9$, $p = 0.002$; lottery probability: $F_{1,17} = 18.5$, $p < 0.001$), but no measurable difference between their slopes (expected accuracy: $t_{17} = -1.1$, $p = 0.273$; lottery probability: $t_{17} = -1.0$, $p = 0.323$). Modeling the comparison between these two variables through logistic regression (Fig. 2b, see "Methods") captured the

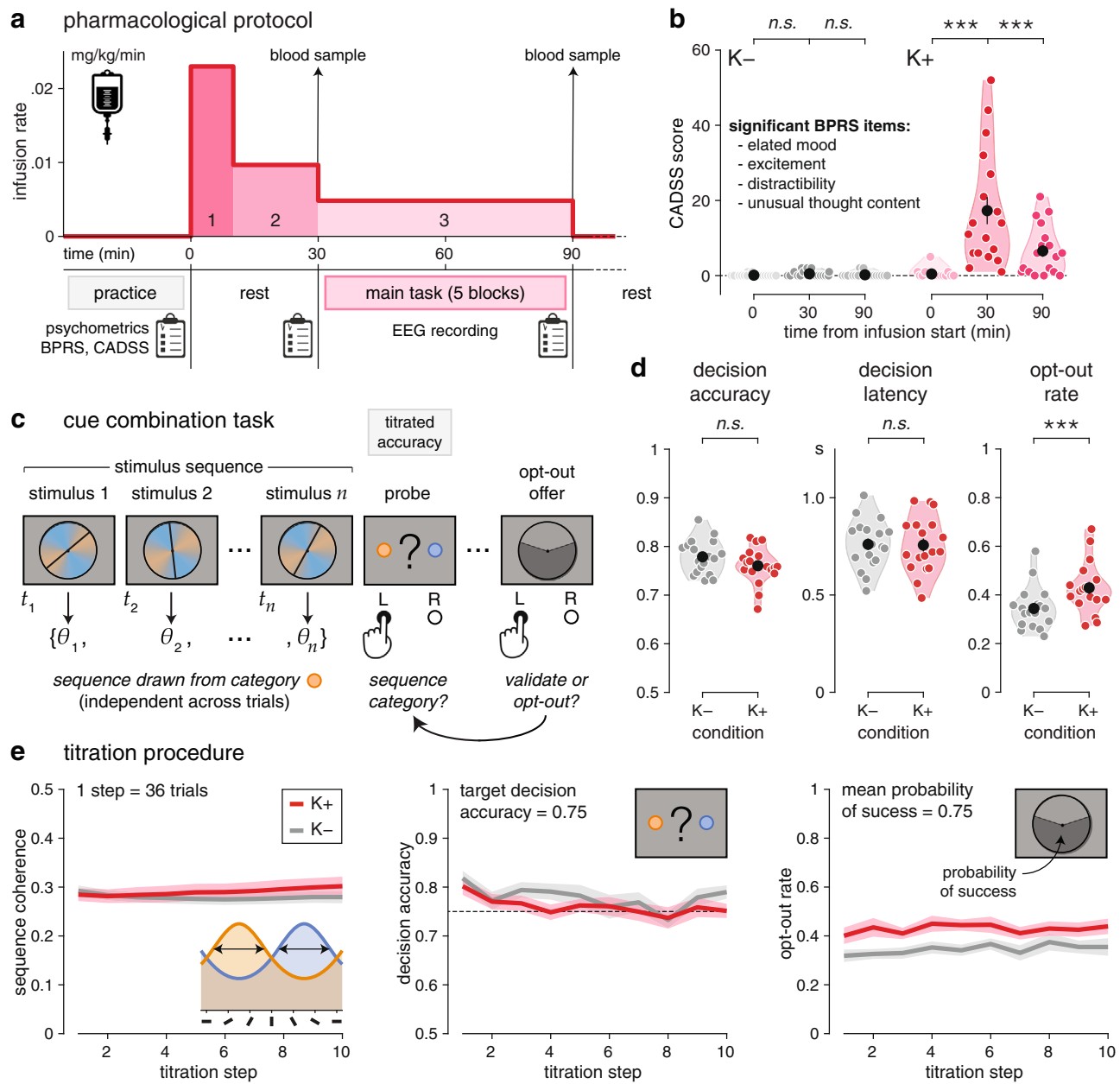

**Fig. 1 Pharmacological protocol and cue combination task. a** Description of the pharmacological protocol. Top: three-stage infusion protocol used to achieve constant plasma concentration of ketamine of 150 ng/mL during task execution. Bottom: task-related events. **b** Scores on the Clinician Administered Dissociative States Scale (CADSS). Only four items of the Brief Psychiatric Rating Scale (BPRS) show increased ratings under ketamine (K+) than placebo (K−). **c** Description of the cue combination task. Each trial consists of a sequence of 4 to 12 oriented stimuli $\{\theta_1, \ldots, \theta_n\}$ presented at an average rate of 2.5 Hz, after which participants are probed for a response (left- or right-handed) regarding the category of the sequence. Accuracy is titrated online by adjusting the coherence of the stimulus sequence—corresponding to the concentration of the distributions of orientations associated with each category. At the end of each trial, participants are offered the possibility to opt out of their decision, in which case they are rewarded not based on decision accuracy, but based on a lottery of described probability of success. **d** Task behavior (left: decision accuracy; middle: decision latency; right: opt-out rate). Participants opt out of their decisions more often under ketamine than placebo. **e** Titration procedure targeting a decision accuracy of 0.75. Left: temporal evolution of sequence coherence across titration steps (1 step = 36 trials). Middle: temporal evolution of decision accuracy across titration steps. Right: temporal evolution of opt-out rate across titration steps. Black dots and error bars indicate group-level means ± s.e.m., whereas colored dots indicate participant-level data. Three stars indicate a significant effect at $p < 0.001$, n.s. a non-significant effect ($N = 18$ participants, uncorrected two-sided $t$ tests). Source data are provided as a Source Data file.

difference between psychometric curves by a lower opt-out criterion under ketamine ($t_{17} = -4.6$, $p < 0.001$). These findings indicate that participants opted out of their decisions for higher expected accuracy (and, conversely, lower probability of success of the lottery) under ketamine—despite matched decision accuracy and accurate metacognitive evaluation. This effect is consistent with elevated decision uncertainty (i.e., lower subjective accuracy of decisions) under ketamine.

**Impaired cognitive inference under ketamine**. To measure the accuracy of cognitive inference under ketamine, we used a

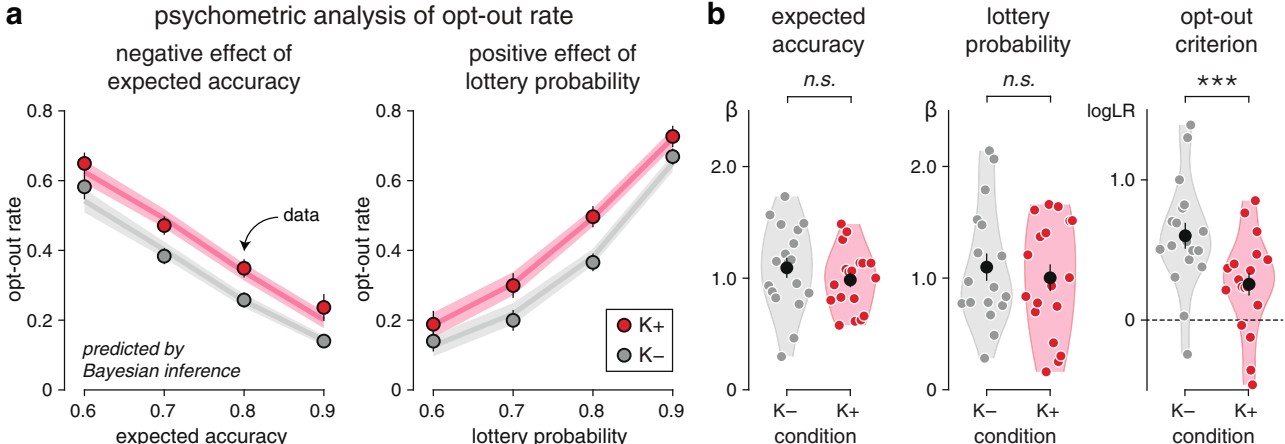

**Fig. 2 Increased decision uncertainty under ketamine. a** Psychometric analysis of opt-out rate. Effects of expected accuracy (left) and lottery probability (right) on opt-out rate. Dots indicate observations (group-level means ± s.e.m.), whereas lines indicate best fits. Psychometric curves show a clear upward shift under ketamine. **b** Best-fitting psychometric parameters (left: sensitivity to expected accuracy; middle: sensitivity to lottery probability; right: opt-out criterion). Participants are equally sensitive to expected accuracy and lottery probability between conditions, but show a lower opt-out criterion under ketamine. Black dots and error bars indicate group-level means ± s.e.m., whereas colored dots indicate participant-level data. Three stars indicate a significant effect at $p < 0.001$, *n.s.* a non-significant effect ($N = 18$ participants, uncorrected two-sided $t$ tests). Source data are provided as a Source Data file.

validated computational model[8] (Fig. 3a) which considers separate sources of internal errors during sensory processing, inference and response selection. We first looked for significant sources of sensory, inference and selection errors under placebo and ketamine using a factorized Bayesian model selection procedure (Fig. 3b, see "Methods"). This analysis, performed separately in the two conditions, identified a single source of internal errors located at the inference stage (placebo: exceedance $p = 0.998$; ketamine: exceedance $p = 0.996$). Neither sensory nor selection errors were detected in either condition (all exceedance $p < 0.010$). The presence of a single source of inference errors in both conditions, validated through a "model recovery" procedure[19] (Supplementary Fig. 2a), validates inference as the main cognitive "bottleneck" in this task.

To quantify the accuracy of this process, we then measured the magnitude of inference errors by two complementary means (Fig. 3c): 1. by estimating participants' decision sensitivity to the true evidence provided by each sequence of stimuli through logistic regression, and 2. by modeling and fitting the spread of inference noise (see "Methods"). These analyses yielded converging results: lower sensitivity to the true evidence under ketamine ($t_{17} = 2.7$, $p = 0.014$), along with a 19% increase in the spread of inference noise (placebo: $0.474 ± 0.030$; ketamine: $0.562 ± 0.046$; $t_{17} = 3.3$, $p = 0.004$). We confirmed these results through Bayesian model selection, by asking which source of internal errors was most likely to increase under ketamine (Fig. 3d; inference errors: exceedance $p = 0.987$). To further validate these results[19], we estimated the variability (inverse sensitivity) of participants' decisions and model simulations separately for each condition and each sequence length $n$ (4, 8 or 12 stimuli, Fig. 3d). As expected by the presence of inference errors which accumulate across stimuli[8], decision variability increased approximately linearly with sequence length (repeated-measures ANOVA, $F_{2,34} = 21.4$, $p < 0.001$). Furthermore, decision variability increased more rapidly under ketamine than placebo (interaction: $F_{2,34} = 3.7$, $p = 0.036$). Both effects were explained by increased inference errors under ketamine. Computing participants' psychophysical kernels using logistic regression (Supplementary Fig. 2b) indicated that ketamine did not strongly impair participants' ability to accumulate evidence over time. Indeed, ketamine dampened participants' psychophysical kernels

but did not selectively decrease the weight of stimuli presented early in each sequence. Accordingly, the moderate inference leak fitted using the model increased only marginally under ketamine (placebo: $0.054 ± 0.014$; ketamine: $0.082 ± 0.014$; difference: $t_{17} = 2.1$, $p = 0.051$).

Both the magnitude of inference errors and the opt-out criterion showed a strong within-participant reliability across conditions (inference errors: Pearson $r = 0.832$, d.f. $= 16$, $p < 0.001$; opt-out criterion: Pearson $r = 0.651$, d.f. $= 16$, $p = 0.003$). Furthermore, these two cognitive parameters correlated negatively with each other (Supplementary Fig. 2c; Pearson $r = -0.541$, d.f. $= 34$, $p < 0.001$): participants making more inference errors used a lower opt-out criterion than participants making less inference errors. We also found a significant relationship between the effects of ketamine on these two cognitive parameters (Supplementary Fig. 2c; Pearson $r = 0.636$, d.f. $= 16$, $p = 0.005$): large increases in inference errors were associated with large decreases in the opt-out criterion. This covariation suggests that the increase in opt-out rate triggered by ketamine does not reflect a task-unspecific effect of ketamine nor a biased subjective probability of success of the lottery, but rather a selective effect of the drug on the subjective accuracy of cognitive inference.

**Degraded neural processing of evidence under ketamine.** To identify the neural locus of ketamine effects on cognitive inference, we characterized the neural processing of each stimulus using time-resolved analyses of task-related EEG signals. Due to the rapid sequential presentation of stimuli in our task, standard event-related potentials (ERPs) cannot be interpreted without confounds (Supplementary Fig. 3). We have therefore relied on a neural coding approach which ties EEG signals to the processing of specific stimulus characteristics[20–23]. First, we described each stimulus $k$ (about 2400 per condition and per participant) by two distinct characteristics: 1. its orientation, and 2. the strength of the evidence it provides to the inference process. Because the orientations of category axes varied from trial to trial, these two characteristics were independent of each other. We then applied multivariate pattern analyses to EEG signals aligned to stimulus onset to estimate the neural "codes" associated with these two characteristics (see "Methods"). Owing to the fine temporal

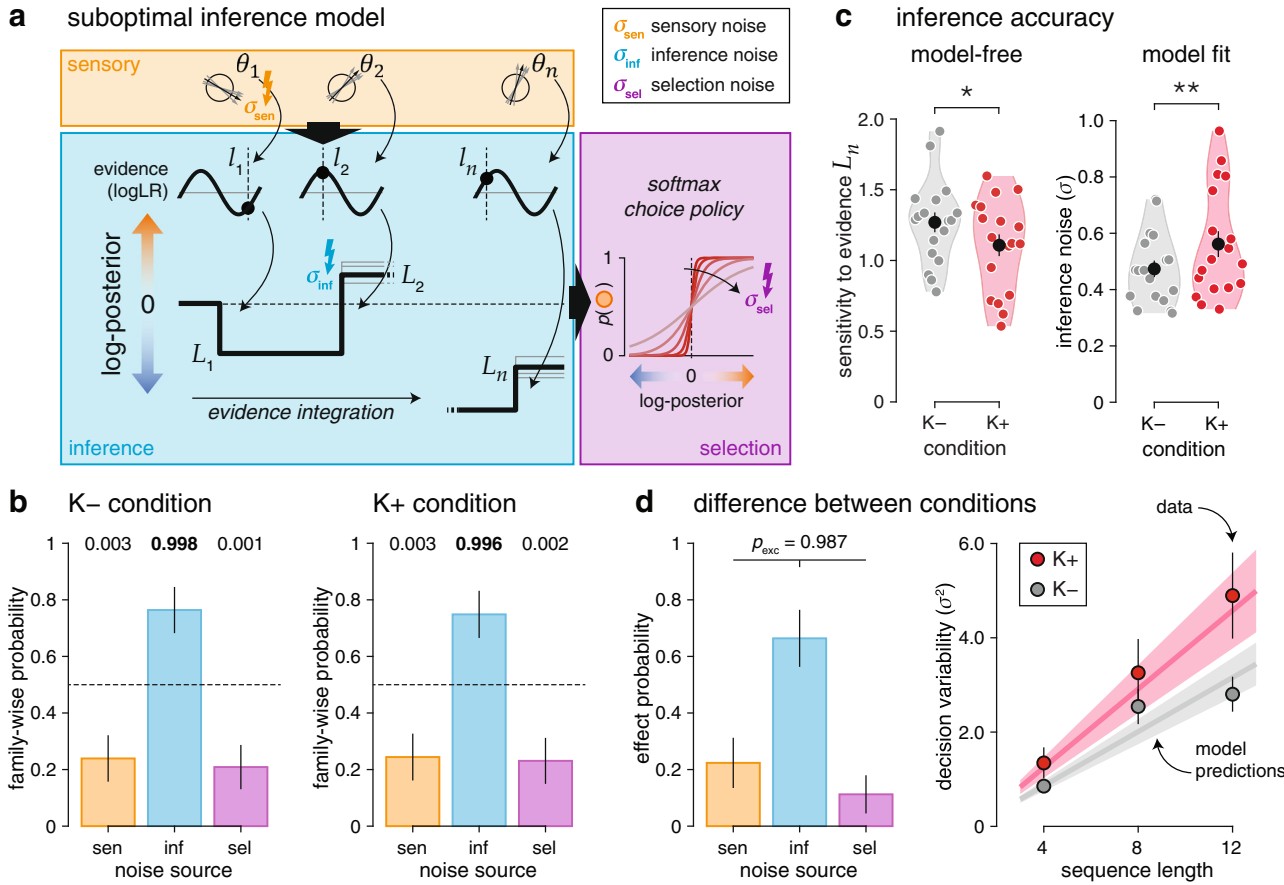

**Fig. 3 Impaired cognitive inference under ketamine. a** Suboptimal inference model. Bayes-optimal inference is achieved by estimating the evidence $l_k$ for each category (here, their log-likelihood ratio) provided by each stimulus of orientation $\theta_k$, accumulating this evidence across the $n$ stimuli to obtain a posterior $L_n$, and using this posterior to select the most likely category given the observed stimuli. Sensory noise is modeled by Gaussian noise of s.d. $\sigma_{sen}$ to each stimulus orientation $\theta_k$. Inference noise is modeled by Gaussian noise of s.d. $\sigma_{inf}$ to the evidence $l_k$ being accumulated. Selection noise is modeled by Gaussian noise of s.d. $\sigma_{sel}$ to the final posterior $L_n$. **b** Factorized Bayesian model selection procedure in the placebo (left) and ketamine (right) conditions. Family-wise probability is defined as the probability of each noise source (sensory, inference, selection) independently of the presence or absence of other noise sources. Bars and error bars correspond to means ± s.d. of estimated Dirichlet distributions. Values above each bar indicate associated exceedance probabilities. Participants show a single source of inference errors in both conditions. **c** Estimation of inference accuracy. Left: sensitivity to the Bayes-optimal posterior $L_n$. Right: best-fitting s.d. of inference noise $\sigma_{inf}$. Participants show lower sensitivity to the true evidence and increased inference noise under ketamine. Black dots and error bars indicate group-level means ± s.e.m., whereas colored dots indicate participant-level data. **d** Effect of ketamine on decision variability. Left: Bayesian model selection procedure for the difference between conditions. Effect probability is defined as the probability of each noise source to explain the increased decision variability under ketamine. Bars and error bars correspond to means ± s.d. of estimated Dirichlet distributions. The effect of ketamine is best explained by increased inference noise. Right: estimated decision variability as function of sequence length. Dots indicate observations (group-level means ± s.e.m.), whereas lines indicate best-fitting predictions of increased inference noise under ketamine. One star indicates a significant effect at $p < 0.05$, two stars at $p < 0.01$ ($N = 18$ participants, uncorrected two-sided $t$ tests). Source data are provided as a Source Data file.

resolution of EEG signals, we could extract the time course of neural information processing within the first hundreds of milliseconds following stimulus onset.

The neural coding of stimulus orientation (Fig. 4a, top) peaked at 100 ms following stimulus onset (jackknifed mean, ketamine: 99.2 ms; placebo: 99.5 ms), and did not differ between ketamine and placebo conditions (Supplementary Fig. 4a, b; peak precision: $t_{17} = 1.7$, $p = 0.109$). This neural code overlapped only slightly across successive stimuli (Supplementary Fig. 4c), and was supported by spectral content up to 16 Hz (measured as the frequency cutoff above which coding precision starts to decrease; see Supplementary Fig. 5). The intact orientation processing observed under ketamine is consistent with behavioral modeling, which did not identify any significant source of sensory errors in either condition (Fig. 3).

The neural coding of the evidence provided by the same stimuli (Fig. 4a, bottom) showed a very different picture: it peaked at 300 ms following stimulus onset (jackknifed mean, ketamine: 292.3 ms; placebo: 314.2 ms), and decreased substantially under ketamine from 250 to 450 ms following stimulus onset (peak $t_{17} = 4.8$, cluster-corrected $p < 0.001$), including its peak (Supplementary Fig. 4a,b; $t_{17} = -2.4$, $p = 0.026$). Furthermore, and in contrast to the neural code of stimulus orientation, the neural code of stimulus evidence overlapped strongly across successive stimuli (Supplementary Fig. 4c) and was supported by spectral content below 8 Hz (Supplementary Fig. 5). This degraded neural processing of stimulus evidence under ketamine mirrors the larger inference errors identified when modeling behavior (Fig. 3). To validate this "brain-behavior" relation[21,22], we tested whether the neural coding of stimulus evidence under ketamine correlated with

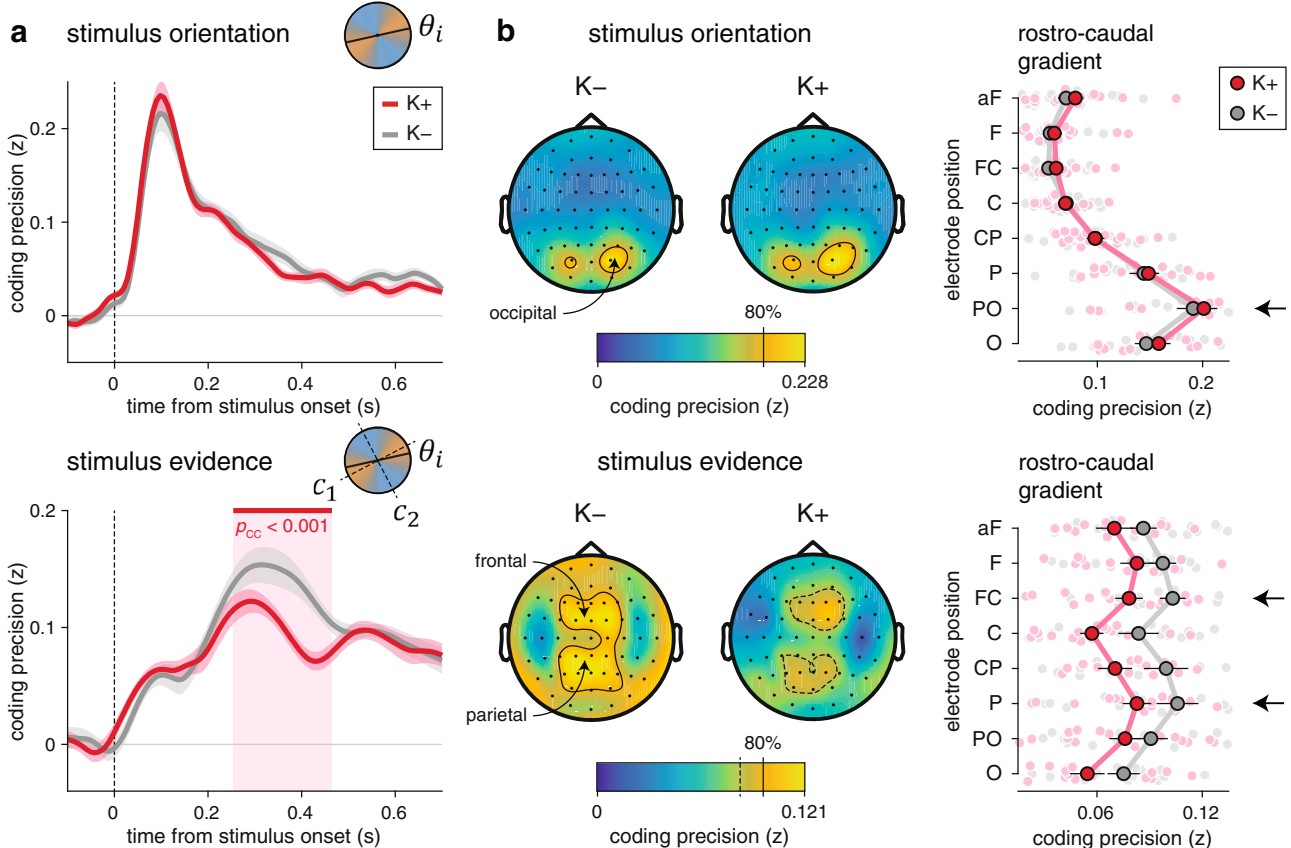

**Fig. 4 Degraded neural processing of evidence under ketamine. a** Time course of the neural coding of stimulus characteristics by EEG signals (top: stimulus orientation; bottom: stimulus evidence). Lines and shaded error bars indicate group-level means ± s.e.m. ($N$ = 18 participants). The neural coding of orientation is preserved under ketamine, whereas the neural coding of evidence is degraded from 250 to 450 ms following stimulus onset. The red-shaded area indicates the significant difference in coding precision between conditions (two-sided $t$ tests corrected for multiple comparisons across time). **b** Spatial topography of the neural coding of stimulus characteristics (top: stimulus orientation; bottom: stimulus evidence). Left: channel-wise coding precision, obtained by averaging neural predictions over time (stimulus orientation: 50–150 ms; stimulus evidence: 250–450 ms). Contours delineate channels with a coding precision larger than 80% of the overall peak. Right: coding precision along a rostro-caudal gradient, from occipital (O) to anterior frontal (aF) channels. Large dots and error bars indicate group-level means ± s.e.m., whereas small dots indicate participant-level data ($N$ = 18 participants). The neural coding of orientation relies on occipital channels, whereas the neural coding of evidence peaks at parietal and frontal channels (black arrows). Source data are provided as a Source Data file.

the contribution of the same stimulus to the upcoming decision (Supplementary Fig. 6a, see "Methods"). We found that stimuli associated with overestimated evidence in EEG signals contributed more strongly to the upcoming decision (Supplementary Fig. 6b,c; $\beta = 0.053 \pm 0.013$, $t_{17} = 4.3$, $p < 0.001$). This relation between neural and behavioral variability indicates that the neural coding of stimulus evidence reflects the noisy representation of momentary evidence being accumulated by participants.

In both conditions, the neural coding of stimulus orientation relied on occipital channels overlying visual cortex (Fig. 4b, top), with a strong rostro-caudal gradient in coding precision. By contrast, the neural coding of stimulus evidence showed in both conditions a much more distributed spatial topography, peaking at parietal and frontal channels overlying associative cortex (Fig. 4b, bottom). Cross-condition generalization analyses (Supplementary Fig. 7a, see "Methods") supported shared neural codes of each characteristic across conditions, and confirmed the lower signal-to-noise ratio for evidence processing under ketamine (Supplementary Fig. 7b, c; stimulus orientation: $t_{17} = 1.4$, $p = 0.178$; stimulus evidence: $t_{17} = -3.7$, $p = 0.002$). Together, these results indicate that ketamine degrades the neural processing of stimulus evidence at parietal and frontal channels, without any detectable alteration of orientation processing at occipital channels.

**Imbalanced neural processing of evidence under ketamine**. To determine whether ketamine degrades the neural processing of stimulus evidence irrespective of the ongoing inference process, we compared the neural coding of stimulus evidence between stimuli consistent with the decision provided at the end of the trial (i.e., the output of the inference process), and stimuli conflicting with the same decision (see "Methods"). A balanced inference process should assign equal weights to all stimuli irrespective of their consistency with the subsequent decision, whereas an imbalanced inference process should result in stronger weights for consistent than conflicting stimuli—a form of "circular" (self-reinforcing) inference.

The orientation of consistent and conflicting stimuli could be decoded equally well from EEG signals, in both conditions (Fig. 5a, left). By contrast, the evidence provided by conflicting stimuli could be decoded less precisely than the evidence provided by consistent stimuli from 115 to 400 ms following stimulus onset (Fig. 5a, right; peak $F_{1,17} = 11.0$, cluster-corrected $p < 0.001$). This coding unbalance was only present under ketamine, not placebo (Fig. 5b; ketamine: $t_{17} = -4.6$, $p < 0.001$; placebo: $t_{17} = -1.0$, $p = 0.342$; interaction: $F_{1,17} = 7.6$, $p = 0.014$). Examining the dynamics of coding imbalance over the course of each sequence (Fig. 5c) provided

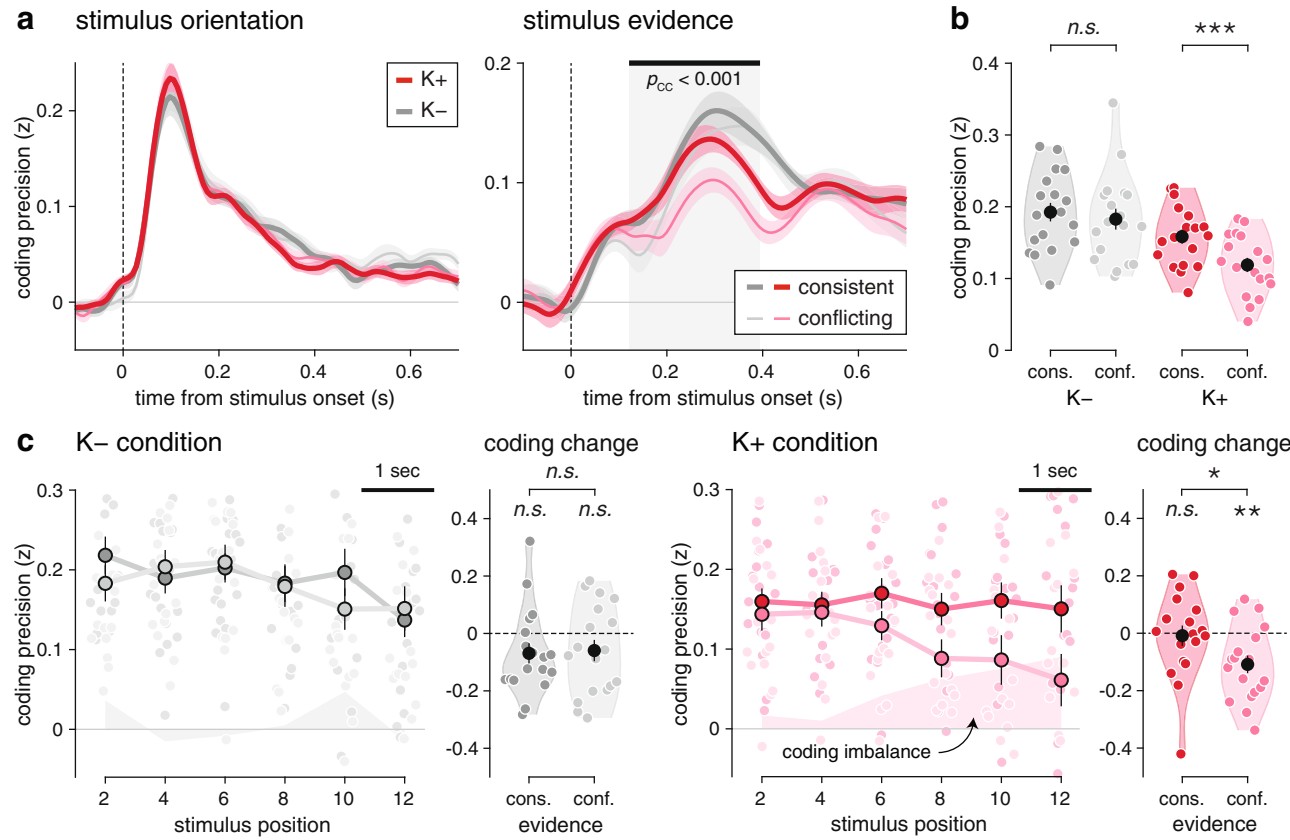

**Fig. 5 Unbalanced neural processing of evidence under ketamine. a** Effect of decision consistency on the neural coding of stimulus orientation (left) and stimulus evidence (right). Thick lines correspond to stimuli consistent with the upcoming decision, whereas thin lines correspond to stimuli conflicting with the upcoming decision. The neural coding of stimulus orientation does not differ between consistent and conflicting stimuli, whereas the neural coding of stimulus evidence is less precise for conflicting stimuli under ketamine. Lines and shaded error bars indicate group-level means ± s.e.m. (N = 18 participants). The gray-shaded area indicates the significant effect of decision consistency on coding precision (two-sided t tests corrected for multiple comparisons across time). **b** Effect of decision consistency on the neural coding of stimulus evidence in the time window showing a ketamine effect (250–450 ms following stimulus onset). Consistent and conflicting stimuli are coded equally precisely under placebo (left), whereas conflicting stimuli are coded less precisely under ketamine (right). **c** Dynamics of coding unbalance under placebo (left) and ketamine (right). Left: coding precision of stimulus evidence as a function of decision consistency and stimulus position in the sequence. Shaded areas indicate the strength of coding unbalance (i.e., the difference in coding precision between consistent and conflicting stimuli). Right: coding change with stimulus position for consistent and conflicting stimuli. The neural coding of consistent and conflicting stimuli is stable over time under placebo, whereas the neural coding of conflicting stimuli decreases over time under ketamine. One star indicates a significant effect at $p < 0.05$, two stars at $p < 0.01$, three stars at $p < 0.001$, n.s. a non-significant effect (N = 18 participants, uncorrected two-sided t-tests). Source data are provided as a Source Data file.

additional information. Evidence coding was balanced throughout each sequence under placebo, corresponding to a steady neural coding of consistent and conflicting stimuli (coding change, consistent: $\beta = -0.069 \pm 0.035$, $t_{17} = -2.0$, $p = 0.067$; conflicting: $\beta = -0.060 \pm 0.037$, $t_{17} = -1.6$, $p = 0.127$; difference: $t_{17} = 0.3$, $p = 0.746$). By contrast, the coding unbalance observed under ketamine increased throughout each sequence, due to a decrease in the neural coding of conflicting evidence (consistent: $\beta = -0.009 \pm 0.035$, $t_{17} = -0.2$, $p = 0.814$; conflicting: $\beta = -0.109 \pm 0.031$, $t_{17} = -3.6$, $p = 0.002$; difference: $t_{17} = 2.2$, $p = 0.044$). Importantly, the dependence of stimulus processing on other contextual variables such as the degree of change between successive stimuli (Supplementary Fig. 8) showed no difference between conditions. This pattern of effects indicates that ketamine dampens selectively the neural processing of conflicting evidence.

**Premature response preparation under ketamine.** The processing imbalance observed under ketamine is expected to drive less

accurate, but also faster decisions in theories of NMDA receptor hypofunction. Here, participants were required to wait until a "go" signal to provide their response (Fig. 1c)—an instruction which they followed in both conditions (see "Methods"). We therefore looked for covert response preparation activity in band-limited EEG power[24,25], a well-validated measure which we could compare between conditions (Fig. 6, see "Methods").

In agreement with previous work[26,27], we found that ketamine decreased baseline power in the alpha (10 Hz) and beta (20 Hz) bands and dampened the strong power suppression in these frequency bands during visual stimulation (Supplementary Fig. 9a, b). Despite these broad effects of ketamine, the additional power suppression triggered by response execution in the alpha band (10 Hz) at bilateral central channels (Fig. 6a) did not differ between conditions. We could thus use this motor signal to predict the provided response (left- vs. right-handed) in the last few seconds preceding its execution—even before the presentation of the "go" signal which probed participants for a response.

Because each sequence consisted of a variable number of ambiguous (often conflicting) stimuli, participants should wait

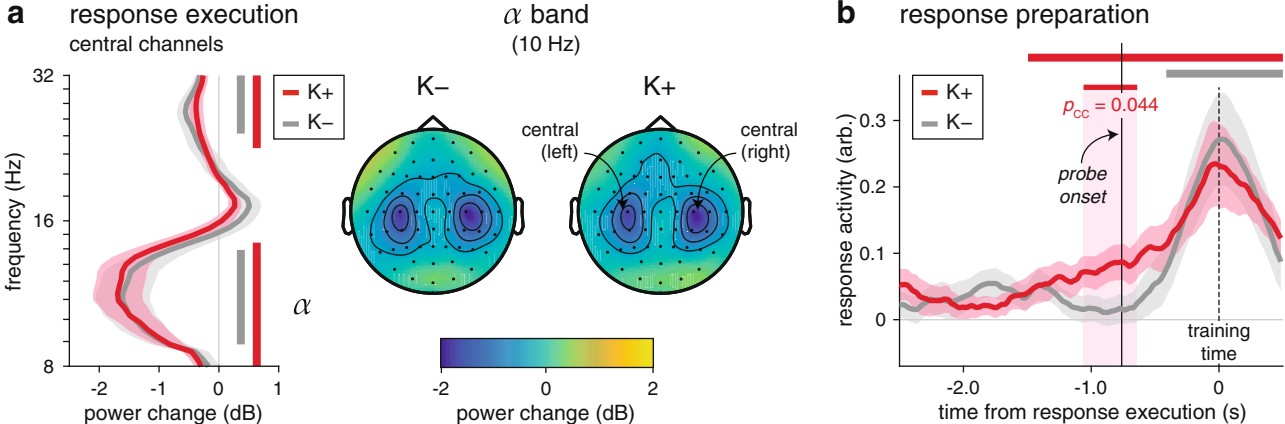

**Fig. 6 Premature response preparation under ketamine. a** Suppression of alpha power at central channels during response execution. Left: power change (x axis) between 8 and 32 Hz (y-axis) between response execution and the preceding stimulus sequence at central sites. The power suppression peaking in the alpha band (10 Hz) is identical between conditions. Right: spatial topography of the suppression of alpha power during response execution. The suppression of alpha power peaks at bilateral central channels overlying human motor cortex, and is identical between conditions. **b** Time course of alpha power projected on the response-predictive axis (response activity) in the last 2.5 s preceding response execution. Response activity predicts the upcoming response well before the "go" signal (probe onset) under ketamine (red curve), but not placebo (gray curve). Response activity is stronger under ketamine surrounding the onset of the "go" signal (indicated by the red-shaded area, two-sided t tests corrected for multiple comparisons across time). Shaded error bars indicate s.e.m. (N = 18 participants). Source data are provided as a Source Data file.

until the "go" signal to prepare and execute their response. First, we verified that the provided response could be decoded from alpha power during response execution (placebo: $t_{17} = 3.9$, $p < 0.001$; ketamine: $t_{17} = 3.7$, $p < 0.001$; difference: $t_{17} = -0.5$, $p = 0.599$). Second, the time course of alpha power projected on this response-sensitive axis matched the expected profile of response preparation under placebo (Fig. 6b and Supplementary Fig. 9c, d). Indeed, the upcoming response could be predicted from alpha power only in the last 400 ms before response execution (peak $t_{17} = 3.9$, cluster-corrected $p < 0.001$)—well after the "go" signal. By contrast, under ketamine, the upcoming response could be predicted over a much longer time period extending well before the "go" signal, from 1460 ms before response execution (peak $t_{17} = 5.2$, cluster-corrected $p < 0.001$). These differences resulted in stronger response activity under ketamine from 1080 to 660 ms before response execution (peak $t_{17} = 2.8$, cluster-corrected $p = 0.044$)—including the onset of the "go" signal ($t_{17} = 2.4$, $p = 0.030$). This premature response preparation is consistent with the early decision times predicted by theories of NMDA receptor hypofunction.

**Premature commitment model of ketamine effects**. We observed a specific set of neural alterations under ketamine: (1) a degraded and progressively unbalanced processing of stimulus evidence in associative cortex (Fig. 5), and (2) a premature response preparation in motor cortex (Fig. 6). To characterize their origin, we developed a process-level account of ketamine effects which makes testable empirical predictions (Fig. 7a). Our model proposes that ketamine triggers a premature commitment to inferred beliefs in the middle of some trials: the covert selection of the category supported by the accumulated evidence before being probed for a decision (a form of "jumping to conclusions"), followed by the selective integration of evidence consistent with the selected category and the discarding of evidence conflicting with the selected category (a form of "confirmation bias").

We simulated the effects of these premature commitments on cognitive inference by perturbing the noisy inference model which best describes participants' behavior under placebo (see "Methods"). We found that occasional premature commitments (corresponding to a probability $p$ of 5% following each stimulus) were sufficient to

reproduce all identified ketamine effects: (1) a moderate increase in inference errors (Figs. 7b), (2) a progressively imbalanced coding of stimulus evidence (Fig. 7c), and (3) a premature response preparation, assuming that this motor activity reflects the covert initiation of the response after commitment to a category (before the "go" signal when a premature commitment has occurred).

**Relation between premature commitments and decision uncertainty**. Similar "jumping to conclusions" effects have already been described, but in participants or conditions associated with high decision confidence[14,15], something which stands at odds with the increased decision uncertainty observed here under ketamine (Fig. 2). To better understand the relation between premature commitments and decision uncertainty, we simulated the model for different probabilities of premature commitments and different levels of decision uncertainty (Fig. 7d). We reasoned that a reduced coding of conflicting stimuli (which should result in a reduced contribution of these stimuli to the subsequent decision) should mechanically increase the amount of accumulated evidence, and therefore decrease decision uncertainty. Simulations showed that premature commitments indeed provide an effective compensation for increased decision uncertainty: the presence of premature commitments reduces the overall probability of opting out (Fig. 7d) and, conversely, the neural signatures of premature commitments are more pronounced in trials ending with a validation of the initial decision than trials ending with opt-out (Supplementary Fig. 10a, b).

In simulations, occasional premature commitments decrease only moderately the overall opt-out rate (from 0.465 to 0.441 for a probability $p$ of 5%). In other words, premature commitments provide only a partial compensation for the large increase in opt-out rate triggered by the elevated decision uncertainty observed under ketamine (from 0.341 to 0.465 in simulations without premature commitments). However, on the trials where a premature commitment has occurred, simulations show a strong reduction in opt-out rate (0.286 instead of 0.465 in simulations without premature commitment).

In the EEG data, we first examined whether the imbalanced neural coding of stimulus evidence was more pronounced in trials ending with a validation of the initial decision (Fig. 8a). As

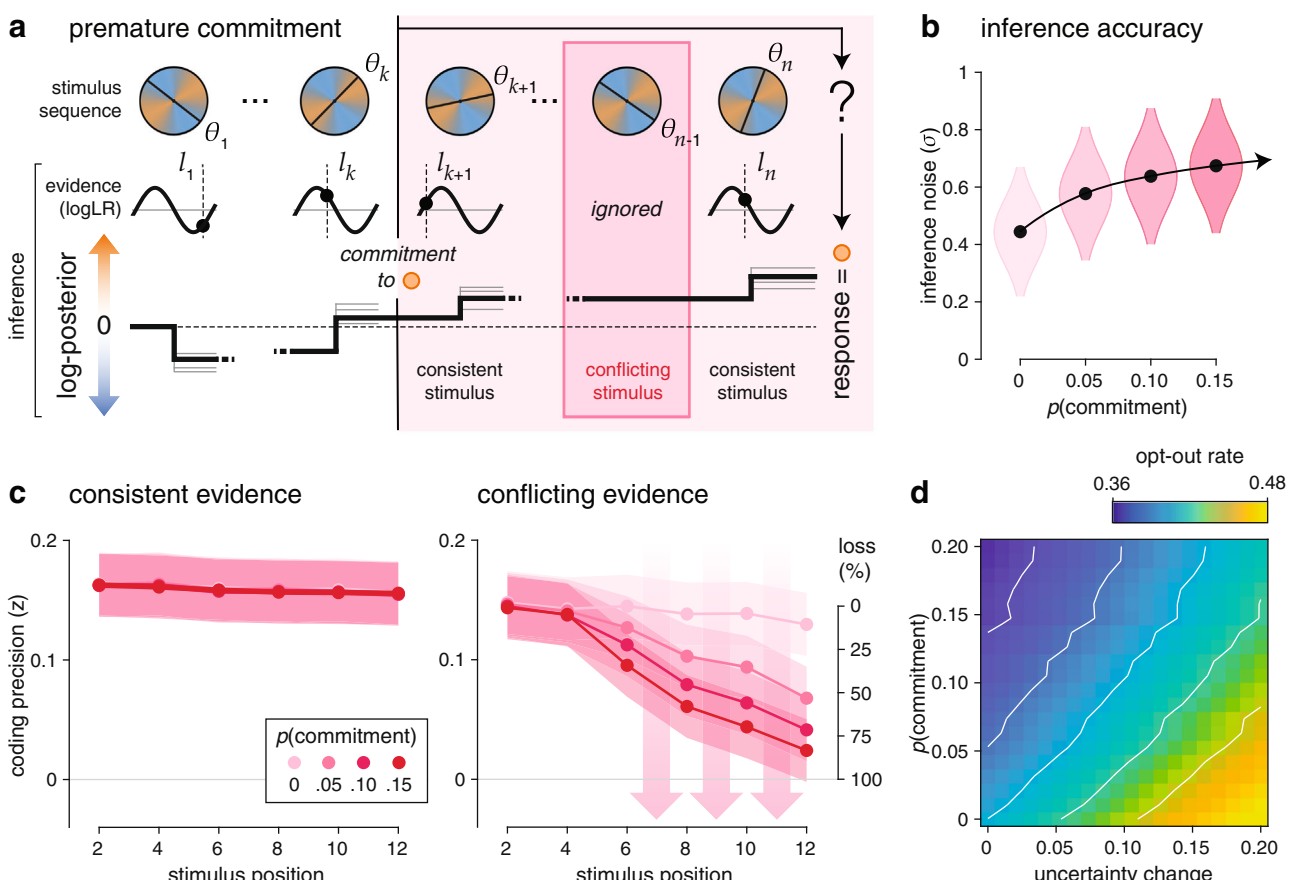

**Fig. 7 Premature commitment model of ketamine effects. a** Description of premature commitments during cognitive inference. Premature commitment to the category supported by the current belief can occur throughout the sequence with a constant hazard rate $p$(commitment) after at least $n_0$ stimuli (here, $n_0 = 4$). The premature commitment (occurring here after stimulus $k$), triggers the selective integration of stimuli consistent with the selected category, and the discarding of conflicting stimuli (here, stimulus $n - 1$). When a premature commitment has occurred (a fraction of all trials), the provided response is always in direction of the selected category (here, orange). **b** Simulated effect of premature commitments on inference accuracy. An increasing probability of premature commitments ($x$ axis, from 0% to 15%) is associated with a moderate increase in apparent (best-fitting) inference noise ($y$ axis). All simulations use the same spread of inference noise measured under placebo. Black dots and shaded violins indicate group-level means ± s.d. ($N = 18$ participants). **c** Simulated effect of premature commitments on coding unbalance for stimuli consistent (left) or conflicting (right) with the decision made by the model at the end of the trial. Coding precision uses the noisy representation of stimulus evidence computed by the model, for increasing probabilities of premature commitments (in shades of red). An increasing probability of premature commitments produces a selective decrease in the precision of conflicting evidence over the course of each sequence. **d** Compensatory effect of premature commitments on opt-out rate. Effects of changes in decision uncertainty (opt-out criterion, $x$ axis) and probability of premature commitments ($y$ axis) on opt-out rate. Increasing probabilities of premature commitments decrease the opt-out rate. Dots and shaded error bars (or violins) indicate group-level means ± s.e.m. of coding precision (or coding change) estimates across trials, averaged across participants ($N = 18$ participants). Source data are provided as a Source Data file.

predicted, these trials showed strong coding imbalance from 110 to 340 ms following stimulus onset (peak $F_{1,17} = 16.4$, cluster-corrected $p < 0.001$), significant under ketamine (160–400 ms, peak $t_{17} = 4.1$, cluster-corrected $p < 0.001$) but also placebo (110–290 ms, peak $t_{17} = 3.5$, cluster-corrected $p = 0.014$). By contrast, trials where participants opted out from their decisions did not show significant coding imbalance, even at the time point level. As in simulations (Supplementary Fig. 10a), ketamine and decision validation had non-interacting relations with coding imbalance (Fig. 8b; main effect of ketamine: $F_{1,17} = 8.2$, $p = 0.010$; main effect of decision validation: $F_{1,17} = 4.9$, $p = 0.040$; interaction: $F_{1,17} = 0.1$, $p = 0.805$).

Second, we tested whether the premature response preparation observed under ketamine was more prominent in trials ending with a validation of the initial decision (Fig. 8c). As predicted (Supplementary Fig. 10b), these trials were associated with significant response activity well before the "go" signal under ketamine (from 1830 ms before response execution), but only

after the "go" signal under placebo (from 390 ms before response execution). Response activity was significantly stronger under ketamine than placebo in a large time window surrounding the "go" signal (peak $t_{17} = 3.5$, cluster-corrected $p = 0.012$). By contrast, trials where participants opted out from their decisions did not show significant response activity before the "go" signal in either condition. To test the timings of these effects, we estimated the onset of the rise in response activity rise in each condition (Fig. 8d, see "Methods"). This analysis supported the premature rise of response activity under ketamine in trials ending with a validation of the initial decision (interaction: jackknifed $F_{1,17} = 6.4$, $p = 0.022$; validation: jackknifed $t_{17} = 2.7$, $p = 0.015$; opt-out: jackknifed $t_{17} = -0.4$, $p = 0.681$).

Our proposed model predicts that premature commitments provide effective compensation for the elevated uncertainty triggered by ketamine at a "fast" time scale within each trial, but does not make specific predictions regarding the temporal unfolding of the different effects of ketamine at a "slow" time

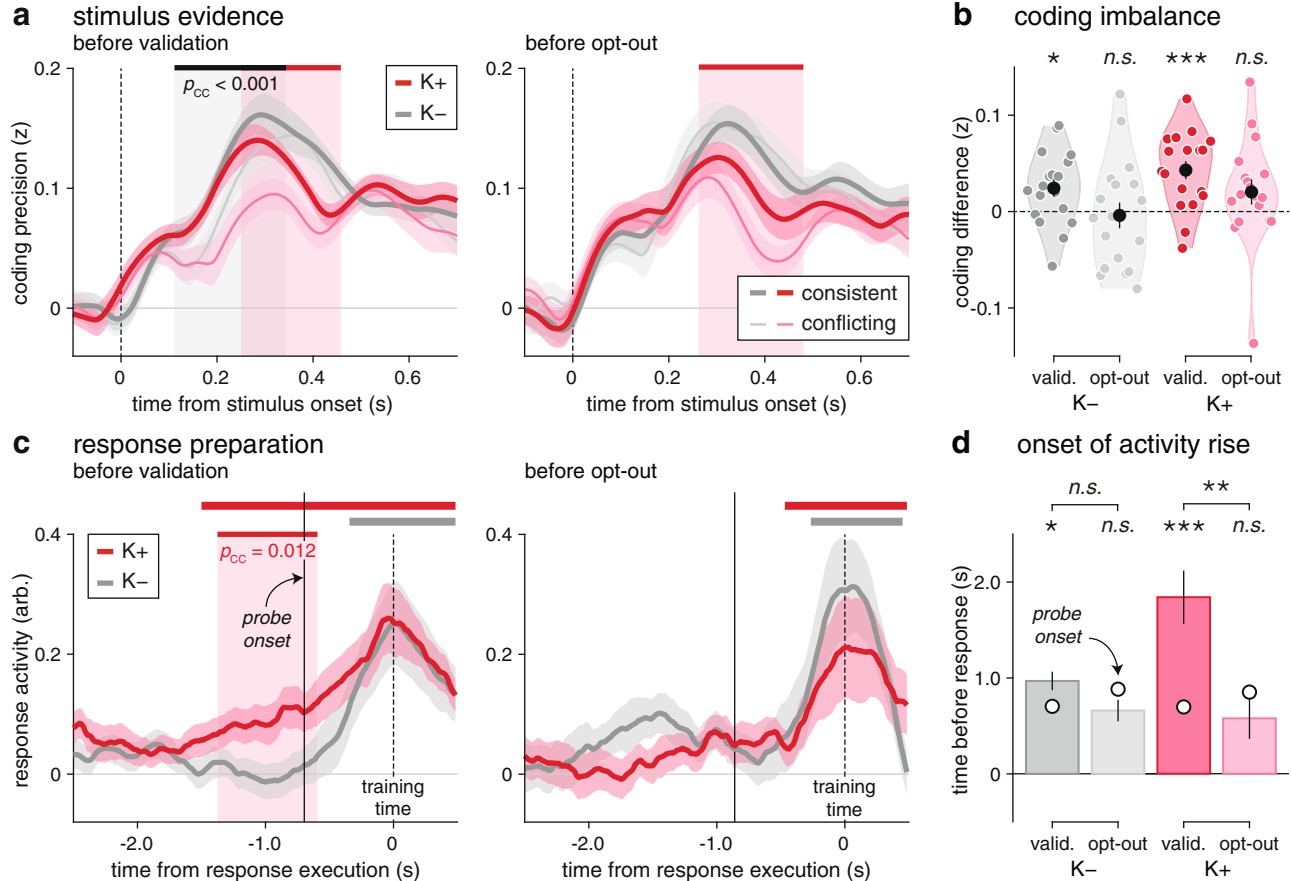

**Fig. 8 Relation between premature commitments and decision uncertainty. a** Relation between coding unbalance and decision validation vs. opt-out. Coding precision of stimulus evidence for consistent (thick lines) and conflicting stimuli (thin lines), in trials ending with decision validation (left) or opt-out (right). Lines and shaded error bars indicate group-level means ± s.e.m. (N = 18 participants). The gray-shaded area indicates the significant effect of decision consistency, whereas the red-shaded areas indicate the significant effects of ketamine (two-sided t tests corrected for multiple comparisons across time). Coding imbalance is more pronounced in trials ending with decision validation. **b** Non-interacting relations of ketamine and decision validation with coding unbalance, averaged in the time window where it is significant across conditions (115–400 ms following stimulus onset). **c** Relation between response preparation and decision validation vs. opt-out. Response activity in the last 2.5 s preceding response execution, in trials ending with decision validation (left) or opt-out (right). Premature response preparation (significant response activity before the "go" signal) is only detected under ketamine in trials ending with decision validation (two-sided t tests corrected for multiple comparisons across time). **d** Estimated onsets of response activity rise. Response activity rises well before the "go" signal under ketamine in trials ending with decision validation. Bars and error bars indicated jackknifed group-level means ± s.e.m. (N = 18 participants). One star indicates a significant effect at p < 0.05, two stars at p < 0.01, three stars at p < 0.001, n.s. a non-significant effect (N = 18 participants, uncorrected two-sided t tests with jackknife adjustment for onsets of response activity rise). Source data are provided as a Source Data file.

scale across trials in the experiment. To assess whether the different effects of ketamine are sustained across time, we computed them separately for the first and second halves of the experiment. All reported effects were sustained across time. In terms of behavior, the effect of ketamine on opt-out rate (Fig. 1d) was sustained across the two halves of the experiment (main effect of ketamine: $F_{1,17} = 18.3$, $p < 0.001$; interaction with time: $F_{1,17} = 0.4$, $p = 0.511$). Similarly, the effect of ketamine on inference noise (Fig. 3c) was sustained across time in the experiment (main effect of ketamine: $F_{1,17} = 10.5$, $p = 0.005$; interaction with time: $F_{1,17} = 0.7$, $p = 0.428$). In the EEG data, the imbalanced coding of evidence (Fig. 5b) observed under ketamine was also sustained across time (main effect of stimulus consistency on coding precision: $F_{1,17} = 24.8$, $p < 0.001$; interaction with time: $F_{1,17} < 0.1$, $p = 0.993$). Similarly, the premature response preparation activity (Fig. 6b) observed under ketamine was significant in both halves of the experiment (1st half: $t_{17} = 2.3$, $p = 0.033$; 2nd half: $t_{17} = 3.1$, $p = 0.006$). No such activity was detectable under placebo in either half (1st half:

$t_{17} = 1.0$, $p = 0.343$; 2nd half: $t_{17} = 0.3$, $p = 0.779$). This effect of ketamine on response preparation was also sustained across time (main effect of ketamine: $F_{1,17} = 8.3$, $p = 0.010$; interaction with time: $F_{1,17} < 0.1$, $p = 0.921$).

## Discussion
Theoretical work has conferred a central role to NMDA receptors in the computational accuracy of neural circuits implementing cognitive inference. Here, by administering sub-anesthetic infusions of ketamine to healthy volunteers engaged in a cognitive task requiring the accumulation of multiple pieces of evidence, we provide direct experimental support for a causal effect of human NMDA receptor hypofunction on cognitive inference. Ketamine produced a specific pattern of neurocognitive alterations, consistent with a premature commitment to inaccurate decisions. In contrast to existing accounts, we propose that this effect aims at resolving the abnormal uncertainty triggered by NMDA receptor hypofunction.

Our experimental protocol was designed to measure the precision of inference and estimate the effect of NMDA receptor hypofunction on this cognitive variable. First, the testing of the same volunteers under ketamine and placebo increased our ability to detect differences between conditions by controlling for individual differences. Second, our cue combination task and validated model[8] allowed dissociating inference errors from sensory noise and selection errors. Instead of measuring decision uncertainty using confidence reports—which could be heavily biased by the dissociative effects of ketamine visible in ratings on psychiatric symptom scales—we relied on the titrated comparison of decision uncertainty with a lottery of known probability of success. Together, these distinctive features of our experimental protocol revealed that ketamine triggers a selective increase in inference errors and an elevated decision uncertainty.

Theoretical[11,12] and empirically motivated[28,29] models describe the impact of NMDA receptor hypofunction on associative, context-dependent processes required for evidence accumulation and accurate decisions. In agreement with these views, we found that ketamine impaired the neural coding of stimulus evidence at frontal and parietal channels at 300 ms following stimulus onset. This impairment indicates that NMDA receptor hypofunction affects the estimation of the momentary evidence provided by each stimulus as a function of category axes which varied randomly across trials—a flexible, context-dependent process known to be supported by neural circuits in parietal and prefrontal cortices[30]. Importantly, this impaired estimation of momentary evidence occurs upstream from the integration of momentary evidence over time—the second process constitutive of cognitive inference[8].

The lack of ketamine effect on orientation coding supports the idea that NMDA receptor hypofunction impairs cognitive inference rather than all aspects of stimulus processing in an undifferentiated fashion. Indeed, orientation coding at occipital channels showed the same strength and latency under ketamine and placebo, and was associated with the same, well-known "repetition suppression" effects described in the literature[31]. The selectivity of observed ketamine effects indicates that the drug—which triggered "dissociative" effects on psychiatric symptom scales—did not merely distract participants from the task. Nevertheless, the lack of sensory effect of ketamine in our task does not imply that ketamine does not produce any sensory deficit in other contexts. Indeed, our task is based on visual stimuli with little to no sensory uncertainty (highly contrasted, without external noise nor temporal masking) and thus sets cognitive inference—i.e., the estimation and integration of evidence provided by successive stimuli—rather than sensory processing as the main cognitive bottleneck on task performance. Further research should employ tasks triggering high sensory uncertainty (e.g., random-dot motion stimuli) to determine whether ketamine can also produce sensory effects beyond its selective effects on cognitive inference observed in our task.

NMDA receptors have been proposed to enable balanced computations in associative cortical circuits, be it through their slow excitation of downstream neurons[32] or other distinguishing features (e.g., their nonlinear gain, their voltage-dependent blockade or their calcium permeability). Indeed, selective inactivation of these receptors results in early decisions that are based on few pieces of evidence[11]. In our task, we found that human NMDA receptor hypofunction does not impair cognitive inference in an unspecific fashion, but produces an imbalanced coding of stimulus evidence. Indeed, under ketamine, participants appeared to assign stronger weight to evidence for one category than the other, thereby increasing its likelihood to be selected at the end of the trial. This imbalanced weighting of evidence, particularly when associated with premature decisions, is seen as

a signature of "confirmation bias" and "circular inference", two mechanisms described in schizophrenia[33,34]. A similar bias has also been reported in the general population, but only after an initial response[35–38]. Here, our participants showed a similar effect under ketamine before any overt response has been made.

Subthreshold response preparation has been repeatedly reported during evidence accumulation in several sensorimotor regions, from the macaque parietal cortex[39,40] to the human motor cortex[24,25]. However, this effect is observed either in "free-response" conditions, or in conditions using a fixed number of stimuli where the "go" signal can be easily anticipated. Under ketamine, our participants showed a similar anticipatory effect in a "cued-response" condition using a variable number of stimuli—i.e., where the "go" signal cannot be anticipated. This early response preparation is maladaptive in such conditions, since the evidence presented late in a trial can sway the decision toward the other category. We propose that the co-occurrence of these two effects of ketamine—imbalanced weighting of evidence and premature response preparation—arises from premature commitments to uncertain beliefs about the category of the stimulus sequence in the middle of some trials.

In simulations, premature commitments alter the shape of psychophysical kernels by decreasing the weights of stimuli presented late in each sequence—i.e., after premature commitments have occurred (Supplementary Fig. 10c). This predicted effect stands in apparent contradiction with participants' psychophysical kernels under ketamine, which did not show such an effect (Supplementary Fig. 2b). However, the shape of psychophysical kernels does not only depend on premature commitments, but also on the ability to accumulate evidence over time without leak (a measure of working memory in our task). Therefore, the similar shapes of participants' psychophysical kernels under ketamine and placebo—despite the presence of premature commitments under ketamine—suggest that ketamine may impair working memory (as described in previous work[41]) and therefore increase the leak in evidence accumulation (Supplementary Fig. 10d). Future research should further examine the relation between these two cognitive effects of ketamine.

Premature commitments—also described as "jumping to conclusions"—are often seen as a signature of overconfidence. Our findings are incompatible with this view, since our participants showed elevated decision uncertainty under ketamine. However, it does not mean that premature commitments do not affect decision uncertainty. Indeed, premature commitments provide a form of effective compensation for the increased decision uncertainty over the course of a single inference process (a single trial in our task), by ignoring the evidence which conflicts with the pre-committed decision. As predicted by simulations, we show that the neural signatures of premature commitment were more pronounced in trials ending with decision validation than opt-out.

This effective compensation predicts a progressive transition from underconfidence when presented with few pieces of evidence—as it is the case in our task—to overconfidence when presented with many pieces of evidence over longer periods of time—as it is typically the case outside the laboratory. Because premature commitments increase the amount of accumulated evidence by filtering out conflicting evidence (Supplementary Fig. 11a), they lead progressively to overconfidence and decreased confidence sensitivity to objective decision accuracy (Supplementary Fig. 11b). In the general population, such "over-compensatory" mechanisms to uncertainty have been contemplated at longer time scales to explain the hardening of attitudes across different areas of psychology, including religion[42], attitudes toward capital punishment[43] and belief in conspiracy theories[44]. Even at short time scales, lacking control (a major source of

cognitive uncertainty) triggers a wide range of illusory percepts—from seeing images in random noise to forming illusory correlations in stock market fluctuations[45]. Here, in line with these different observations, we formally characterize how NMDA receptor hypofunction triggers a premature commitment to inaccurate decisions when confronted with high levels of uncertainty. Note, however, that our computational model remains agnostic about the nature of the relation between these two effects of ketamine. Also, the fact that both effects were sustained across the two halves of the experiment does not rule out a possible delay between their onsets at a different time scale.

Furthermore, low-dose infusions of ketamine have been proposed as a pharmacological model of symptoms observed at early stages of psychosis[16,28,29]—a stage where patients report elevated uncertainty rather than the overconfidence characteristic of later stages of the disease. This early stage of the disease is extremely difficult to study because patients are typically not receiving health care before developing a full-blown psychosis. The two effects of ketamine observed in our task could potentially reconcile these two stages in a unified account: an initial state of high uncertainty[46] and premature commitments[47] which progressively results in overconfidence and a weaker relation between confidence and decision accuracy—both of which being characteristic of psychosis[48]. Confirming this speculative hypothesis will require testing patients diagnosed with schizophrenia in our task at different stages of their illness. Such clinical investigation could clarify whether "jumping to conclusions" (a form of premature commitments) aims at resolving the abnormal uncertainty experienced by patients at early stages of their illness. It may also ultimately help determine whether treatment at early stages of psychosis should aim at increasing tolerance to uncertainty[49], through psychotherapeutic[50] or pharmacological[51] approaches.

## Methods

**Participants**. A total of 24 adult participants were recruited through public advertisement in the Paris area. The recruited participants were between 18 and 39 years of age, right-handed, reported no history of neurological or psychiatric disease, and no family history of psychotic disorders. Participants reported no addiction to psychoactive drugs, nor history of psychotropic medication. Participants had normal or corrected-to-normal vision, no history of cardiac disease, and blood pressure under 140/90 mmHg. Female participants were not pregnant nor breastfeeding. Participants provided written informed consent and received a flat €250 in compensation for their participation in the study. The study received ethical approval from relevant authorities: the Agence Nationale de Sécurité du Médicament et des Produits de Santé and the Comité de Protection des Personnes Ile-de-France III (ID RCB: 2013-002056-33). The study was registered under reference NCT02235012.

The first three participants were used to pilot the different aspects of the study (including parameters of the pharmacological protocol and those of the cue combination task). The EEG data of a fourth participant were lost, and two additional participants did not complete the cue combination task under ketamine as a result of adverse effects. The first participant had a vasovagal episode with hypotension, vomiting, but no loss of consciousness. The episode ended spontaneously a few minutes after the infusion was stopped and the participant was placed in a supine position with the legs elevated. The second participant experienced nausea. The symptoms likewise stopped spontaneously shortly after the infusion was stopped and did not require additional treatments. $N = 18$ participants were thus included in all analyses (7 females, age: $26.8 \pm 5.5$ years). This sample size is similar to those generally employed for comparable studies[16].

**Pharmacological protocol**. The study relies on a double-blind, placebo-controlled, randomized crossover protocol. Each participant performed the cue combination task described below twice, once under ketamine and once under placebo (sodium chloride), during two experimental sessions taking place on separate days. The allocation to receive ketamine (or placebo) on the first session was randomized using a randomization block size of two participants. Both participants and experimenters were blind to this allocation. For the ketamine session, a preparation of racemic ketamine 0.1% was prepared using 2 ml of a 5 ml phial of injectable Panpharma ketamine 250 mg/5 ml (containing 288.4 mg of ketamine chlorhydrate, corresponding to 250 mg of base ketamine), which was added to a 100 ml bag of saline solution (Macropharma sodium chloride 0.9%), of which 2 ml had been

extracted. For the placebo session, the 100 ml bag of saline solution was used with no further preparation. The ketamine and placebo bags were indistinguishable.

Ketamine (or saline) was administered intravenously using a three-stage infusion protocol: a 10 min bolus stage (0.023 mg/kg/min), followed by a 20 min stabilization stage (0.0096 mg/kg/min) and a third open-ended stage (0.0048 mg/kg/min) until task completion (~60 min). The pump used was a programmable Volumat Agilia pump (Fresenius Kabi France SAS), programmed with the above protocol and which switched sequentially from one stage to the next automatically. This protocol was inspired from previously published research[52], and adapted after performing simulations using the pharmacokinetic parameters of a three-compartment model[53] to achieve stable plasma concentration of 150 ng/ml of ketamine during task execution. This target plasma concentration was chosen to achieve subtle cognitive alterations, while avoiding the emergence of additional symptoms that would further distract participants from the task. This low target plasma concentration is also known to maximize tolerability, with few adverse effects[54].

**Clinical measures**. Participants underwent psychiatric symptom measurements during each experimental session at three time points: (1) before infusion start ($t = 0$), (2) before the start of the cue combination task ($t = 30$ min), and (3) at the end of the cue combination task ($t \approx 90$ min), right before infusion stop. The measurements consisted of the 24-item Brief Psychiatric Rating Scale[55] (BPRS) which is designed to assess general psychiatric dimensions, and the 23-item Clinician-Administered Dissociative States Scale[56] (CADSS) which is designed to assess "dissociative" symptoms that are expected under ketamine. These scales have been extensively validated, standardized, and are frequently used.

Participants also underwent physical measurements (blood pressure, heart rate, blood oxygen saturation) before infusion start ($t = 0$) and at the end of each experimental block of the cue combination task (every 15–20 min). The level and stability of ketamine plasma concentration was controlled by two blood samples taken before the start of the cue combination task ($t = 30$ min), and at the end of the cue combination task ($t \approx 90$ min). Blood samples were immediately centrifugated for 10 min at $1700 \times g$ and 4 °C using a Multifuge 3 S centrifuge (Fisher Scientific SAS). The resulting plasma was frozen at −20 °C and sent to the pharmacology lab to be analyzed. Ketamine plasma concentration was estimated using a Waters 600 high-performance liquid chromatography system and a Waters 2996 photodiode array detector (Waters Corporation).

**Cue combination task**. The task consisted of a variant of the "weather prediction" task[8] in which participants are asked to identify the generative category (deck) of a sequence of oriented stimuli (draw of cards) among two alternatives depicted by two colors (orange and blue) which differed in terms of their generative distributions over orientation. Stimuli corresponded to oriented black bars presented in the foreground of a colored disc displaying an angular gradient between orange and blue (through gray)—the two cardinal colors being spaced by 90 degrees and varying pseudo-randomly along the circle across trials. The colored disc was presented 800 ± 80 ms prior to the onset of the first stimulus, and remained present on the screen until the end of the stimulus sequence. On each trial, a sequence of 4, 8 or 12 stimuli was drawn from a von Mises probability distribution centered either on the orientation indicated by orange or the orientation indicated by blue, with a coherence $\kappa$ updated twice during each experimental block (i.e., every 36 trials) to achieve a target accuracy of 75% using a Bayesian psychophysical titration procedure. The titration procedure relied on the noisy inference model validated in this task[8] to estimate the coherence $\kappa$ required to reach the target accuracy. The estimate for the spread of inference noise was initialized to the mean value observed in previous datasets ($\sigma = 0.5$), and updated every 36 trials using maximum likelihood estimation. This titration procedure was used to match decision accuracy to the mean probability of success (0.75) of the lotteries presented at the end of each trial.

The number of stimuli in each sequence was sampled pseudo-randomly and uniformly across trials. Each sequence was presented at an average rate of 2.5 Hz, using an inter-stimulus interval of 400 ± 40 ms. The last stimulus of each sequence was followed by a change in the fixation point ("go" signal) which probed the participant for a response, by pressing either of two buttons of a Cedrus response pad (Cedrus Corporation) with their left or right index finger—using a response mapping (e.g., left for orange, right for blue) fully counterbalanced between participants. Participants had a maximum of 2 s to provide their response. No feedback was provided about the true generative category of the sequence of stimuli, except during the training block at the beginning of each experimental session, where pairs of tones with increasing frequencies (440–880 Hz) followed correct responses, pairs of tones with decreasing frequencies (880–440 Hz) followed errors, and two low-frequency tones (220–220 Hz) followed a timeout. Visual stimuli were presented in front of a gray background using the Psychtoolbox-3 toolbox[57–59] and additional custom scripts written for MATLAB (Mathworks). The display CRT monitor had a resolution of 1024 by 768 pixels and a refresh rate of 75 Hz. Participants viewed the stimuli while seating at a distance of ~60 cm from the screen in a normally lit room.

Following a short (0.5 s) or long (4.0 s) delay after each provided response, a lottery depicted by a grayscale pie chart was presented to participants. The darker area indicated the probability of success of the lottery (0.6, 0.7, 0.8 or 0.9), which

varied unpredictably and uniformly across trials. Participants were asked to decide whether they wanted to be rewarded on the current trial based either on the accuracy of their previous response (validation) or on the outcome of the presented lottery (opt out). Participants validated their previous response by pressing again the same button, or opted out from their previous response by pressing the other button. Participants had a maximum of 2 s to provide their validation or opt-out response. The mean probability of success of the lottery (0.75) was chosen to match the titrated accuracy of participants' responses, such that participants could not rely on a "default" strategy. Instead, participants were explicitly encouraged during training to compare the probability of success of the presented lotteries with the perceived accuracy of their decisions to decide whether to validate or opt out.

Each experimental session consisted of six blocks of 72 trials each (each lasting ~10 min): a training block performed before infusion start, and five test blocks performed from $t = 30$ min after infusion start. Participants took short rest periods between each block, during which physical measurements (blood pressure, heart rate, blood oxygen saturation) were obtained. The initial training block was used to achieve approximate convergence for the Bayesian psychophysical titration procedure described above.

**Behavioral modeling of decision errors.** We used a validated model of decision-making to decompose the sources of human decision errors in this task across sensory, inference and response selection stages of processing[8]. In a given trial, after observing $n$ stimuli $\theta_1, \ldots, \theta_n$, the Bayes-optimal decision maker accumulates stimulus-wise log-likelihoods ratios $\ell_k = \log(p(\theta_k|A)/p(\theta_k|B))$ to form the log-posterior belief $\mathcal{L} \equiv p(A|\theta_{1:n}) = \sum_{k=1}^{n} \ell_k$, and chooses the category (A or B) based on the sign of $\mathcal{L}$ (A if $\mathcal{L} > 0$, B otherwise). The Bayes-optimal choice is therefore deterministically related to the stimulus sequence. Because stimulus sequences are drawn from von Mises probability distributions centered on orthogonal orientations with coherence $\kappa$, the log-likelihood ratios $\ell_k$ associated with orientation $\theta_k$ for a categorization axis $\theta^*$ can be written as:

$$\ell_k = 2\kappa \sin(2(\theta_k - \theta^*)) \quad (1)$$

Sensory errors were modeled by introducing noisy orientation percepts $\hat{\theta}_k = \theta_k + \varepsilon_k$, where $\varepsilon_k \sim \mathcal{N}(0, \sigma_{sen}^2)$ are zero-mean Gaussian variables, independent across stimuli, with sensory noise variance $\sigma_{sen}^2$. Inference errors were modeled by introducing variability at the inference stage through noisy log-likelihood ratios $\hat{\ell}_k = \ell_k + \varepsilon_k$, where $\varepsilon_k \sim \mathcal{N}(0, \sigma_{inf}^2)$ are zero-mean Gaussian variables, independent across stimuli, with inference noise variance $\sigma_{inf}^2$. Response selection errors were modeled by sampling each response $r$ from the log-posterior belief $\mathcal{L}$ through a "softmax" selection policy, $p(r = A|\theta_{1:n}) \propto \exp(\beta\mathcal{L})$, where $\beta$ is the "inverse temperature" of the policy ($\beta = 0$, random choices; $\beta = 1$, posterior sampling; $\beta \to \infty$, greedy choices). This response selection policy is indistinguishable from adding Gaussian noise to the log-posterior belief[8].

Model fitting was performed through maximum likelihood estimation using the "interior point" algorithm of the fmincon routine in MATLAB (version R2019b). Bayesian model selection (both fixed-effects and random-effects) was based on approximating the model evidence by the Bayesian Information Criterion. In all model fits, the concentration parameter $\kappa$ was used as scaling parameter by setting it to its true, generative value ($\kappa = 0.5$). When characterizing the sources of errors in each condition, we relied on a factorized, "family-wise" approach[60,61] which considers all possible combinations of the three noise sources described above. When identifying the noise source which is most likely to increase under ketamine, we compared three models in which only a single source of noise was allowed to vary between the two conditions (the other two being constant).

We further implemented a "model recovery" procedure to test the robustness of our model fitting and selection procedures. When characterizing the sources of errors in each condition, the recovery procedure consists in simulating our three candidate models of interest (model 1: sensory errors, model 2: inference errors, model 3: response selection errors), and applying our model fitting and selection procedures to obtained simulations to test whether we can "recover" accurately the simulated model. When identifying the noise source which is most likely to increase under ketamine, the three candidate models now differ in the single noise source allowed to vary between the two conditions (model 1: sensory errors; model 2: inference errors; model 3: response selection errors). For model simulations, we used as parameter values the posterior means obtained by fitting the corresponding model to each participant. This recovery procedure provides an external validation for the models being tested: their sources of errors are recoverable from behavior[19].

**Behavioral modeling of opt-out decisions.** We modeled opt-out decisions by a logistic regression model with two parametric trial-wise regressors: (1) the expected accuracy $\hat{p}_{cor}$ of the decision provided by the participant regarding the category of the stimulus sequence, and (2) the probability of success $p_{lot}$ of the presented lottery. The expected accuracy of each decision was computed by conditioning the predicted log-posterior $\mathcal{L}$ of the best-fitting noisy inference model to the decision provided by the participant—i.e., by computing the mean of a truncated normal distribution whose mean and variance is given by the statistics of the log-posterior $\mathcal{L}$ on this trial. The mean expected accuracy across trials was adjusted to the mean lottery probability (0.75), as instructed explicitly to the participants. Both regressors were entered as log-odds ratios ($\hat{\ell}_{cor} \equiv \text{logit}(\hat{p}_{cor})$ for the expected accuracy and

$\ell_{lot} \equiv \text{logit}(p_{lot})$ for the lottery probability) in the logistic regression:

$$p(\text{opt out}) = \frac{1}{1 + \exp(\beta_0 + \beta_1 \hat{\ell}_{cor} - \beta_2 \ell_{lot})} \quad (2)$$

where the three fitted parameters are: (1) the opt-out criterion $\beta_0$ (a lower criterion means a higher probability of opting out), (2) the opt-out sensitivity to the expected accuracy $\beta_1$ (a higher expected accuracy predicts a lower probability of opting out), and (3) the opt-out sensitivity to the lottery probability $\beta_2$ (a higher lottery probability predicts a higher probability of opting out).

**Simulations of premature commitments.** We performed simulations of premature commitments to assess their impact on the signatures of cognitive inference identified in behavior and EEG data. Premature commitments were modeled as a selective perturbation of the noisy inference process which best describes participants' behavior across conditions. The spread of inference noise was set to its best fitting value under placebo, and only the probability of premature commitments was varied across simulations. Premature commitments could occur after each stimulus, starting from the 4th stimulus in each sequence (the smallest sequence length used in the experiment), following a flat hazard rate $p$. The occurrence of a premature commitment after stimulus $k$ was associated with the covert selection of a category based on the sign of the log-posterior belief $\mathcal{L}_k$ at the time of commitment, and followed by the selective integration of evidence consistent with the selected category and the discarding of evidence conflicting with the selected category.

The effect of occasional premature commitments on inference noise was quantified by fitting the noisy inference model to simulated responses. Coding precision in the model was estimated by correlating the noisy representations of stimulus evidence computed by the model $|\hat{\ell}_k|$ with the true (objective) stimulus evidence $\ell_k$. This correlation was corrupted by additive "observation" noise following a zero-mean Gaussian distribution, to yield coding precision estimates of the same magnitude as the ones obtained when decoding stimulus evidence from EEG data. Belief strength in the model was quantified as the magnitude of the log-posterior belief after each stimulus $|\mathcal{L}_k|$.

**Electroencephalography.** EEG data were recorded using a 64-channel BioSemi ActiveTwo system (BioSemi, Amsterdam, The Netherlands) at a sampling frequency of 1024 Hz. The 64 scalp electrodes were positioned according to the 10/20 system. After recording, the raw EEG data were down-sampled to 512 Hz, high-pass filtered at 1 Hz using a 4th-order Butterworth filter to remove slow (sweat-induced) drifts, and referenced to the average of all electrodes. EEG data were epoched around each trial, from 0.5 s before the onset of the colored disc indicating the mean orientations of the two categories for the current trials, and until 0.5 s after the validation or opt-out response provided by the participant. Individual epochs were inspected visually to remove epochs containing large jumps, and to identify bad electrodes whose activities were reconstructed using spherical interpolation using neighboring electrodes. Remaining stereotyped artifacts (e.g., eye blinks) were then removed by decomposing the EEG data into independent sources of brain activity using Independent Component Analysis (ICA), and pruning eye blink components from the EEG data manually for each participant. These preprocessing steps were performed using the EEGLAB[62] and FieldTrip[63] toolboxes for MATLAB, and additional custom-written scripts.

**Multivariate pattern analyses of EEG data.** We applied multivariate pattern analyses to stimulus-locked EEG epochs (from 0.2 s before to 0.8 following stimulus onset) to estimate the neural patterns associated with two characteristics of each stimulus k, where k corresponds to the position of the stimulus in the current sequence: (1) its orientation $\theta_k$ described by $\cos(2\theta_k)$ and $\sin(2\theta_k)$, and (2) the strength of the evidence provided by the stimulus, described by its absolute log-likelihood ratio $|\ell_k|$. Importantly, owing to features of our experimental task (in particular the changes in category means across trials), these four stimulus characteristics showed no significant correlation with each other for a given stimulus k.

The EEG data features used for multivariate pattern analyses corresponded to the analytical representations (decompositions into real and imaginary parts) of low-pass filtered EEG signals, computed using the Hilbert transform. The low-pass frequency cutoff was optimized separately for the decoding of stimulus orientation (16 Hz) and stimulus evidence (8 Hz) across conditions. The decomposition of EEG data into real and imaginary parts allowed the decoder to use the temporal basis of each EEG signal instead of its waveform. This transform doubled the number of data features provided to the linear decoder (i.e., 128 data features for 64 electrodes).

Multivariate pattern analyses relied on an inverted linear encoding model. First, we solved the following linear encoding equation: $D_1 = W_{enc}C_1$, where $C_1$ is the design matrix on the training set ($m$ regressors $\times n_1$ training epochs), $D_1$ is the EEG data matrix at time $t$ following sample onset on the training set (128 data features $\times n_1$ training epochs), and $W_{enc}$ is the encoding weight matrix (128 data features $\times m$ regressors) to be estimated. Note that $m = 2$ for the decoding of stimulus orientation (using $\cos(2\theta_{1:n})$ and $\sin(2\theta_{1:n})$ as regressors), whereas $m = 1$ for the decoding of stimulus evidence (using $|\ell_{1:n}|$ as single regressor). This

estimation was obtained based on ordinary least squares:

$$W_{enc} = D_1 C_1^T (C_1 C_1^T)^{-1} \qquad (3)$$

where $X^T$ is the transpose of X and $X^{-1}$ is the pseudoinverse of X. We then inverted the encoding weight matrix $W_{enc}$ to obtain neural predictions $\hat{C}_2$ for the regressors on the test set ($k$ regressors $\times n_2$ test epochs) from the EEG data matrix $D_2$ at the same time $t$ following sample onset on the test set (128 data features $\times n_2$ test epochs): $\hat{C}_2 = W_{dec} D_2$, where $W_{dec}$ is the decoding weight matrix ($k$ regressors $\times$ 128 data features) obtained using:

$$W_{dec} = (W_{enc}^T W_{enc})^{-1} W_{enc}^T \qquad (4)$$

After applying this procedure for each cross-validation fold ($N = 10$ interleaved folds for all analyses), we computed the linear correlation coefficient between neural predictions $\hat{C}_2$ and ground-truth values $C_2$ of the stimulus characteristic. The coding precision metric reported in the main text corresponds to the Fisher transform of the correlation coefficient, which is approximately normally distributed, such that we could compute standard parametric statistics at the group level.

The multivariate pattern analyses described above were conducted for each participant and each condition. At the group level, we used standard parametric tests (paired $t$ tests, repeated-measures ANOVAs) to assess the statistical significance of observed differences in coding precision between conditions across tested participants. Neural coding latency was computed for each stimulus characteristic and each condition by estimating the peak of coding precision using a jackknifing (leave-one-out) procedure[64]. The type 1 error rate arising from multiple comparisons was controlled for using non-parametric cluster-level statistics computed across time points[65]. All findings reported in the main text were robust to changes in the method used for computing EEG patterns (e.g., by applying regularized ridge-regression decoding instead of the inverted encoding model described above) and to the number of cross-validation folds. When computing coding precision across a time window, we averaged neural predictions $\hat{C}_{2,t}$ across the time window $t \in [t_1, t_2]$ prior to computing the linear correlation coefficient with ground-truth values $C_2$.

**Spectral decomposition and analyses of EEG data**. The spectral power of band-limited EEG oscillations between 8 and 32 Hz was estimated using the "multi-tapering" time-frequency transform implemented in FieldTrip[63] (Slepian tapers, 8 cycles and 3 tapers per window, corresponding to a frequency smoothing of 25%), in a time window surrounding each response regarding the category of the stimulus sequence (from 2.5 s before until 0.5 s following response execution).

We then trained a linear classifier on EEG power estimated at response execution for each frequency $f$ to predict the response (left- or right-handed) provided the participant, using the cross-validation procedure described above. We could then project EEG power at frequency $f$ for each trial from the test set (not used for training the linear classifier) on the spatial component defined by the difference between left- and right-handed responses at response execution, across the whole time window of interest. This signal projection (named "response activity") was signed in direction of the provided response, such that a positive projection predicts (correctly) the response provided by the participant, and a negative projection predicts (incorrectly) the other response.

We estimated the onset of response activity rise as the last zero-crossing of the temporal derivative of response activity before response execution in each condition. We relied on a jackknifing (leave-one-out) approach[64] to obtain the group-level means and s.e.m. of these estimates and their statistical significance, computed after low-pass filtering time courses of spectral power in the alpha (10 Hz) and beta (20 Hz) bands at 2 Hz using a 6th-order Butterworth filter.

**Reporting summary**. Further information on research design is available in the Nature Research Reporting Summary linked to this article.

## Data availability
Individual anonymized behavioral and EEG datasets have not been made publicly accessible as participants did not provide explicit written consent regarding the posting of this data on public repositories. This data is available from the corresponding author upon request. The aggregated data displayed in the figures are provided as Source Data with this paper. Source data are provided with this paper.

## Code availability
The core functions and scripts used for running the experiment, analyzing the behavior and EEG data, fitting the computational model of noisy inference and confidence to behavior in the cue combination task, and simulating the premature commitment model of ketamine effects, are publicly accessible at Zenodo[66] (https://doi.org/10.5281/zenodo.5752796).

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

## Acknowledgements

This work was supported by a starting grant from the European Research Council (ERC-StG-759341) awarded to V.W., a junior researcher grant from the French National Research Agency (ANR-14-CE13-0028-01) awarded to V.W., a young investigator award from the Fyssen Foundation to V.W., a donation from the Schizo-oui association to R.G., and a starting grant from the Institut de Psychiatrie et de Neurosciences de Paris awarded to R.G. A.S. was funded by a doctoral fellowship from the Fondation pour la Recherche Médicale.

## Author contributions

P.D., R.G. and V.W. conceptualized the project. A.S., L.H.A. and V.W. performed the experiments and processed the data. V.W. conducted the data analyses. A.S. and V.W. interpreted the results and wrote the original manuscript. A.S., L.H.A., F.V., P.D., R.G. and V.W. reviewed and edited the manuscript. R.G. and V.W. acquired funding and supervised the project.

## Competing interests

F.V. has been invited to scientific meetings, consulted and/or served as speaker and received compensation by Lundbeck, Servier, Recordati, Janssen, Otsuka, and LivaNova. R.G. has received compensation as a member of the scientific advisory board of Janssen, Lundbeck, Roche, SOBI, Takeda. He has consulted and/or served as speaker for Astra Zeneca, Boehringer-Ingelheim, Pierre Fabre, Lilly, Lundbeck, MAPREG, Otsuka, Pileje, SANOFI, Servier, LVMH, and has received research support from Servier. Co-founder and stock shareholder: Regstem. All other authors declare no competing interests.
