## [Peer Review File · Nature Communications]

Premature commitment to uncertain decisions during human
NMDA receptor hypofunctionREVIEWER COMMENTS

Reviewer #1 (Remarks to the Author):

This study provides a computational and neurophysiological assessment of the impact of the NMDA-receptor blocker ketamine on behavior and neural activity in a high-level perceptual accumulation task in healthy human participants. The authors combine a state-of-the-art pharmacological protocol with sophisticated behavioral modeling and data analysis procedures. They report that ketamine increases (i) decision uncertainty (measured as tendency to "opt out" rather at the end of the trial) as well as (ii) noise in the accumulation process increases. Ketamine also reduces (iii) the precision of the encoding of stimulus evidence (decoupled by from the encoding of raw sensory input) in EEG signals and, in particular, (iv) coding precision for specifically for evidence inconsistent with the upcoming choice late in the trial. This latter effect is paralleled by another ketamine-specific one (in EEG signatures of motor preparation): (v) motor response preparation before the end of the evidence sequence. These two ketamine-specific effects are (vi) more strongly expressed on trials with higher than with lower decision confidence (gauged by the "validate" vs. "opt-out" decision). This complex pattern of results is interpreted with the help of simulations of a process model involving premature decisions at different rates and latencies from trial onsets. Importantly, there is no change in the time course of evidence weighting ("psychophysical kernel"), an observation that is somewhat underemphasized at present.

I much enjoyed reading this paper. It is very timely, original, and superbly executed. I think it can make an important contribution to the mechanistic understanding of cognitive inference and also has profound implications for the emerging understanding of the cognitive effects of clinical disorders associated with hypofunction of the NMDA-receptor, including schizophrenia. The study is very elegantly designed, the analysis is deep, the effects are rich, and the paper is overall well written. It is a beautiful demonstration of the potential of well-controlled pharmacological interventions in humans with computational modeling of behavior and the underlying neural dynamics.

While I am enthusiastic about the general approach and main findings, there are a number of major issues regarding the analyses and presentation, which I feel need addressing. In general, I think that the authors draw rather strong conclusions reaching far beyond the primary results (e.g., about the trajectory of psychosis), several of which are not sufficiently backed by the existing data.

Major:

1. Clarify the logic of the interpretation and substantiate it with further analyses.

Here is how I understand your interpretation of the main findings summarized above as formalized in your process model and at a conceptual level in Supplementary Figure 10. The *primary* effect of ketamine is to increase decision uncertainty (increase in opt-out rate). This is "compensated for" by the tendency toward premature commitment, which, in turn induces coding imprecision for conflicting samples (conflicting samples are ignored). This produces an increase in inference noise (i.e. pre-commitment is a systematic source of accumulation "noise"). The pre-commitment, in turn, increases decision confidence (through biased sampling) on a subset of trials. (Please correct me if I have misunderstood anything here.)

Now, I can follow this account and agree that it is consistent with part of the results presented. But I have two general issues with it: (i) The account assumes two opposing effects of the same cause (NMDA hypo-function): overall under-confidence and over-confidence on specific trials. (ii) The account also implies a temporal sequence, on a timescale that remains elusive, which is not supported by the current analyses. My specific issues are as follows.

a. Reconcile the "primary" increase and the "secondary" decrease (on trials with pre-commitment) in decision uncertainty induced by ketamine.

It seems that the above accounts requires that, collapsed across the experiment, the primary, ketamine-induced uncertainty increase is bigger than the secondary uncertainty-reduction through

pre-commitment. This could be because the former occurs earlier in time and/or because it is bigger in magnitude over all trials. I feel this would need to be clarified and actually tested in the data so as to corroborate the conclusion. Here some ideas for this, which are not mutually exclusive.

- The premature commitment / process model could be extended by a pre-commitment / opt-out decision, in order to test whether it can also explain the overall increase in uncertainty. I understand that you think of that uncertainty effect as the source of the tendency to pre-commitment, and so the current model only seeks to explain the remaining features of the data (inference noise and coding imbalance). But I do think a combined simulation of different levels of uncertainty (primary ketamine effect) and of different pre-commitment rates and a combined assessment of the overall opt-out rate and the effects in the current Figure 7 would provide a more complete assessment of the overall pattern of results.

- My understanding (which may be wrong, see below) is that you think of the primary and secondary effects as unfolding across trials within your experiment. If so, you can strengthen the support for this conclusion by tracking the trajectory over time: i.e., analyze the data as a function of trial blocks (built into the design or defined posthoc), to check whether under-confidence comes first (i.e., present in K+ trials from the first block), followed by a tendency to pre-commitment in later K+ blocks. The latter should predict a relative increase in overall confidence (compared to the early K+ blocks).

b. Clarify timescale of the hypothesized trajectory.

What is the relevant timescale that you assume in the above two-stage account? Is the "compensation" happening (as I understood) across trials of each drug session, or within single trials? This remained elusive for me, especially with respect to Figure S10, in which time seems to refer to within-trial time.

c. Clarify the link to the trajectory of psychosis.

Related to point 1b, you suggest a putative link to the development of psychosis, which I found interesting but had some difficulty in following it in detail: Is the idea that the above-described two-stage process of ketamine effects on cognition (across a few tens of minutes) mimics the trajectory of psychosis (across several years)? Again, how does this relate to Supplementary Figure 10, which depicts the time course within a trial?

2. Discuss absence of effect of ketamine on psychophysical kernels and reconcile this with unbalanced coding later in the trial.

There seems to be a discrepancy between the neural data indicating premature commitment and your process model on the one hand and the apparent lack of a ketamine effect on psychophysical kernels (Supp Fig 2B) on the other hand. The premature commitment later in the trial should progressively decrease sensitivity for later evidence, which I don't see in the kernel shape. Shouldn't the premature commitment / coding imbalance pull the right part of the kernel down?

It is very important to elaborate on this point in discussion, for two reasons: First, for consistency of your current account with the full pattern of results. Second, because the kernel shape has been in the focus of circuit modeling work on E/I balance / NMDA receptors (e.g. Lam et al, bioRxiv, 2017; Cavanagh et al, eLife 2020, their model predictions, not the data).

3. There are a number of conclusions which I feel are not sufficiently supported by the data and should be toned down or dropped.

a. Abstract: "These behavioral effects of ketamine were associated in electrical brain activity with distorted coding of beliefs in associative cortex followed by premature response commitment in motor cortex."

Several things here (and related places in the main text):

- "coding of beliefs" is likely to be understood as coding of "prior" or "posterior", which has not been assessed. Please replace by "distorted coding of evidence".

- Inferences about "associative cortex" are not justified by sensor-level topographies of EEG data, which can also originate, at least in part, from sub-cortical sources and/or combinations of sources in

some (higher) sensory cortical areas. The data show that the topographies of coding precision for stimulus orientation and sensory evidence are distinct, whereby the former is consistent with a source in early visual cortex. This is sufficient as a conclusion.

- Finally, "associated with" can be read as "correlated with", which has not been tested (e.g., using across-subjects correlations).

b. Ketamine effect on sensory processing.

You titrate decision difficulty (in terms of "stimulus coherence" parameter κ) so as to match choice accuracy between the ketamine and placebo conditions. I understand the rationale for measuring drug effects on uncertainty unconfounded by accuracy differences. But the procedure has implications for the reporting and interpretation of other results. For one, the effect of ketamine on titrated difficulty (stimulus coherence κ) should be reported, ideally in a main figure, so that the reader can judge the impact of the drug on overall performance. For another, any inferences about changes in sensory noise under the drug are confounded by the differences in (the variance of) sensory input between drug conditions, which is certainly going to have some effect on sensory processing due to mechanisms like adaptation. You show convincingly that the drug increases inference noise, which is an intriguing and important result. I neither see why it is necessary to further claim that there is no effect on sensory processing at all, nor do I see how this is possible conclusively with the current design.

c. Relation to E/I balance and cortical circuits.

At several places you motivate your analyses from, or interpret your results within, the framework of cortical circuit models of evidence accumulation. But your study is cast at an abstract computational level and without an assessment of circuit models, the behavior of which may easily escape intuition. Now, I think your approach is perfectly valid and informative and it is also OK to motivate it from the focus that the circuit modeling work by Wang, Murray and colleagues has placed on the NMDA receptor. But I do suggest you tone all claims throughout the paper, that your tests and/or results actually correspond to predictions from circuit models under NMDA or E/I manipulations. Most importantly, you seem to equate the "unbalanced coding" of choice-consistent vs. inconsistent evidence with an increase in E/I ratio. As motivation of this analysis you state (p.12) that "theories of NMDA receptor hypofunction in neural circuits predict a specific processing imbalance which destabilizes evidence accumulation". This prediction does not seem to be a straightforward consequence from the cited circuit models. I suggest dropping it or providing a specific motivation for it.

Please note that the primary prediction of circuit models for changes in E/I balance pertain to changes in psychophysical kernels, which are absent here (see point 2).

d. Implications for psychosis.

Psychosis is a complex disorder with a long and only partly understood developmental trajectory. I am aware that ketamine is often used as "a pharmacological model of psychosis", but this is not uncontested and at best an incomplete model. Now, you here present an exceptionally careful ketamine study of perceptual evidence accumulation in healthy participants with several intriguing results. From those results, you then derive far-reaching ideas about alterations of cognition in psychosis. I found those ideas to be very thought-provoking, and I generally think that the choice of the degree of speculation entailed in a discussion is (within limits) up to the authors of a paper. Even so, I wonder whether some of these speculations were better placed in a review or opinion piece about psychosis, rather than in a primary research article about the effects of a transient pharmacological intervention in healthy participants. This pertains in particular to the final part of the Discussion, which I found to be a stretch.

Minor:

- It would help to include discussion of the relationship of the current results to those from the most prominent prior study on ketamine effects on visual evidence accumulation in monkeys (Cavanagh et al, eLife, 2020). Another related study (on visual search) is by Shen et al., J Neurosci, 2010. I this

context, and zooming out a bit, a common feature of the current results and those from Cavanagh et al, is that subjects tend to prefer more "risky" choices under ketamine: choosing a lottery over validating one's own choice in the current study, and choosing the more variable option in Cavanagh et al. I wonder if this analogy just exists at this high level of description or may inform about some common mechanism at deeper level. I would be very curious to hear your thoughts on this.

- Figure 5A right panel: the grey box indicating the time window at which ketamine has an effect, has the same color as the shaded error bar of coding precision of conflicting stimulus evidence. Perhaps change the color of either one or the other lines, to make them more visible. Same for Fig 8A.

- Supplementary Fig. 4 and main text, p.10-11: text states that decoding of stimulus orientation was supported by spectral content up to 16 Hz and refers to Supplementary Fig 4; but this figure shows that decoding precision is high all the way up to the highest analyzed frequency (32 Hz). Please clarify. Specify what "cutoff frequency" refers to here (I assume a low-pass filter).

- Supplementary Fig. 8. seems quite important, in that it shows the impact of ketamine on ongoing EEG activity. But this is incomplete (should at least also show spectra for central/motor sites) and currently disconnected from the main paper. There is only one cursory brief statement on those results in Results on p. 15, which I failed to understand. The results from this figure should be related to previous EEG/MEG work on ketamine effects on ongoing and stimulus-induced oscillations (e.g., work by Peter Uhlhaas and colleagues).

- Figure 1c legend: Please define "coherence of the stimulus sequence" here. (This figure would be a great place to show the impact of ketamine on kappa -- see Major points above).

- Consider replacing "coding unbalance" by "coding imbalance": "Unbalance" is usually used as a verb.

Reviewer #2 (Remarks to the Author):

In this report the authors investigated how aberrant NMDA activities contribute to changes in decision processes using continuous intravenous administration of sub-anesthetic ketamine (vs. placebo). I am not an expert on ketamine or NMDA so I only offer comments on the task, modeling, EEG, and interpretations.

Here, participants completed an evidence integration task while EEG was recorded. Psychometrics and computational modeling revealed that under ketamine, participants had more decision uncertainty (premature responding and low ratings of confidence), likely due to a degradation of evidence information quality (not sensory processing or choice variability). This careful modeling was further supported by leveraging established EEG signatures of sensory vs. accumulation processes, with ketamine rather convincingly only affecting the latter signature. Additional analyses investigate how this process contributed to premature responding (with EEG motor signatures as well). Ultimately, the authors characterize the full decision process and the generalized inference to thought disorders as described by low decision thresholds and biased evidence accumulation (perhaps the development of too-strong priors?).

Given the complexity and thoroughness of the analyses, it is commendable that the authors have made the paper so understandable. I think this manuscript offers an important advancement in understanding the complex mechanisms underlying major mental health disorders. I have no major concerns with the methods, results, or interpretation, so I only offer some minor points for the author's consideration.

Minor Points

1) Since the methods are at the end, it will be difficult for readers to understand the analyses without an earlier and better understanding of the task specifics, or at least a sentence in Fig 1c describing

how the orientation and color interact in the decision integration process.

2) The authors present a very conclusive interpretation that ketamine affected evidence accumulation and not early sensory or decision variability. However, these two features had some indistinguishability during model recovery and Supplemental Fig 2b seems to indicate that there may be a mix of a linear combination of evidence accumulation and a non-linear effect of evidence estimation underlying ketamine effects. That noted, the EEG appeared to support the original interpretation; however I would urge the finer-grained investigation of ketamine effects on these early EEG weights with stimulus orientation given the group separation in parameter estimation at high degrees of stimulus orientation (Supp Fig 2b). It isn't clear if Supp Fig 6c provides EEG evidence to address this question or not.

3) For studies with analysis-weighted EEG activities, I always suggest displaying the ERPs as well. Here I suggest a supplemental figure overlaying the ERPs from the relevant topographical areas (Fig 4b) with the decoding weights. ERPs are canonical phase patterns that facilitate an interpretation across experimental designs and interpretations. By comparing classifier weightings with known ERP features, many researchers will be able to draw deeper and broader conclusions about these findings.

Signed, James F. Cavanagh

Reviewer #3 (Remarks to the Author):

Salvador et al investigates the effects of acute ketamine infusion on evidence accumulation during a line orientation judgment task using computational modeling and EEG. The primary findings are that subjects showed intact discrimination ability but increased decision certainty that was also reflected in reduced low frequency (8-16 Hz) activity over parietal and frontal regions. There was a strong correlation between behavioral and EEG readouts, which is a strength of the study.

Overall, the combination of the computational approach and EEG appears elegant, and the statistical effects appear robust. However, I have significant concerns about how the study is framed and interpreted. These could likely be addressed in a revision but would require considerable work.

First and foremost, this is not a paper on schizophrenia. The same experiment could easily have been performed using schizophrenia subjects and could have directly tested whether they had analogous deficits and, if so, whether such deficits correlated with delusions or hallucinations. Hopefully, they will one day perform the study. In the interim, however, the potential relevance to schizophrenia can only be a discussion point and a rationale for future studies directly addressing the hypothesis. It should not show up in the title and elsewhere it should be made clear that while the paper may have implications for schizophrenia, these are secondary to the main point of the study and would need to be confirmed in studies of schizophrenia patients themselves.

In contrast, the paper needs to be justified based on its contribution to basic mechanisms of decision making. As detailed in the manuscript, the concept that NMDAR are involved in slow accumulation of information through reverberating circuits was proposed by Wang and Wong almost 15 years ago but (to my knowledge) has not been tested experimentally in humans. Despite the elegant data presentation in the present manuscript, it is hard to figure out if the pattern they are seeing supports the proposals by Wang & Wong, refutes them, or is somewhere in between. At the least, there should be a much more complete introduction to the Wang & Wong hypothesis, a clear statement of how this experiment was meant to rigorously test the hypotheses, and a clear statement of the a priori hypotheses (followed by a clear statement of whether or not these were confirmed). Ideally, some attempt to be made to incorporate the approaches used by Wang & Wong is establishing the hypotheses.

The choice of task needs to be better justified both from a decision making and schizophrenia

perspective. From a decision-making perspective, a more traditional approach is to use a task (e.g. coherent dots) in which slow information extraction takes place within a trial rather than across trials. Here, the task seems designed specifically not to require sensory integration. There should be a better description of why this specific task was chosen to test the underlying hypotheses. If the authors wish to retain their discussions regarding schizophrenia, it should also be noted that line-judgement ability is relatively intact in schizophrenia although other aspects of visual sensory processing and orientation judgement are impaired (e.g. PMID: 32291128). To my read, the present findings are consistent with the previously reported findings in schizophrenia. If not, the discrepancies should be noted and would argue against strong homologies between ketamine-induced alterations and schizophrenia. In the section on schizophrenia, it should be noted that the chosen task is one where schizophrenia patients have limited deficits, as opposed to components such as motion detection, and that different sensory-level results might have been obtained with a different task. An advantage of this task, however, is that it allows testing of the cognitive aspects of evidence accumulation independent of sensory aspects, which might be seen as a positive.

A disappointing aspect of the study is that there is little attempt to link to the abundant neurophysiological literature on either decision making overall or within schizophrenia specifically. At present, the specific pattern of deficit is linked only vaguely to E-I imbalance, which is relatively uninformative since all information processing in the brain relates to interactions between glutamate (E) and GABA (I) systems. There is no discussion of why ketamine produces effects only on some aspects of E-I interaction in this task and not others since sensory processing also requires E/I balance.

The introduction cites work proposing that NMDAR hypofunction may lead to hallucinations, which may be true following chronic ketamine exposure. However, hallucinations are not typically seen following acute ketamine (including apparently in this study). This study is thus not a strong test of the hypothesis that cognitive inference alterations are involved in the pathophysiology of hallucinations through E-I modulation. However, if the authors wish to raise the issue they should specifically state that the lack of ketamine-induced hallucinations in the present study argues against direct relevance of the task to these symptoms, although the alterations may be related to other symptoms that were observed during the infusions (e.g. mood, excitement or distractibility).

There is no attempt to interrelate the spectral signature of the deficit to underlying circuit motifs such as interneuron subtypes, or to isolate specific brain regions that might be involved such as LIP or SMA. There is also no attempt to link to known neuro-oscillatory disturbances in schizophrenia which have been demonstrated in both decision making (e.g. 3753429, 27913408) and more basic sensory (e.g. 31301757) paradigms. Again, any discrepancies between present results and prior observations in schizophrenia should be noted, as they help inform about the validity of ketamine models.

The manuscript presents EEG results in terms of "coding precision," which is a somewhat unusual metric. It would be useful, at least in the supplement to show actual time-frequency histograms from the task and to explain more fully the transition from power and/or ITC measures to the coding precision metric. The detailed link between behavior and EEG, which has mostly been done to this point in monkeys, should be the selling point of the paper. There should also be a more complete description of why only 8-32 Hz activity was assessed, whereas sensory responses tend to occur in the theta frequency range, and other relevant processes (e.g. entrainment) may occur at even slower frequencies. Ketamine also has known effects on gamma power, so the reason for not including higher frequencies should also be discussed.

Some of the terminology needs to be better defined. For example, the term "cognitive inference" seems to subsume processes that might otherwise be called executive processing or working memory both of which have already been studied extensively in ketamine models. For example, in the Wisconsin Card Sorting Task, subjects must accumulate evidence over trials to determine what category of information is relevant to the decision involved. In working memory tasks, information must be maintained over time. It seems like this task mostly taps into evidence accumulation, which would seem to be only one component of cognitive inference. It would be important to know whether

the “cognitive inference” term is meant to subsume concepts such as executive processing or working memory that have already been studied in relationship to ketamine or is meant to be a different process. Rather than focusing on schizophrenia, the introduction should focus much more on what sensory and/or cognitive tasks have already been studied with ketamine and how the present study adds to the existing literature.

The term “belief” also appears to be used promiscuously. In the task subjects make decisions about specific trials, but it trivializes the term somewhat to call their decision a “belief.” From a clinical point of view, it would be more useful to use the term to reflect higher-level concepts. For example, the participant may develop a “belief” that the experimenter is trying to manipulate his/her behavior or that there is a deeper meaning to the interaction than just the simple experiment that they are being asked to perform. These beliefs may interact with the concept of “salience,” which is extensively discussed in theoretical formulations of delusions and is thought to interrelate with dopamine function. Patients with schizophrenia may also develop “beliefs” that thoughts are being inserted or subtracted from their brains. The present terminology conflates these very different uses. It could be that decision uncertainty is related to higher-level delusions, but that would need to be proven. Even in the present study, it should be possible to examine the relationship between decision uncertainty and unusual thought content (or other symptoms), which I did not see in the manuscript but might have missed. In general, since the CADSS data were collected it would be useful to know whether or not there were correlations between decision-making or EEG results and symptoms. Both positive and negative results should be reported. Positive results would strengthen some of the claims of the manuscript with respect to schizophrenia.

Overall, this is an intriguing paper in that it combines detailed computational modeling of decision making with high-density EEG and ketamine challenge and thus can directly hypothesize related to the role of NMDAR in decision making as proposed by Wang & Wong, as well as others. As such, it is likely to have a significant effect on the field. The present version addresses the issue only at the level of nonspecific E-I imbalance, which does not take advantage of the richness of the data set. In contrast, there is extensive discussion of potential relevance to delusions in schizophrenia which is highly speculative and not supported by any of the internal data of the study. It should be noted that schizophrenia patients rarely develop delusions about line orientation. There are however syndromes such as Capgras that involve delusional beliefs about faces and identities, but this syndrome is not specifically related to schizophrenia. It is reasonable to speculate about how the present results might be tied to symptoms of schizophrenia such as delusions, but only as a rationale for recommending that similar studies be performed in schizophrenia itself.

Signed:
Daniel C. Javitt
Columbia University

Reviewer #1

This study provides a computational and neurophysiological assessment of the impact of the NMDA-receptor blocker ketamine on behavior and neural activity in a high-level perceptual accumulation task in healthy human participants. The authors combine a state-of-the-art pharmacological protocol with sophisticated behavioral modeling and data analysis procedures. They report that ketamine increases (i) decision uncertainty (measured as tendency to "opt out" rather at the end of the trial) as well as (ii) noise in the accumulation process increases. Ketamine also reduces (iii) the precision of the encoding of stimulus evidence (decoupled by from the encoding of raw sensory input) in EEG signals and, in particular, (iv) coding precision for specifically for evidence inconsistent with the upcoming choice late in the trial. This latter effect is paralleled by another ketamine-specific one (in EEG signatures of motor preparation): (v) motor response preparation before the end of the evidence sequence. These two ketamine-specific effects are (vi) more strongly expressed on trials with higher than with lower decision confidence (gauged by the "validate" vs. "opt-out" decision). This complex pattern of results is interpreted with the help of simulations of a process model involving premature decisions at different rates and latencies from trial onsets. Importantly, there is no change in the time course of evidence weighting ("psychophysical kernel"), an observation that is somewhat underemphasized at present.

I much enjoyed reading this paper. It is very timely, original, and superbly executed. I think it can make an important contribution to the mechanistic understanding of cognitive inference and also has profound implications for the emerging understanding of the cognitive effects of clinical disorders associated with hypofunction of the NMDA-receptor, including schizophrenia. The study is very elegantly designed, the analysis is deep, the effects are rich, and the paper is overall well written. It is a beautiful demonstration of the potential of well-controlled pharmacological interventions in humans with computational modeling of behavior and the underlying neural dynamics.

We thank the reviewer for her/his very positive and careful assessment of our manuscript.

While I am enthusiastic about the general approach and main findings, there are a number of major issues regarding the analyses and presentation, which I feel need addressing. In general, I think that the authors draw rather strong conclusions reaching far beyond the primary results (e.g., about the trajectory of psychosis), several of which are not sufficiently backed by the existing data.

Thank you for your insightful comments. We have done our best to address all of the points you raised using additional analyses and clarifications throughout the manuscript. We believe that these extensive revisions have significantly improved the manuscript.

Major:

- 1. Clarify the logic of the interpretation and substantiate it with further analyses.*

*Here is how I understand your interpretation of the main findings summarized above as formalized in your process model and at a conceptual level in Supplementary Figure 10. The *primary* effect of ketamine is to increase decision uncertainty (increase in opt-out*

rate). This is "compensated for" by the tendency toward premature commitment, which, in turn induces coding imprecision for conflicting samples (conflicting samples are ignored). This produces an increase in inference noise (i.e. pre-commitment is a systematic source of accumulation "noise"). The pre-commitment, in turn, increases decision confidence (through biased sampling) on a subset of trials. (Please correct me if I have misunderstood anything here.)

Thank you, this is an adequate description of our interpretation of the effects of ketamine in our task. Indeed, we propose that premature commitments compensate for the elevated decision uncertainty triggered by ketamine. Note, however, that our definition of 'compensatory' is descriptive in this case: premature commitments *mechanically* increase decision confidence by ignoring conflicting stimuli occurring after a pre-commitment. These conflicting stimuli would otherwise have acted to decrease decision confidence (or even change the participant's provisional decision before the end of the trial).

Our hypothesis of premature commitments under ketamine is supported in the data by two specific effects of ketamine on how participants accumulate evidence and prepare their decisions in the task: 1. the imbalanced coding of evidence - an effect which increases gradually over the course of individual trials, and 2. the premature response preparation activity. Note that these two EEG effects do not involve participants' later choices of opting out of their initial decision. This is important, because it means that the data features that provide key support for premature commitments are independent of the data features that provide support for increased decision uncertainty.

Before moving to the additional analyses suggested by the reviewer, we want to clarify here that the first aim of our process model is to show that premature commitments can indeed explain these two effects of ketamine in quantitative terms. As rightly pointed out by the reviewer, premature commitments also explain the magnitude of the increase in inference noise observed under ketamine. As illustrated on Fig. 7, the premature commitment model explains these different effects of ketamine using a single mechanism whose variants have been described in the literature as a form of 'confirmation bias'. Note that we cite this work in the discussion (Sharot et al., 2011; Lefebvre et al., 2017; Luu and Stocker, 2018; Talluri et al., 2018). The second aim of our process model is to show that premature commitments increase decision confidence by ignoring conflicting evidence. In the model, and in the data, we could show that the two signatures of premature commitments described above (the imbalanced coding of evidence, and the premature response preparation activity) are more pronounced in trials ending with a validation of the initial decision than trials ending with opt-out.

In other words, our process model does not make predictions regarding what the reviewer calls the primary effect of ketamine on decision uncertainty. Premature commitments could be added in the same way to a model with high or low decision uncertainty. But in both cases, our process model predicts that trials where a premature commitment has occurred are associated with a lower fraction of opting out than trials without premature commitments. Because we know from participants' behavior that ketamine increases decision uncertainty (i.e., more opt-out decisions despite matched decision accuracy), premature commitments effectively provide some compensation for the elevated decision uncertainty triggered by ketamine.

We hope that these points clarify the aims and scope of our process model: it does not link directly the primary and secondary effects of ketamine described by the reviewer, but explains how the secondary effect of ketamine (premature commitments) provides effective compensation for the

primary effect of ketamine (increased decision uncertainty). We think this is important also because the literature often describes premature commitments ('jumping to conclusions') as consequences of overconfidence.

Now, I can follow this account and agree that it is consistent with part of the results presented. But I have two general issues with it: (i) The account assumes two opposing effects of the same cause (NMDA hypo-function): overall under-confidence and over-confidence on specific trials. (ii) The account also implies a temporal sequence, on a timescale that remains elusive, which is not supported by the current analyses. My specific issues are as follows.

We understand the reviewer's points. Regarding point (i), our process model does not assume two strictly opposed effects of ketamine on decision uncertainty. Indeed, our model hypothesizes that ketamine increases decision uncertainty (effect 1) and triggers premature commitments during inference (effect 2) - a second effect which, in turn, increases decision confidence on specific trials where they occur. We do not see a strong issue with the fact that ketamine produces these two effects, for two reasons. First, these two effects are not fully opposing, in the sense that premature commitments (effect 2) increase decision confidence through a different mechanism than the overall increase in decision uncertainty triggered by ketamine (effect 1). The increased decision uncertainty (effect 1) is best fitted by an overall shift (bias) in the confidence signal used to compare the subjective accuracy of the initial decision to the lottery probability (see Fig. 2 and the subsection entitled **Increased decision uncertainty under ketamine**). By contrast, premature commitments (effect 2) increase decision confidence by ignoring decision-conflicting stimuli. Second, we do observe these two effects of ketamine in the data: increased opting out (effect 1) and the combination of an imbalanced coding of evidence with a premature response preparation (effect 2). To clarify these points, which we fully agree are important, we have modified the subsection entitled **Relation between premature commitments and decision uncertainty** in the revised manuscript (p. 17):

"Similar 'jumping to conclusions' effects have already been described, but in participants or conditions associated with high decision confidence, something which stands at odds with the increased decision uncertainty observed here under ketamine (Fig. 2). To better understand the relation between premature commitments and decision uncertainty, we simulated the model for different probabilities of premature commitments and different levels of decision uncertainty (Fig. 7d). We reasoned that a reduced coding of conflicting stimuli (which should result in a reduced contribution of these stimuli to the subsequent decision) should mechanically increase the amount of accumulated evidence, and therefore decrease decision uncertainty. Simulations showed that premature commitments indeed provide an effective compensation for increased decision uncertainty: the presence of premature commitments reduces the overall probability of opting out (Fig. 7d) and, conversely, the neural signatures of premature commitments are more pronounced in trials ending with a validation of the initial decision than trials ending with opt-out (Supplementary Fig. 10ab)."

Regarding point (ii), our interpretation is indeed that premature commitments tend to occur because of the elevated decision uncertainty triggered by ketamine. However, this interpretation does not necessarily mean that premature commitments increase at a 'slow' time scale over the course of the experiment (~1 hour) - see our response to the next point. What our process model predicts is that premature commitments increase decision confidence at a 'fast' time scale over the course of a single trial (~few seconds). This is what Fig. 8 and the associated subsection entitled **Relation between premature commitments and decision uncertainty** show in the data. We

agree with the reviewer that we did not conduct extensive analyses of our process model to quantify the effects of premature commitments on decision confidence (measured in our task by the fraction of opt-out decisions at the end of each trial). We have now performed new analyses to address this point, and we have clarified that our process model makes predictions only at the 'fast' time scale - within each trial. To assess whether the two effects of ketamine are sustained over time across trials, we computed them separately for the first and second halves of the experiment - something which we now report at the end of the Results section (p. 20):

"Our proposed model predicts that premature commitments provide effective compensation for the elevated uncertainty triggered by ketamine at a 'fast' time scale within each trial, but does not make specific predictions regarding the temporal unfolding of the different effects of ketamine at a 'slow' time scale across trials in the experiment. To assess whether the different effects of ketamine are sustained across time, we computed them separately for the first and second halves of the experiment. All reported effects were sustained across time. In terms of behavior, the effect of ketamine on opt-out rate (Fig. 1d) was sustained across the two halves of the experiment (main effect of ketamine: $F_{1,17} = 18.3$, $p < 0.001$; interaction with time: $F_{1,17} = 0.4$, $p = 0.511$). Similarly, the effect of ketamine on inference noise (Fig. 3c) was sustained across time in the experiment (main effect of ketamine: $F_{1,17} = 10.5$, $p = 0.005$; interaction with time: $F_{1,17} = 0.7$, $p = 0.428$). In the EEG data, the imbalanced coding of evidence (Fig. 5b) observed under ketamine was also sustained across time (main effect of stimulus consistency on coding precision: $F_{1,17} = 24.8$, $p < 0.001$; interaction with time: $F_{1,17} < 0.1$, $p = 0.993$). Similarly, the premature response preparation activity (Fig. 6b) observed under ketamine was significant in both halves of the experiment (1st half: $t_{17} = 2.3$, $p = 0.033$; 2nd half: $t_{17} = 3.1$, $p = 0.006$). No such activity was detectable under placebo in either half (1st half: $t_{17} = 1.0$, $p = 0.343$; 2nd half: $t_{17} = 0.3$, $p = 0.779$). This effect of ketamine on response preparation was also sustained across time (main effect of ketamine: $F_{1,17} = 8.3$, $p = 0.010$; interaction with time: $F_{1,17} < 0.1$, $p = 0.921$)."

a. Reconcile the "primary" increase and the "secondary" decrease (on trials with pre-commitment) in decision uncertainty induced by ketamine.

It seems that the above accounts requires that, collapsed across the experiment, the primary, ketamine-induced uncertainty increase is bigger than the secondary uncertainty-reduction through pre-commitment. This could be because the former occurs earlier in time and/or because it is bigger in magnitude over all trials. I feel this would need to be clarified and actually tested in the data so as to corroborate the conclusion. Here some ideas for this, which are not mutually exclusive.

We thank the reviewer for this suggestion, which prompted additional analyses of our premature commitment model of ketamine effects regarding its effects on decision confidence. Let us here describe the procedures for simulating the model and characterizing its effects on decision confidence.

The value of the key parameter of the model - the 'hazard rate' (time-wise probability) of a premature commitment, or $p(\text{commitment})$ - was set from two effects shown on Fig. 7: 1. the increase in inference noise in the ketamine condition (Fig. 7b), and 2. the imbalanced coding of evidence in the ketamine condition (Fig. 7c). We found that a hazard rate of a premature commitment of 5% is sufficient to predict quantitatively the effects of ketamine on inference noise, and on the imbalanced coding of evidence. We therefore set this parameter to 5% in all simulations aimed at reproducing the pattern of effects observed in the human data - or varied it across a wide range in simulations aimed at characterizing the range of effects of premature commitments. As in the manuscript, and based on the temporal profile of coding imbalance in the data, we allowed

premature commitments to occur after a minimum of 4 stimuli (the minimal number of stimuli per trial in our task). The value of inference noise was set in all simulations to its group-level value fitted in the placebo condition. To anticipate a later point raised by the reviewer (point 2), we included in all simulations a leak in the evidence accumulation process - or inference 'leak' - whose value was set to capture the shape of the psychophysical kernels shown in Supplementary Fig. 2. We will come back to the effect of premature commitments on psychophysical kernels below. Last, the value of the opt-out criterion (used to compare the subjective accuracy of the initial decision to the lottery probability of success) was set to match the overall fraction of opt-out decisions observed in the data: 34% in the placebo condition, 43% in the ketamine condition.

To test the prediction made by the reviewer, that *"the ketamine-induced uncertainty increase is bigger than the uncertainty-reduction through pre-commitment"*, we ran three distinct sets of simulations of the premature commitment model ($N = 100$ simulations per set, each set including the trials seen by each of the 18 tested participants). In the first set, we simulated a 'baseline' model without premature commitment and set its opt-out criterion to match the fraction of opt-out decisions observed in the placebo condition. In the second set, we simulated a model with both effects of ketamine: premature commitments (hazard rate = 5%) and an opt-out criterion set to match the larger fraction of opt-out decisions observed under ketamine. Last, in the third set, we disabled premature commitments for the model used in the second set - effectively simulating only the ketamine-induced uncertainty increase without the compensatory effect of premature commitments.

As predicted by the reviewer, we found that premature commitments decrease only moderately the overall fraction of opt-out decisions (set 2 with premature commitments: 44.1%, set 3 with no premature commitment: 46.5%). In other words, premature commitments provide overall only a partial compensation for the large increase in opt-out decisions triggered by ketamine (baseline set 1: 34.1%, set 3 with increased decision uncertainty but no premature commitment: 46.5%). However, on the specific trials where a premature commitment occurs, simulations of the model with premature commitments (set 2) show a strong reduction in the fraction of opt-out decisions (trials with premature commitment: 28.6%, trials without premature commitment: 47.3%).

Together, these additional analyses show that premature commitments provide an overall partial compensation for the increased decision uncertainty triggered by ketamine, as predicted by the reviewer. We now report and discuss these additional analyses in the main text (p. 18, subsection **Relation between premature commitments and decision uncertainty**), and thank the reviewer for stimulating them:

"In simulations, occasional premature commitments decrease only moderately the overall opt-out rate (from 0.465 to 0.441 for a probability p of 5%). In other words, premature commitments provide only a partial compensation for the large increase in opt-out rate triggered by the elevated decision uncertainty observed under ketamine (from 0.341 to 0.465 in simulations without premature commitments). However, on the trials where a premature commitment has occurred, simulations show a strong reduction in opt-out rate (0.286 instead of 0.465 in simulations without premature commitment)."

- *The premature commitment / process model could be extended by a pre-commitment / opt-out decision, in order to test whether it can also explain the overall increase in uncertainty. I understand that you think of that uncertainty effect as the source of the tendency to pre-commitment, and so the current model only seeks to explain the remaining features of the data (inference noise and coding imbalance). But I do think a combined simulation of different levels of uncertainty (primary ketamine effect) and of different pre-*

commitment rates and a combined assessment of the overall opt-out rate and the effects in the current Figure 7 would provide a more complete assessment of the overall pattern of results.

We thank the reviewer for this suggestion. As reported in an earlier response, we have run a combined simulation of the two ketamine effects (the uncertainty increase, and the presence of premature commitments) on the overall opt-out rate to show that the two effects have opposing influences. We varied the sample-wise probability of a premature commitment from 0 to 20%, and the uncertainty increase from 0 to 0.2 (expressed as a shift in the opt-out criterion, as in the main text) - two ranges which produce opt-out rates that are commensurate with observed values in tested participants:

As can be seen above, the two ketamine effects have opposing and non-interacting influences on the overall opt-out rate. We agree with the reviewer that such a combined simulation shows much more explicitly how premature commitments (varying along the y-axis) effectively compensate for the increased uncertainty triggered by ketamine (varying along the x-axis). We have now added this plot to Figure 7 of the revised manuscript.

The other effects shown on Fig. 7 - the effective inference noise measured in behavior and the coding imbalance measured in the EEG data - do not depend in our process model on the level of uncertainty, but only on the probability of a premature commitment. The information provided on Fig. 7 is therefore sufficient to visualize how these two parameters of the inference process are influenced by premature commitments in our process model. We hope that the reviewer will find this additional information and revisions useful to convey that premature commitments do provide an effective compensation for the increased uncertainty triggered by ketamine.

In practice, the fact that the probability of premature commitments can be estimated in the data using features that are independent of opt-out decisions (namely, the increase in inference noise and the coding imbalance under ketamine) is a useful feature of our task and process model. In other words, while premature commitments and decision uncertainty both influence the overall opt-out rate, there are data features that afford to dissociate them.

Remember that, for a probability of premature commitments that matches the ketamine effects in terms of inference noise and decision uncertainty (a probability p of pre-commitment after each stimulus of about 5%), the compensation produced by premature commitments on the overall opt-out rate is limited. It is only for the specific trials where premature commitments occur that the probability of opting out is substantially reduced.

- My understanding (which may be wrong, see below) is that you think of the primary and secondary effects as unfolding across trials within your experiment. If so, you can

strengthen the support for this conclusion by tracking the trajectory over time: i.e., analyze the data as a function of trial blocks (built into the design or defined posthoc), to check whether under-confidence comes first (i.e., present in K+ trials from the first block), followed by a tendency to pre-commitment in later K+ blocks. The latter should predict a relative increase in overall confidence (compared to the early K+ blocks).

We actually do not believe that the two effects of ketamine unfold at a 'slow' time scale across trials within the experiment, and apologize for the lack of clarity of the original manuscript in this regard. The unfolding described in Supplementary Fig. 11 (Supplementary Fig. 10 in the original manuscript) rather corresponds to time *within a single trial* - for a single decision that would be based on more samples of evidence. As explained in response to an earlier comment, our process model does not hypothesize a specific temporal relation between the two effects of ketamine on: 1. decision uncertainty, and 2. premature commitments. In other words, the present study shows that ketamine produces both an increase in decision uncertainty and a number of premature commitments during evidence accumulation - an effect which compensates for the increased uncertainty at the expense of lower accuracy on trials when it occurs. We have clarified in the revised manuscript (p. 20) that our process model does not make specific predictions regarding the pattern of temporal unfolding of the two effects across trials within the experiment.

Note also that, from a pharmacological standpoint, tested participants performed the task during a continuous perfusion of ketamine, at a time (from 30 min after the onset of the perfusion) where ketamine plasma concentration has reached a stable level over the course of the experiment (Supplementary Fig. 1).

Nevertheless, we have performed the analysis suggested by the reviewer and computed the different effects of ketamine computed separately for the first and second half of the experiment - something which we now report at the end of the Results section (p. 20):

“Our proposed model predicts that premature commitments provide effective compensation for the elevated uncertainty triggered by ketamine at a 'fast' time scale within each trial, but does not make specific predictions regarding the temporal unfolding of the different effects of ketamine at a 'slow' time scale across trials in the experiment. To assess whether the different effects of ketamine are sustained across time, we computed them separately for the first and second halves of the experiment. All reported effects were sustained across time. In terms of behavior, the effect of ketamine on opt-out rate (Fig. 1d) was sustained across the two halves of the experiment (main effect of ketamine: $F_{1,17} = 18.3$, $p < 0.001$; interaction with time: $F_{1,17} = 0.4$, $p = 0.511$). Similarly, the effect of ketamine on inference noise (Fig. 3c) was sustained across time in the experiment (main effect of ketamine: $F_{1,17} = 10.5$, $p = 0.005$; interaction with time: $F_{1,17} = 0.7$, $p = 0.428$). In the EEG data, the imbalanced coding of evidence (Fig. 5b) observed under ketamine was also sustained across time (main effect of stimulus consistency on coding precision: $F_{1,17} = 24.8$, $p < 0.001$; interaction with time: $F_{1,17} < 0.1$, $p = 0.993$). Similarly, the premature response preparation activity (Fig. 6b) observed under ketamine was significant in both halves of the experiment (1st half: $t_{17} = 2.3$, $p = 0.033$; 2nd half: $t_{17} = 3.1$, $p = 0.006$). No such activity was detectable under placebo in either half (1st half: $t_{17} = 1.0$, $p = 0.343$; 2nd half: $t_{17} = 0.3$, $p = 0.779$). This effect of ketamine on response preparation was also sustained across time (main effect of ketamine: $F_{1,17} = 8.3$, $p = 0.010$; interaction with time: $F_{1,17} < 0.1$, $p = 0.921$).”

Together, these additional results show that the different effects of ketamine are sustained across time in the experiment. We also now make it clearer in the discussion that the compensation that

we describe is descriptive - in the sense that premature commitments effectively increase decision confidence in the ketamine condition where participants show higher decision uncertainty (p.23):

“Premature commitments – also described as ‘jumping to conclusions’ – are often seen as a signature of overconfidence. Our findings are incompatible with this view, since our participants showed elevated decision uncertainty under ketamine. However, it does not mean that premature commitments do not affect decision uncertainty. Indeed, premature commitments provide a form of effective compensation for the increased decision uncertainty over the course of a single inference process (a single trial in our task), by ignoring the evidence which conflicts with the pre-committed decision. As predicted by simulations, we show that the neural signatures of premature commitment were more pronounced in trials ending with decision validation than opt-out.”

We hope that these different results and changes to the manuscript will clarify our interpretation of the different effects of ketamine.

b. Clarify timescale of the hypothesized trajectory.

What is the relevant timescale that you assume in the above two-stage account? Is the "compensation" happening (as I understood) across trials of each drug session, or within single trials? This remained elusive for me, especially with respect to Figure S10, in which time seems to refer to within-trial time.

As described above in response to your previous points, the two effects of ketamine (on decision uncertainty and premature commitments) co-occur within single trials. We have modified the manuscript to clarify that our definition of ‘compensation’ is descriptive and does *not* imply a serial unfolding in time. In other words, premature commitments *mechanically* increase decision confidence by ignoring conflicting stimuli occurring after a pre-commitment. These conflicting stimuli would otherwise have acted to decrease decision confidence (or even change the participant’s provisional decision before the end of the trial), and it is in this sense that premature commitments provide an effective compensation for the elevated decision uncertainty triggered by ketamine. We hope that these changes have clarified our model-based interpretation of the findings, which remains agnostic regarding the temporal evolution of the two effects of ketamine.

c. Clarify the link to the trajectory of psychosis.

Related to point 1b, you suggest a putative link to the development of psychosis, which I found interesting but had some difficulty in following it in detail: Is the idea that the above-described two-stage process of ketamine effects on cognition (across a few tens of minutes) mimics the trajectory of psychosis (across several years)? Again, how does this relate to Supplementary Figure 10, which depicts the time course within a trial?

Based on the different reviews of the manuscript, we have toned down the interpretation of our findings in relation to the use of ketamine as a pharmacological model of behavioral symptoms observed at early stages of psychosis. We have rewritten the manuscript around the synaptic effect of ketamine - a NMDA-receptor antagonist - and now outline in the last paragraph of the discussion how future research should test our findings and interpretations in clinically diagnosed patients.

Related to this point, we have rewritten the Discussion section to clarify how our findings speak against existing descriptions of ‘jumping to conclusions’ as behavioral consequences of overconfidence - both in patients suffering from schizophrenia, but also in the general population. Here, during low-dose infusions of ketamine, we find that participants show both elevated decision

uncertainty and occasional premature commitments (a form of ‘jumping to conclusions’). Interestingly, because premature commitments provide effective compensation for the elevated decision uncertainty, they generate overconfidence for protracted decisions based on several pieces of evidence. We therefore note that premature commitments may be the cognitive mechanism that drives overconfidence rather than a behavioral consequence of overconfidence. In the original manuscript, we stressed that the elevated decision uncertainty reported by participants under ketamine is consistent with reports made by patients at prodromal (early) stages of psychosis, but not at later stages of the disease. Because the link between the symptoms observed under NMDA receptor hypofunction in our task and the symptoms reported by patients at early stages of psychosis is only tentative - although the subject of influential theories in the field, we have toned down the discussion in this regard and now state that future research should examine the interplay between decision uncertainty and ‘jumping to conclusions’ symptoms at early stages of psychosis, something that has not been done because these early stages are only transient and thus difficult to study in clinical studies.

We have kept Supplementary Fig. 10 (Supplementary Fig. 11 in the revised manuscript) to show how premature commitments occurring in our process model can drive overconfidence - a relation which runs opposite to standard accounts of ‘jumping to conclusions’ as a behavioral consequence (not a cause) of overconfidence. But we now make these observations in the discussion without discussing them as a possible account of psychogenesis in patients suffering from schizophrenia (pp. 23-24):

“This effective compensation predicts a progressive transition from underconfidence when presented with few pieces of evidence over seconds – as it is the case in our laboratory-based task – to overconfidence when presented with many pieces of evidence over longer periods of time – as it is typically the case outside the laboratory (Supplementary Fig. 11). Because premature commitments increase the amount of accumulated evidence by filtering out conflicting evidence (Supplementary Fig. 11a), they lead progressively to overconfidence and decreased confidence sensitivity to objective decision accuracy (Supplementary Fig. 11b).”

We hope that these several changes to the discussion have clarified our interpretation of our findings.

2. Discuss absence of effect of ketamine on psychophysical kernels and reconcile this with unbalanced coding later in the trial.

There seems to be a discrepancy between the neural data indicating premature commitment and your process model on the one hand and the apparent lack of a ketamine effect on psychophysical kernels (Supp Fig 2B) on the other hand. The premature commitment later in the trial should progressively decrease sensitivity for later evidence, which I don't see in the kernel shape. Shouldn't the premature commitment / coding imbalance pull the right part of the kernel down?

It is very important to elaborate on this point in discussion, for two reasons: First, for consistency of your current account with the full pattern of results. Second, because the kernel shape has been in the focus of circuit modeling work on E/I balance / NMDA receptors (e.g. Lam et al, bioRxiv, 2017; Cavanagh et al, eLife 2020, their model predictions, not the data).

We thank the reviewer for this important comment. The reviewer is entirely correct that the presence of premature commitments mechanically decreases sensitivity to evidence presented after the pre-commitment has occurred - because the decision-maker is effectively ignoring

evidence samples that are inconsistent with his/her decision. The reviewer is also correct that the psychophysical kernels measured under ketamine do not show such a decreased sensitivity to later evidence (Supplementary Fig. 2b). However, these two observations do not necessarily imply a discrepancy between the coding imbalance visible in the EEG data (Fig. 5c) and the psychophysical kernels measured from behavior (Supplementary Fig. 2b). This is because the coding imbalance measured in the EEG data reflects the processing of evidence upstream from the accumulation of evidence, whereas the psychophysical kernel measured from behavior reflects the decision variable at the end of the stimulus sequence, downstream from the accumulation of evidence.

The distinct positions of these two measures with respect to evidence accumulation are important, because evidence accumulation is known to be ‘leaky’ in our task (Supplementary Fig. 2b) and earlier versions of it (Drugowitsch, Wyart et al., 2016, *Neuron*). Premature commitments and leaky evidence accumulation correspond to distinct properties of inference which are not mutually exclusive. In other words, under ketamine, evidence accumulation can be subject both to premature commitments and a leak. The combined influences of these two effects can produce the psychophysical kernels observed under ketamine on Supplementary Fig. 2b - i.e., a larger sensitivity to later evidence due to the leak despite the presence of premature commitments in a fraction of all trials. To show this in practice, we have simulated decisions from our process model where the evidence accumulation process is subject both to premature commitments (with $p(\text{commitment}) = 5\%$ as in earlier simulations) and a leak λ set to match the psychophysical kernel measured under ketamine ($\lambda = 0.150$):

In the simulations shown above, the gray curve corresponds to simulations of the process model without premature commitment and a leak λ set to match the psychophysical kernel measured under placebo ($\lambda = 0.054$). Importantly, the inference noise (σ_{inf}) is set to the same value ($\sigma_{\text{inf}} = 0.471$) in both simulations. As can be seen, the combined presence of premature commitments and an increased leak under ketamine readily explains the difference between the psychophysical kernels measured from participants’ behavior. In terms of model fits, this combination of premature commitments and leak can also explain the marginal increase in inference leak λ measured under ketamine (placebo: 0.054 +/- 0.014; ketamine; 0.082 +/- 0.014; paired t-test, $t(17) = 2.1$, $p = 0.051$).

These analyses, prompted by the reviewer’s comments, are important because they suggest that ketamine may increase the evidence accumulation leak more than what is visible in the psychophysical kernels, because this increased leak co-occurs with premature commitments which

pull psychophysical kernels in the opposite direction. We now discuss these findings in the new paragraph of the Discussion section (p. 23):

“In simulations, premature commitments alter the shape of psychophysical kernels by decreasing the weights of stimuli presented late in each sequence – i.e., after premature commitments have occurred (Supplementary Fig. 10c). This predicted effect stands in apparent contradiction with participants’ psychophysical kernels under ketamine, which did not show such an effect (Supplementary Fig. 2b). However, the shape of psychophysical kernels does not only depend on premature commitments, but also on the ability to accumulate evidence over time without leak (a measure of working memory in our task). Therefore, the similar shapes of participants’ psychophysical kernels under ketamine and placebo – despite the presence of premature commitments under ketamine – suggest that ketamine may impair working memory (as described in previous work) and therefore increase the leak in evidence accumulation (Supplementary Fig. 10d). Future research should further examine the relation between these two cognitive effects of ketamine.”

As indicated above, remember that premature commitments can be dissociated from changes in evidence accumulation leak by the coding imbalance and the premature response preparation activity visible in the EEG data. We have conducted extensive simulations to characterize the effects of premature commitments and evidence accumulation leak on the effective inference noise and leak measured from our validated inference model (Drugowitsch, Wyart et al., 2016, *Neuron*). For this purpose, we have independently varied the probability of a premature commitment and the strength of the evidence accumulation leak, and measured their effect on the effective inference noise and leak values fitted using our behavioral model. We have added the results of these simulations to Supplementary Fig. 10 (subpanels c and d) of the revised manuscript:

As can be seen on the left panel, premature commitments and evidence accumulation leak have opposing effects on the effective inference leak measured from behavior: as suggested by the reviewer, premature commitments decrease the effective inference leak. By contrast, as can be seen on the right, premature commitments and leak have synergistic effects on the effective inference noise measured from behavior. In the absence of premature commitments, a leaky in the evidence accumulation process has no effect on the measured inference noise. And in the absence of leak, premature commitments have only little effects on the measured inference noise. But in the presence of a leak, premature commitments have a significant effect on the measured inference noise.

Assuming that the main effect of ketamine is to trigger premature commitments based on its effects on the EEG data (imbalanced coding of evidence and premature response preparation), one can estimate the probability of a premature commitment $p(\text{commitment})$ (y-axis) and the underlying evidence accumulation leak (x-axis) based on the effective values of inference noise and inference leak measured from behavior. This analysis indicates that ketamine is associated with a hazard rate of pre-commitment of about 5%, and an increased evidence accumulation leak (0.054 under placebo, about 0.15 under ketamine). Now, for a hazard rate of pre-commitment of 5%, we can also display psychophysical kernels for different strengths of leak, from no leak (0) to strong leak (0.20). In these simulations, we display psychophysical kernels only for sequences of 12 stimuli:

As can be seen on the left, even moderate levels of evidence accumulation leak are sufficient to mask the effects of premature commitments when they are sparse in the data ($p(\text{commitment}) \sim 5\%$). By contrast, as can be seen on the right, probabilities of premature commitments larger than the ones triggered by ketamine ($p(\text{commitment}) \sim 5\%$) are necessary to flip psychophysical kernels from moderate recency (in gray) to primacy (in red).

We thank the reviewer for stimulating these additional analyses which resolve an apparent discrepancy between the presence of premature commitments and the overall recency effect apparent in psychophysical kernels under ketamine.

3. There are a number of conclusions which I feel are not sufficiently supported by the data and should be toned down or dropped.

a. Abstract: "These behavioral effects of ketamine were associated in electrical brain activity with distorted coding of beliefs in associative cortex followed by premature response commitment in motor cortex."

Several things here (and related places in the main text):

- "coding of beliefs" is likely to be understood as coding of "prior" or "posterior", which has not been assessed. Please replace by "distorted coding of evidence".

We agree, and have changed "coding of beliefs" with "coding of evidence".

- Inferences about "associative cortex" are not justified by sensor-level topographies of EEG data, which can also originate, at least in part, from sub-cortical sources and/or combinations of sources in some (higher) sensory cortical areas. The data show that the topographies of coding precision for stimulus orientation and sensory evidence are distinct,

whereby the former is consistent with a source in early visual cortex. This is sufficient as a conclusion.

We agree, and have changed the abstract and main text accordingly.

- Finally, "associated with" can be read as "correlated with", which has not been tested (e.g., using across-subjects correlations).

We agree with this comment. We now say in the abstract: **"At the behavioral level, ketamine triggered inference errors and elevated decision uncertainty. At the neural level, ketamine was associated with imbalanced coding of evidence and premature response preparation in electroencephalographic (EEG) activity."** Note also that our experimental design was not powered to focus on between-subject correlations: we used a within-subject design such that each participant was tested under placebo and under ketamine, and that we could rely in repeated-measures statistical tests to account for between-subject variability when measuring the effects of ketamine on behavior and brain activity.

b. Ketamine effect on sensory processing.

You titrate decision difficulty (in terms of "stimulus coherence" parameter κ) so as to match choice accuracy between the ketamine and placebo conditions. I understand the rationale for measuring drug effects on uncertainty unconfounded by accuracy differences. But the procedure has implications for the reporting and interpretation of other results. For one, the effect of ketamine on titrated difficulty (stimulus coherence κ) should be reported, ideally in a main figure, so that the reader can judge the impact of the drug on overall performance. For another, any inferences about changes in sensory noise under the drug are confounded by the differences in (the variance of) sensory input between drug conditions, which is certainly going to have some effect on sensory processing due to mechanisms like adaptation. You show convincingly that the drug increases inference noise, which is an intriguing and important result. I neither see why it is necessary to further claim that there is no effect on sensory processing at all, nor do I see how this is possible conclusively with the current design.

We understand the reviewer's point. As the reviewer notes, we have made the a priori decision to titrate decision difficulty - in terms of the between-stimulus coherence κ of each stimulus sequence - to match decision accuracy between the two conditions. We now report the titrated value of κ over time in the experiment for each condition (ketamine and placebo) in a new subpanel of Figure 1 (Fig. 1e) to show that ketamine did not have a major effect on overall performance. Indeed, the titrated values of κ did not differ significantly between conditions (main effect of ketamine: $F_{1,17} = 0.8$, $p = 0.451$). This means that the variance of sensory input did not differ significantly between drug conditions, something that could not be assessed in the previous version of the manuscript. Note that the titration procedure, which we applied twice per experimental block to update the overall coherence κ of the stimulus sequence, does not provide a fine-grained measure of participants' accuracy in each condition. We used it to match the overall accuracy of decisions with the mean probability of success of the lottery (fixed to 0.75 across all participants and conditions). The new subpanel Fig. 1e also shows that the titration procedure was successful in that, excluding the very first titration step (initialized based on a previous study - Drugowitsch, Wyart et al., 2016, *Neuron*), decision accuracy did not vary significantly over time in the experiment (main effect of time: $F_{8,136} = 1.4$, $p = 0.183$).

Nevertheless, we agree with the reviewer that the absence of measurable sensory deficit in our task (with specific parameters described in the previous paragraph, and in the revised manuscript) does not imply that ketamine (and NMDA receptor hypofunction) does not produce any sensory deficit in other sensory conditions. We have now clarified this point explicitly in a new paragraph of the discussion. We stress in this new paragraph (p. 22) that the absence of sensory deficit in our task (in terms of sensory noise measured in behavior or orientation coding in the EEG data) provides support that ketamine has a specific effect on cognitive inference, rather than a general distracting (inattention) effect which could have confounded our findings. In other words, we now use the absence of measurable sensory deficit in our task as a control for the fact that ketamine did not produce a general distracting effect from the cognitive task. We believe that this is important, given that larger doses of ketamine would likely have produced such effects - this is something that is observed when using intramuscular injections of ketamine (e.g., in animals).

We also now mention in the discussion that ketamine did not alter in any way the classical adaptation-like effects on sensory processing - which were identical in the two conditions. This is what is illustrated in Supplementary Fig. 8ab, where we show a strong and sustained 'surprise' effect on the neural coding of stimulus orientation from 160 to 480 ms following stimulus onset in both conditions. The facts that ketamine did not significantly increase the variance of sensory input (due to our titration procedure) and did not affect well-known adaptation-like effects on sensory processing provide further evidence that ketamine has a selective effect on cognitive inference in our task rather than an unspecific distracting effect.

c. Relation to E/I balance and cortical circuits.

At several places you motivate your analyses from, or interpret your results within, the framework of cortical circuit models of evidence accumulation. But your study is cast at an abstract computational level and without an assessment of circuit models, the behavior of which may easily escape intuition. Now, I think your approach is perfectly valid and informative and it is also OK to motivate it from the focus that the circuit modeling work by Wang, Murray and colleagues has placed on the NMDA receptor. But I do suggest you tone all claims throughout the paper, that your tests and/or results actually correspond to predictions from circuit models under NMDA or E/I manipulations. Most importantly, you seem to equate the "unbalanced coding" of choice-consistent vs. inconsistent evidence with an increase in E/I ratio. As motivation of this analysis you state (p.12) that "theories of NMDA receptor hypofunction in neural circuits predict a specific processing imbalance which destabilizes evidence accumulation". This prediction does not seem to be a straightforward consequence from the cited circuit models. I suggest dropping it or providing a specific motivation for it. Please note that the primary prediction of circuit models for changes in E/I balance pertain to changes in psychophysical kernels, which are absent here (see point 2).

We understand the reviewer's comment - which relates to a comment made by Reviewer #3. We agree with the reviewer's description that our process model is cast at an abstract computational level (the 'algorithmic' level in Marr's three-level decomposition) rather than at an implementational level - which is where the E/I model by Wong and Wang is described. Therefore, we agree that it is much more adequate to describe and discuss our findings at the algorithmic level where our process model operates and our study is grounded. Such an algorithmic description avoids tying our findings to a specific implementation of evidence accumulation, and therefore makes our discussion more general. We thank the reviewer for prompting these changes throughout the revised manuscript.

We believe that it is important to note that alteration of NMDA receptors leads to earlier and more noisy decisions in circuit models and in our participants. This similarity, however, does not provide a formal test of these circuit models, due to several important differences between the work conducted using circuit models and our work. We now mention the most important differences in the Discussion section of the revised manuscript (and in the paragraphs below).

First, the work using the circuit model by Wong and Wang is based on an evidence accumulation task which differs quite substantially from our task: a random-dot motion task where the sensory evidence is delivered continuously and difficult to quantify at each point in time. Our cue combination task delivers sensory evidence in a discrete fashion, and each piece of evidence can be quantified directly and decoded from EEG signals. Also, the sensory signal is weak and uncertain in the random-dot motion task, whereas it is the relation between strong sensory signals and the category of the sequence that is uncertain in our task. In other words, the source of uncertainty is located at the sensory level in random-dot motion tasks, whereas it is located at the inference (categorization) level in our task - validated in healthy participants in our earlier work (Drugowitsch, Wyart et al., 2016, *Neuron*). These important differences between task parameters make it difficult to formally compare the predictions of the circuit model by Wong and Wang to our findings - as noted by the reviewer.

Second, the work using the circuit model by Wong and Wang has studied the effects of strong inactivations of NMDA receptors - which were only partial and limited (on our purpose, see below) in our pharmacological study using low-dose ketamine infusions. The plasma concentration levels achieved using our protocol (100-150 ng per mL) have been reported to be sufficient to induce cognitive perturbations, but low enough to avoid stronger symptoms that would have likely distracted participants from the cognitive task (see our response to the previous point). Our finding that ketamine impairs specifically the coding of evidence supports the idea that ketamine did not distract participants from the cognitive task, something that would likely have happened with higher doses of ketamine in naive participants.

Third, in the tested circuit models which often only describe a single brain area, there is formally no difference between committing to a decision and executing a motor response. In our data, we show that ketamine triggers premature commitments to uncertain decisions without early motor responses - i.e., 'covert' commitments that were not allowed in the single-circuit model by Wong and Wang. Overall, we have extensively modified the Introduction and Discussion sections to clarify how our work relates to earlier accounts of NMDA receptor inactivation on evidence accumulation and decision-making under uncertainty.

d. Implications for psychosis.

Psychosis is a complex disorder with a long and only partly understood developmental trajectory. I am aware that ketamine is often used as "a pharmacological model of psychosis", but this is not uncontested and at best an incomplete model. Now, you here present an exceptionally careful ketamine study of perceptual evidence accumulation in healthy participants with several intriguing results. From those results, you then derive far-reaching ideas about alterations of cognition in psychosis. I found those ideas to be very thought-provoking, and I generally think that the choice of the degree of speculation entailed in a discussion is (within limits) up to the authors of a paper. Even so, I wonder whether some of these speculations were better placed in a review or opinion piece about psychosis, rather than in a primary research article about the effects of a transient pharmacological intervention in healthy participants. This pertains in particular to the final part of the Discussion, which I found to be a stretch.

We understand this point, which was also raised by Reviewer #3. We agree with the reviewer that our manuscript is not a paper on schizophrenia, but a characterization of the perturbations resulting from NMDA receptor hypofunction using low-dose ketamine infusions. This synaptic alteration is one of the prominent hypotheses for the pathophysiology of the early stages of schizophrenia, but the reviewer is fully correct that this link only affords us to draw tentative connections to behavioral symptoms observed in schizophrenia in the discussion. We have thus removed the connection of ketamine infusions to early-stage psychosis in the title. We have also removed this connection from the abstract and only provide a tentative connection (shorter than in the original version of the manuscript) in the last paragraph of the Discussion section. We are now much more explicit that these tentative connections should be substantiated by studies of patients suffering from schizophrenia. We hope that the reviewer will find the revised manuscript much tighter in its discussion of our findings.

Minor:

- It would help to include discussion of the relationship of the current results to those from the most prominent prior study on ketamine effects on visual evidence accumulation in monkeys (Cavanagh et al, eLife, 2020). Another related study (on visual search) is by Shen et al., J Neurosci, 2010. In this context, and zooming out a bit, a common feature of the current results and those from Cavanagh et al, is that subjects tend to prefer more "risky" choices under ketamine: choosing a lottery over validating one's own choice in the current study, and choosing the more variable option in Cavanagh et al. I wonder if this analogy just exists at this high level of description or may inform about some common mechanism at deeper level. I would be very curious to hear your thoughts on this.

We thank the reviewer for drawing connections between our work and recent work using ketamine in another visual evidence accumulation task in non-human primates (in particular the study by Cavanagh et al.). Note, however, that the choice used in our task to measure decision uncertainty - i.e., the comparison between the accumulated evidence in favor of the initial decision and the probability of success of the presented lottery - does not measure the attitude of tested participants toward risk (as defined in behavioral economics). Indeed, the average probability of success of the lottery (0.75) was chosen to match the average accuracy of the initial decision (0.75, achieved through the titration procedure).

Nevertheless, the larger opt-out rate observed under ketamine can indeed be interpreted in two ways: 1. a lower subjective accuracy of the initial decision (elevated decision uncertainty), or 2. a higher subjective probability of success of the presented lottery. The fact that the ketamine increase in opt-out rate correlates - across tested participants - with the ketamine increase in inference noise (see Supplementary Fig. 2c, bottom panel) suggests that ketamine increases decision uncertainty rather than decreases the uncertainty associated with the presented lottery. We now clarify this point briefly in the main text at the end of the subsection entitled **Impaired cognitive inference under ketamine**: **"This covariation suggests that the increase in opt-out rate triggered by ketamine does not reflect a task-unspecific effect of ketamine nor a biased subjective probability of success of the lottery, but rather a selective effect of the drug on the subjective accuracy of cognitive inference."**

- Figure 5A right panel: the grey box indicating the time window at which ketamine has an effect, has the same color as the shaded error bar of coding precision of conflicting stimulus

evidence. Perhaps change the color of either one or the other lines, to make them more visible. Same for Fig 8A.

Thank you, this has been corrected by making the shaded box more transparent.

- *Supplementary Fig. 4 and main text, p.10-11: text states that decoding of stimulus orientation was supported by spectral content up to 16 Hz and refers to Supplementary Fig 4; but this figure shows that decoding precision is high all the way up to the highest analyzed frequency (32 Hz). Please clarify. Specify what "cutoff frequency" refers to here (I assume a low-pass filter).*

As the reviewer has assumed, we have used a low-pass filter to determine which frequency bands contribute to the neural coding of stimulus orientation and stimulus evidence. In practice, we have varied the cutoff frequency, from 4 Hz up to 32 Hz, to measure the cutoff frequency for which the neural coding of each stimulus characteristic is maximal - see panel b of Supplementary Fig. 4 to see the neural coding precision of each variable at its peak. We found that the neural coding of stimulus orientation peaks for a frequency cutoff of approximately 16 Hz (the orange curves in the subpanel), whereas the neural coding of stimulus evidence peaks for a frequency cutoff of approximately 8 Hz (the blue curves in the subpanel). These results (more visible on the curves shown on Supplementary Fig. 4b than the time-frequency diagrams shown on Supplementary Fig. 4a) indicate that the neural coding of stimulus orientation relies on spectral content up to 16 Hz (above which coding precision starts to decrease), whereas the neural coding of stimulus evidence relies on spectral content up to 8 Hz (above which coding precision starts to decrease). We have now clarified this reasoning in the results (p. 11):

"This neural code overlapped only slightly across successive stimuli (Supplementary Fig. 3c), and was supported by spectral content up to 16 Hz (measured as the frequency cutoff above which coding precision starts to decrease; see Supplementary Fig. 4)."

- *Supplementary Fig. 8. seems quite important, in that it shows the impact of ketamine on ongoing EEG activity. But this is incomplete (should at least also show spectra for central/motor sites) and currently disconnected from the main paper. There is only one cursory brief statement on those results in Results on p. 15, which I failed to understand. The results from this figure should be related to previous EEG/MEG work on ketamine effects on ongoing and stimulus-induced oscillations (e.g., work by Peter Uhlhaas and colleagues).*

We have re-built Supplementary Fig. 8 (now Supplementary Fig. 9 in the revised manuscript) to show what is suggested by the reviewer, and now describe the effects of ketamine on power spectra in the Results section.

The legend of the original Supplementary Fig. 8 was misleading, in that we displayed power spectra and time-frequency diagrams for a broad cluster of occipital, parietal and central channels where alpha power peaks in the inter-trial interval (baseline). The new subpanel (a) of the new Supplementary Fig. 9 shown below now shows the effects of ketamine on power spectra and time-frequency diagrams at occipital channels, whereas the new subpanel (b) shows the effects of ketamine on power spectra and time-frequency diagrams at central channels:

The dampened suppression of low-frequency power in the alpha and beta bands during task execution (here, from the presentation of the stimulus sequence until the response) is indeed highly consistent with the results reported by Grent-'t-Jong et al. (2018, *Brain*) under ketamine, but also in patients suffering from schizophrenia. Our findings add to this existing literature by showing that the dampened dynamics of alpha- and beta-band power under ketamine during task execution are likely due to weaker baseline power in these two frequency bands (measured during the inter-trial interval). We also show in subpanels (c) and (d) that, while ketamine does not affect the decoding of response laterality (left vs. right hand) in the alpha band (Supplementary Fig. 8c), ketamine strongly impairs the decoding of the same behavioral variable in the beta band. This is interesting, because alpha- and beta-band power dynamics typically show the same effects under placebo.

We thank the reviewer for prompting these revisions which position our study with respect to existing work. We originally decided not to insist on these effects that are relatively tangential to our findings, but we agree that they are nevertheless relevant in light of the existing literature on

the effects of ketamine (and schizophrenia) on EEG activity. We now report them briefly in the results (pp. 14-15):

“In agreement with previous work [Grent-'t-Jong et al., 2018, Brain], we found that ketamine decreased baseline power in the alpha (10 Hz) and beta (20 Hz) and dampened the strong power suppression in these frequency bands during visual stimulation (Supplementary Fig. 9ab). Despite these broad effects of ketamine, the additional power suppression triggered by response execution in the alpha band (10 Hz) at bilateral central channels (Fig. 6a) did not differ between conditions. We could thus use this motor signal to predict the provided response (left- vs. right-handed) in the last few seconds preceding its execution – even before the presentation of the ‘go’ signal which probed participants for a response.”

| - Figure 1c legend: Please define “coherence of the stimulus sequence” here. (This figure would be a great place to show the impact of ketamine on kappa -- see Major points above).

Thank you, we now define “coherence of the stimulus sequence” as “the concentration of the distributions of orientations associated with each category”. We also show the information about the titration procedure in a new subpanel of Figure 1 (Fig. 1e).

| - Consider replacing “coding unbalance” by “coding imbalance”: “Unbalance” is usually used as a verb.

Thank you, this has been done throughout the revised manuscript.

Reviewer #2

In this report the authors investigated how aberrant NMDA activities contribute to changes in decision processes using continuous intravenous administration of sub-anesthetic ketamine (vs. placebo). I am not an expert on ketamine or NMDA so I only offer comments on the task, modeling, EEG, and interpretations.

Here, participants completed an evidence integration task while EEG was recorded. Psychometrics and computational modeling revealed that under ketamine, participants had more decision uncertainty (premature responding and low ratings of confidence), likely due to a degradation of evidence information quality (not sensory processing or choice variability). This careful modeling was further supported by leveraging established EEG signatures of sensory vs. accumulation processes, with ketamine rather convincingly only affecting the latter signature. Additional analyses investigate how this process contributed to premature responding (with EEG motor signatures as well). Ultimately, the authors characterize the full decision process and the generalized inference to thought disorders as described by low decision thresholds and biased evidence accumulation (perhaps the development of too-strong priors?).

Given the complexity and thoroughness of the analyses, it is commendable that the authors have made the paper so understandable. I think this manuscript offers an important advancement in understanding the complex mechanisms underlying major mental health disorders. I have no major concerns with the methods, results, or interpretation, so I only offer some minor points for the author's consideration.

We thank the reviewer for his very positive assessment of our work. Given the comments made by the other two reviewers regarding the use of ketamine as a pharmacological model of symptoms observed at early stages of psychosis, we have refocused the manuscript on the synaptic effect of ketamine on NMDA receptors. As a consequence, the title, abstract, introduction and discussion sections have been revised to account for this change of focus. We now state at the end of the Discussion section that future research should be carried out in patients suffering from schizophrenia - at prodromal stages of the disease - to test whether the specific pattern of cognitive alterations observed under ketamine is also visible in patients.

Minor Points

1) Since the methods are at the end, it will be difficult for readers to understand the analyses without an earlier and better understanding of the task specifics, or at least a sentence in Fig 1c describing how the orientation and color interact in the decision integration process.

Thank you for this comment, we have added a sentence in the Results section to explain how stimulus sequences are generated with respect to the two colors.

2) The authors present a very conclusive interpretation that ketamine affected evidence accumulation and not early sensory or decision variability. However, these two features had some indistinguishability during model recovery and Supplemental Fig 2b seems to indicate that there may be a mix of a linear combination of evidence accumulation and a non-linear effect of evidence estimation underlying ketamine effects. That noted, the EEG appeared to

support the original interpretation; however I would urge the finer-grained investigation of ketamine effects on these early EEG weights with stimulus orientation given the group separation in parameter estimation at high degrees of stimulus orientation (Supp Fig 2b). It isn't clear if Supp Fig 6c provides EEG evidence to address this question or not.

The non-linear evidence estimation profile displayed on the left subpanel of Supplementary Fig. 2b is captured by the computational model: this is because the normative evidence estimation profile is expected to be sinusoidal, not linear, for category-specific distributions of orientations that follow von-Mises distributions. Therefore, our computational model does assume this non-linearity and provides a good account of the human evidence estimation profiles (in the left subpanel of Supplementary Fig. 2b, dots correspond to the human data, whereas lines correspond to predictions of the best-fitting computational model) - in particular, both the human and the simulated profiles show the same non-linearity. We have now added the definition of the stimulus evidence (the log-likelihood ratio between the two categories A and B associated with each stimulus) to the Methods section of the revised manuscript.

Nevertheless, we have also performed a finer-grained investigation using an additional free parameter in our computational model. We added an additional parameter η which controls how much participants exaggerate this non-linearity in a way that results in a pure counting-like behavior for large η values (the optimal non-linearity corresponding to $\eta = 0$). Counting-like behavior would mean that participants would simply sum the number of stimuli consistent with each category and decide based on the difference between the two sums. Fitting this parameter to the human using a fixed value across conditions showed that participants tend to exaggerate the non-linearity of the evidence estimation profile ($\eta = 0.294 \pm 0.093$, $t(17) = 3.2$, $p = 0.006$). However, fitting this parameter to different values for the ketamine and placebo conditions degraded the quality of the fits (exceedance $p = 0.059$) and did not reveal any significant difference (ketamine: 0.216 ± 0.108 ; placebo: 0.402 ± 0.107 ; $t(17) = 1.8$, $p = 0.091$).

Because this analysis does not reveal any significant effect, we have chosen not to describe it in the revised manuscript which is already very dense in terms of findings.

3) For studies with analysis-weighted EEG activities, I always suggest displaying the ERPs as well. Here I suggest a supplemental figure overlaying the ERPs from the relevant topographical areas (Fig 4b) with the decoding weights. ERPs are canonical phase patterns that facilitate an interpretation across experimental designs and interpretations. By comparing classifier weightings with known ERP features, many researchers will be able to draw deeper and broader conclusions about these findings.

Signed, James F. Cavanagh

We understand that event-related potentials (ERPs) are the most standard EEG measure - which makes findings more readily comparable across studies. However, owing to the rapid sequential presentation of stimuli in our task (every 400 ms with jitter), ERPs cannot be interpreted without confounds and the neural coding approach we have adopted is the most suited to identify which EEG signals reflect the processing of individual stimulus characteristics (e.g., the evidence provided by stimulus at position k and not position $k+1$ or $k-1$). Note that we have developed this approach across several past studies (Wyart et al., 2012, *Neuron*; Wyart et al., 2015, *J. Neurosci.*; King and Wyart, 2021, *J. Neurosci.*) using streams of visual stimuli.

Nevertheless, we agree with the reviewer that readers may be interested in knowing what ERPs look like in our task. We therefore have added a new Supplementary Figure (#3 in the revised manuscript) to the revised manuscript which displays ERPs and their main difference between the ketamine and placebo sessions. As can be seen, ERPs have a dampened periodicity due to the approximate periodicity of the stimulus presentation schedule (one stimulus every 400 ms with jitter). This makes it non-trivial to know how much EEG activity at time t after the onset of stimulus k reflects the processing of stimulus k with latency t , or the processing of stimulus $k-1$ with latency $t+400$ ms, or the processing of stimulus $k+1$ with latency $t-400$ ms. Another problem is that it is impossible to know how much EEG activity at the same time t reflects the processing of stimulus orientation, of stimulus evidence, of both, or the processing of another characteristic.

The neural coding approach we have adopted in the manuscript affords a straightforward solution to these different problems, by decoding characteristics that are specifically tied to one specific processing stage of one specific stimulus. The Supplementary Figure entitled **Temporal properties of neural coding** (#4 in the revised manuscript) shows that the neural coding of successive stimuli is indeed significantly overlapping in EEG signals (the same signals that are averaged across stimulus epochs to compute ERPs). Similarly, the neural coding of distinct characteristics of the same stimulus (orientation and evidence) is also strongly overlapping in EEG signals (see the earlier studies cited above for very similar findings). This strong overlap makes it extremely difficult to interpret unambiguously ERP effects.

Nevertheless, ERPs recorded at lateral occipital electrodes show classical early peaks (P1 around 90 ms following stimulus onset, and N1 around 180 ms following stimulus onset). Neither of these ERPs shows a significant difference between the ketamine and placebo sessions (subpanel c). This absence of difference between conditions for early visual ERPs provides additional support that ketamine does not significantly alter early sensory processing in our task:

Nevertheless, ketamine affects ERPs at parietal channels, with a positive difference peaking at 150 ms following stimulus onset, and a negative difference peaking at 410 m following stimulus onset. As explained above, it is highly difficult to interpret these differences or their latencies because they appear roughly periodic (subpanel c, bottom panel) with the same period (400 ms) as the stimulus presentation rate (ISI = 400 ms plus jitter). We think that this new Supplementary Figure has two advantages: 1. showing that early visual ERPs do not differ significantly between the two conditions, and 2. showing that ERP analyses alone are not interpretable - thereby providing a clearer justification for our neural decoding approach.

Reviewer #3

Salvador et al investigates the effects of acute ketamine infusion on evidence accumulation during a line orientation judgment task using computational modeling and EEG. The primary findings are that subjects showed intact discrimination ability but increased decision certainty that was also reflected in reduced low frequency (8-16 Hz) activity over parietal and frontal regions. There was a strong correlation between behavioral and EEG readouts, which is a strength of the study.

Overall, the combination of the computational approach and EEG appears elegant, and the statistical effects appear robust. However, I have significant concerns about how the study is framed and interpreted. These could likely be addressed in a revision but would require considerable work.

We thank the reviewer for his positive appreciation of our approach and analyses, and more generally for his insightful comments. We understand his concerns regarding our broader interpretation of the findings regarding early-stage psychosis. We have followed the reviewer's suggestion of focusing the discussion of our findings in terms of the synaptic effects of acute ketamine infusion rather than the use of ketamine as a pharmacological model of early-stage psychosis. In practice, we have made in this regard extensive modifications of the manuscript to follow the reviewer's suggestions. We have also done our best to clarify the way we report EEG data analyses whenever they resort to specific methods. We hope that the reviewer will find that the revised manuscript offers a more adequate account of our novel findings.

First and foremost, this is not a paper on schizophrenia. The same experiment could easily have been performed using schizophrenia subjects and could have directly tested whether they had analogous deficits and, if so, whether such deficits correlated with delusions or hallucinations. Hopefully, they will one day perform the study. In the interim, however, the potential relevance to schizophrenia can only be a discussion point and a rationale for future studies directly addressing the hypothesis. It should not show up in the title and elsewhere it should be made clear that while the paper may have implications for schizophrenia, these are secondary to the main point of the study and would need to be confirmed in studies of schizophrenia patients themselves.

We understand this point, and fully agree that our manuscript is not a paper on schizophrenia. We purposely chose not to study patients suffering from schizophrenia, but rather to characterize the perturbations resulting from NMDA receptor hypofunction using low-dose ketamine infusions. This synaptic alteration, as the reviewer notes, is one of the prominent hypotheses for the pathophysiology of the early stages of schizophrenia, but the reviewer is fully correct that this link only affords us to draw tentative connections to behavioral symptoms observed in schizophrenia in the discussion. We have thus removed the connection of ketamine infusions to psychosis in the title. We have also removed any reference to schizophrenia in the title as well as in the abstract, and only provide a tentative connection (shorter than in the original version of the manuscript) at the very end of the Discussion section. We are now much more explicit that these tentative connections should be supported by studies of patients suffering from schizophrenia.

In contrast, the paper needs to be justified based on its contribution to basic mechanisms of decision making. As detailed in the manuscript, the concept that NMDAR are involved in slow accumulation of information through reverberating circuits was proposed by Wang and

Wong almost 15 years ago but (to my knowledge) has not been tested experimentally in humans. Despite the elegant data presentation in the present manuscript, it is hard to figure out if the pattern they are seeing supports the proposals by Wang & Wong, refutes them, or is somewhere in between. At the least, there should be a much more complete introduction to the Wang & Wong hypothesis, a clear statement of how this experiment was meant to rigorously test the hypotheses, and a clear statement of the a priori hypotheses (followed by a clear statement of whether or not these were confirmed). Ideally, some attempt to be made to incorporate the approaches used by Wang & Wong in establishing the hypotheses.

We understand the reviewer's comment. Our work stands at the intersection of circuit models of evidence accumulation (such as the canonical model proposed by Wong and Wang) where NMDA receptor inactivation is linked to early and noisy decisions, but also of behavioral symptoms of confirmation biases and 'jumping to conclusions' - notably described in patients suffering from schizophrenia. There are however important differences between our work and the earlier work using the circuit model by Wong and Wang - differences that were not clear enough in the original version of the manuscript.

First, the work using the circuit model by Wong and Wang is based on an evidence accumulation task which differs quite substantially from our task: a random-dot motion task where the sensory evidence is delivered continuously and difficult to quantify at each point in time. Our cue combination task delivers sensory evidence in a discrete fashion, and each piece of evidence can be quantified directly and decoded from EEG signals. Also, the sensory signal is weak and uncertain in the random-dot motion task, whereas it is the relation between strong sensory signals and the category of the sequence that is uncertain in our task. In other words, the source of uncertainty is located at the sensory level in random-dot motion tasks, whereas it is located at the inference (categorization) level in our task - validated in healthy participants in our earlier work (Drugowitsch, Wyart et al., 2016, Neuron). These important differences between task parameters, which we now describe explicitly in the discussion, make it difficult to formally compare the predictions of the circuit model by Wong and Wang to our findings.

A second important difference is that the work using the circuit model by Wong and Wang has studied the effects of strong inactivations of NMDA receptors. In our pharmacological study using low-dose ketamine infusions, NMDA receptor inactivation was only partial and limited. We targeted a plasma concentration of ketamine of 150 ng per mL which had been reported to be sufficient to induce cognitive perturbations, but low enough to avoid stronger psychotomimetic symptoms that might have interfered with task performance (by distracting participants from the task).

Another important point - raised by Reviewer #1 - is that an imbalance between excitation and inhibition in the circuit model by Wong and Wang does not make straightforward predictions regarding the imbalanced coding of evidence that we report under ketamine. Similarly, the circuit model by Wong and Wang describes a single brain area, and there is formally no difference between committing to a decision and executing a motor response. In our data, ketamine triggers premature commitments to uncertain decisions without early motor responses - i.e., 'covert' commitments that were not possible in the single-area model by Wong and Wang. And more generally, our premature commitment model is cast at an abstract 'algorithmic' level (in terms of Marr's three-level decomposition) rather than at a neural 'implementational' level - which is where the E/I model by Wong and Wang is described. It is therefore more adequate to describe and discuss our findings at the algorithmic level where our process model operates and our study is grounded. Such an algorithmic description avoids tying our findings to a specific implementation of evidence accumulation (such as the one by Wong and Wang).

We believe that it is important to emphasize that alteration of NMDA receptors leads to earlier and more noisy decisions in circuit models and in our participants under ketamine. This similarity, however, does not provide a formal test of these circuit models, due to the several important differences described above. We now mention the most important differences in the Discussion section of the revised manuscript, and thank the reviewer for prompting these changes throughout the revised manuscript.

The choice of task needs to be better justified both from a decision making and schizophrenia perspective. From a decision-making perspective, a more traditional approach is to use a task (e.g. coherent dots) in which slow information extraction takes place within a trial rather than across trials. Here, the task seems designed specifically not to require sensory integration. There should be a better description of why this specific task was chosen to test the underlying hypotheses. If the authors wish to retain their discussions regarding schizophrenia, it should also be noted that line-judgement ability is relatively intact in schizophrenia although other aspects of visual sensory processing and orientation judgement are impaired (e.g. PMID: 32291128). To my read, the present findings are consistent with the previously reported findings in schizophrenia. If not, the discrepancies should be noted and would argue against strong homologies between ketamine-induced alterations and schizophrenia. In the section on schizophrenia, it should be noted that the chosen task is one where schizophrenia patients have limited deficits, as opposed to components such as motion detection, and that different sensory-level results might have been obtained with a different task. An advantage of this task, however, is that it allows testing of the cognitive aspects of evidence accumulation independent of sensory aspects, which might be seen as a positive.

We agree with the reviewer that our observation of intact orientation processing under ketamine deserves further discussion in light of the existing literature, in three directions.

First, we have modified the introduction and discussion to justify more explicitly our choice of cognitive task. We now describe the specific features of our task that we have leveraged to measure the effects of ketamine on cognitive inference. This includes the discrete delivery of evidence that we could quantify precisely - a feature that we used to estimate the coding of individual pieces of evidence in the EEG data during cognitive inference. The presented stimuli were associated with little to no sensory uncertainty (highly contrasted, without external noise nor backward masking) but high category uncertainty (due to the low coherence of each stimulus sequence). These specific features of the task required participants to integrate the evidence provided by successive stimuli to make an accurate decision. We now describe the key features of the task in the second paragraph of the Discussion section (p. 21):

“Our experimental protocol was designed to measure the precision of inference and estimate the effect of NMDA receptor hypofunction on this cognitive variable. First, the testing of the same volunteers under ketamine and placebo increased our ability to detect differences between conditions by controlling for individual differences. Second, our cue combination task and validated model allowed dissociating inference errors from sensory noise and selection errors. Instead of measuring decision uncertainty using confidence reports – which could be heavily biased by the dissociative effects of ketamine visible in ratings on psychiatric symptom scales – we relied on the titrated comparison of decision uncertainty with a lottery of known probability of success. Together, these distinctive features of our experimental protocol revealed that ketamine triggers a selective increase in inference errors and an elevated decision uncertainty.”

Second, we agree that the absence of measurable sensory deficit in our task does not imply that NMDA receptor hypofunction cannot produce sensory deficits in other conditions. We have now clarified this point explicitly in a new paragraph of the discussion. We stress in this new paragraph (p. 22) that the absence of sensory deficit in our task (in terms of sensory noise measured in behavior or orientation coding in the EEG data) provides support that ketamine has a specific effect on cognitive inference, rather than a general distracting (inattention) effect which could have confounded our findings:

“The lack of ketamine effect on orientation coding supports the idea that NMDA receptor hypofunction impairs cognitive inference rather than all aspects of stimulus processing in an undifferentiated fashion. Indeed, orientation coding at occipital channels showed the same strength and latency under ketamine and placebo, and was associated with the same, well-known ‘repetition suppression’ effects described in the literature. The selectivity of observed ketamine effects indicates that the drug – which triggered ‘dissociative’ effects on psychiatric symptom scales – did not merely distract participants from the task. Nevertheless, the lack of sensory effect of ketamine in our task does not imply that ketamine does not produce any sensory deficit in other contexts. Indeed, our task is based on visual stimuli with little to no sensory uncertainty (highly contrasted, without external noise nor temporal masking) and thus sets cognitive inference – i.e., the estimation and integration of evidence provided by successive stimuli – rather than sensory processing as the main cognitive bottleneck on task performance. Further research should employ tasks triggering high sensory uncertainty (e.g., random-dot motion stimuli) to determine whether ketamine can also produce sensory effects beyond its selective effects on cognitive inference observed in our task.”

Third, and as indicated in our earlier response, we have followed the reviewer’s comment regarding the connection of our study to schizophrenia. We now make it clear in the revised manuscript that we are not studying patients suffering from schizophrenia, and removed all references to schizophrenia in the title and in the abstract. Conceptually, we have reframed the manuscript on the effect of NMDA receptor hypofunction on cognitive inference and its neural correlates.

We now state explicitly at the very end of the discussion - the only place in the revised manuscript where we discuss our findings in terms of psychosis - that ketamine has been proposed as a pharmacological model of certain behavioral symptoms observed at early stages of psychosis. Interestingly, the paper cited by the reviewer in his comment suggests that some perceptual processes are preserved in schizophrenia. In agreement with this view, we find in our ketamine study that NMDA receptor hypofunction can impair cognitive inference without affecting significantly sensory processing. Like the reviewer, we believe that it is important that our findings validate the hypothesis that NMDA receptor hypofunction has a specific impact on cognitive inference. If we had observed an effect of ketamine on orientation coding (a processing stage occurring upstream from inference), it would have been difficult to know whether ketamine has a specific effect on the coding of evidence over and beyond its effect on orientation coding.

We hope that the reviewer will find these different modifications and clarifications useful to highlight the implications of our findings, and put them in context of the existing literature.

A disappointing aspect of the study is that there is little attempt to link to the abundant neurophysiological literature on either decision making overall or within schizophrenia specifically. At present, the specific pattern of deficit is linked only vaguely to E-I imbalance, which is relatively uninformative since all information processing in the brain relates to

interactions between glutamate (E) and GABA (I) systems. There is no discussion of why ketamine produces effects only on some aspects of E-I interaction in this task and not others since sensory processing also requires E/I balance.

Based on earlier comments from the reviewer, shared by comments from Reviewer #1, we have now expressed our findings at the algorithmic level where our process model operates and our study is grounded. In doing so, we now avoid discussing our findings only in terms of an excitation-inhibition imbalance - an earlier choice which limited the interpretability of our findings outside this particular framework.

We now connect our findings more extensively to the existing literature, and identify open questions emerging from observed differences. We agree that our findings do not only speak to the hypothesis of an excitation-inhibition imbalance triggered by NMDA receptor hypofunction. We have therefore conceptually reframed the manuscript on the more general question of whether and how NMDA receptor hypofunction impairs cognitive inference. An important difference between our modeling framework and the circuit model by Wong and Wang is that these circuit models typically only describe the evidence accumulation stage rather than the full cascade of processing stages - from sensory processing (at the input of the evidence accumulation stage) to response selection (at the output of the evidence accumulation stage). Our cognitive model - which we fit to human behavior - affords to estimate the different sources of suboptimality at these different stages of processing (see Drugowitsch, Wyart et al., 2016, Neuron, for a more complete treatment of this dissociation). We now state in a new paragraph of the discussion (p. 22, copied above) why the specific impairment of cognitive inference despite intact orientation processing is interesting.

As discussed in response to an earlier point by the reviewer (also raised by Reviewer #1), we now discuss how our choice of task (in which the accuracy of sensory processing is not the main cognitive bottleneck on performance) may explain the absence of ketamine effect on sensory processing. We now state in the discussion that the use of stimuli with little to no sensory uncertainty has allowed us to study specifically the effect of NMDA receptor hypofunction on cognitive inference. We also stress that future research should further investigate the effects of ketamine infusions on sensory processing in conditions of high sensory uncertainty. We thank the reviewer (and Reviewer #1 for a similar comment) for stimulating these changes: we agree that our findings do not only speak to the idea of an imbalance between excitation and inhibition triggered by NMDA receptor hypofunction. By avoiding to tie our findings too strongly to one particular mechanism (E/I imbalance) that is not the target of our process model, we believe that the revised manuscript offers a more general interpretation and discussion of our findings.

The introduction cites work proposing that NMDAR hypofunction may lead to hallucinations, which may be true following chronic ketamine exposure. However, hallucinations are not typically seen following acute ketamine (including apparently in this study). This study is thus not a strong test of the hypothesis that cognitive inference alterations are involved in the pathophysiology of hallucinations through E-I modulation. However, if the authors wish to raise the issue they should specifically state that the lack of ketamine-induced hallucinations in the present study argues against direct relevance of the task to these symptoms, although the alterations may be related to other symptoms that were observed during the infusions (e.g. mood, excitement or distractibility).

As indicated earlier, we have modified the manuscript to clarify that the aim of our study was not to mimic schizophrenia, and certainly not to induce delusions or hallucinations. We have re-focused the manuscript on the undisputable synaptic effects of ketamine on NMDA receptors, and only

mention tentative connections of our findings to behavioral symptoms observed at early stages of psychosis in the last paragraph of the Discussion section. As we have clarified in earlier responses, we have used low-dose ketamine infusions as a pharmacological model of prodromal schizophrenia - certainly not a model of delusions or hallucinations. Given the level of plasma concentration that we targeted (150 ng per mL), we aimed at producing ketamine effects on the minute cognitive and neural mechanisms of cognitive inference, while avoiding a state of 'clinically' visible disturbances that would likely have distracted participants from performing our task. We have therefore removed all statements in the manuscript that could mislead readers to think of a direct connection between our findings and clinical symptoms such as delusions and hallucinations. For example, in the introduction (p. 3):

“Theoretical models of neural circuits have identified n-methyl-d-aspartate (NMDA) synaptic receptors as necessary for the accurate integration of noisy input. Indeed, hypofunction of NMDA receptors has been proposed to destabilize the attractor-like dynamics observed in these circuits, by altering the strength of recurrent synaptic connectivity. At the cognitive level, this synaptic alteration is thought to impair inference in a way that can trigger decision biases, ‘jumping to conclusions’, but also deficits in confidence. Together, this theoretical work confers a central role to NMDA receptors in the computational precision of neural circuits implementing cognitive inference. However, and despite the breadth of this work, direct experimental characterization of the effects of NMDA receptor hypofunction on inference and confidence during uncertain decisions is still missing.”

Given the level of plasma concentration that we targeted, we did not design our study to perform between-subject correlations between cognitive or neural effects of ketamine observed in the task and psychiatric symptom scales (specific BPRS items or CADSS score). As described in the methods, we developed our study to afford within-subject comparisons between ketamine and placebo sessions by testing each participant in two separate sessions. This within-subject design afforded us to measure the effects of ketamine independently of between-subject variability in task performance that would otherwise decrease statistical power. We now report the distinctive features of our experimental protocol in the discussion (p. 21):

“Our experimental protocol was designed to measure the precision of inference and estimate the effect of NMDA receptor hypofunction on this cognitive variable. First, the testing of the same volunteers under ketamine and placebo increased our ability to detect differences between conditions by controlling for individual differences. Second, our cue combination task and validated model allowed dissociating inference errors from sensory noise and selection errors. Instead of measuring decision uncertainty using confidence reports – which could be heavily biased by the dissociative effects of ketamine visible in ratings on psychiatric symptom scales – we relied on the titrated comparison of decision uncertainty with a lottery of known probability of success. Together, these distinctive features of our experimental protocol revealed that ketamine triggers a selective increase in inference errors and an elevated decision uncertainty.”

The testing of $N = 20$ participants in this protocol is sufficient to identify within-subject effects (through the repeated-measures comparison between task variables in the ketamine and placebo sessions), but is not likely to be enough to perform between-subject correlations that are thought to require larger samples to be sufficiently powered. In other words, we did not aim to correlate cognitive or neural effects of ketamine in our task with psychiatric symptom scales. We measured these scales to verify that ketamine increased specific symptoms consistent with effects described in the literature. We note in the discussion that future research should test patients suffering from schizophrenia to study the relation between the impairments observed in the task and specific symptoms measured using validated scales (such as the BPRS or CADSS).

There is no attempt to interrelate the spectral signature of the deficit to underlying circuit motifs such as interneuron subtypes, or to isolate specific brain regions that might be involved such as LIP or SMA. There is also no attempt to link to known neuro-oscillatory disturbances in schizophrenia which have been demonstrated in both decision making (e.g. 3753429, 27913408) and more basic sensory (e.g. 31301757) paradigms. Again, any discrepancies between present results and prior observations in schizophrenia should be noted, as they help inform about the validity of ketamine models.

We understand the reviewer's interest in these measures. However, our scalp EEG study with 64 channels was not designed to dissociate specific interneuron subtypes, or isolate specific brain regions through source reconstruction. These additional measures are also relatively tangential to our model-based investigation of the effects of ketamine on cognitive inference. The analyses described in the main text have been selected and designed to test specific hypotheses related to our computational model of cognitive inference.

Nevertheless, we agree that connecting our study to earlier work using ketamine is important. Therefore, based on the reviewer's point and another comment from Reviewer #1, we have extended our analyses of the effects of ketamine on low-frequency power in the alpha and beta bands. These analyses are now illustrated in Supplementary Fig. 9, and show clear agreement with earlier work (e.g., Dias et al., 2013, *Front. Psychol.*; Grent-'t-Jong et al., 2018, *Brain*):

The subpanel (a) shows the effects of ketamine on power spectra and time-frequency diagrams at occipital channels, whereas the subpanel (b) shows the effects of ketamine on power spectra and time-frequency diagrams at central channels. The dampened suppression of low-frequency power in the alpha and beta bands during task execution (here, from the presentation of the stimulus sequence until the response) is indeed highly consistent with the results reported by Dias et al. (2013, *Front. Psychol.*) and Grent-'t-Jong et al. (2018, *Brain*). Our findings add to this existing

literature by showing that the dampened dynamics of alpha- and beta-band power under ketamine during task execution may be due to weaker baseline power in these two frequency bands (measured during the inter-trial interval). We also show in subpanels (c) and (d) that, while ketamine does not affect response preparation activity in the alpha band (Supplementary Fig. 8c), ketamine strongly impairs the same activity measured in the beta band (Supplementary Fig. 8d). This is interesting, because low-frequency power typically shows the same effects under placebo:

We thank the reviewer for prompting these revisions which position our study with respect to existing work. As mentioned above, we originally decided not to insist on these effects that are relatively tangential to our study. But we agree that they are nevertheless relevant in light of the existing literature on the effects of ketamine (and schizophrenia) on EEG activity. We now report these findings briefly in the results (p. 15):

“In agreement with previous work (Dias et al., 2013, *Front. Psychol.*; Grent-’t-Jong et al., 2018, *Brain*), we found that ketamine decreased baseline power in the alpha (10 Hz) and beta (20 Hz) bands and dampened the strong power suppression in these frequency bands during visual stimulation (Supplementary Fig. 9ab). Despite these broad effects of ketamine, the additional power suppression triggered by response execution in the alpha band (10 Hz) at bilateral central channels (Fig. 6a) did not differ between conditions. We could thus use this motor signal to predict the provided response (left- vs. right-handed) in the last few seconds preceding its execution – even before the presentation of the ‘go’ signal which probed participants for a response.”

We hope that the reviewer will find these revisions useful to show that ketamine produced the same effects on oscillatory power in our study as in earlier work.

The manuscript presents EEG results in terms of “coding precision,” which is a somewhat unusual metric. It would be useful, at least in the supplement to show actual time-frequency histograms from the task and to explain more fully the transition from power and/or ITC measures to the coding precision metric. The detailed link between behavior and EEG, which has mostly been done to this point in monkeys, should be the selling point of the paper. There should also be a more complete description of why only 8-32 Hz activity was assessed, whereas sensory responses tend to occur in the theta frequency range, and other relevant processes (e.g. entrainment) may occur at even slower frequencies. Ketamine also has known effects on gamma power, so the reason for not including higher frequencies should also be discussed.

We understand that our rationale for analyzing EEG signals in terms of the coding precision of specific stimulus characteristics needs to be made more explicit.

Event-related potentials (ERPs) are arguably the most standard EEG measure. However, owing to the rapid sequential presentation of stimuli in our task (every 400 ms with jitter), ERPs cannot be unambiguously related to the processing of a particular stimulus in the sequence. In this context, the neural coding approach we have adopted affords a straightforward solution to these different problems, by decoding characteristics that are specifically tied to one specific processing stage of one specific stimulus. Note that we have developed this approach across several past studies (Wyart et al., 2012, *Neuron*; Wyart et al., 2015, *J. Neurosci.*; King and Wyart, 2021, *J. Neurosci.*).

The Supplementary Figure entitled **Temporal properties of neural coding** (Supplementary Fig. 4 in the revised manuscript) shows that the neural coding of successive stimuli is indeed significantly overlapping in EEG signals (the same signals that are averaged across stimulus epochs to compute ERPs, now shown as Supplementary Fig. 3 in response to a point from Reviewer #2). Similarly, the neural coding of distinct characteristics of the same stimulus (orientation and evidence) is also strongly overlapping in EEG signals (see the earlier studies cited above for very similar findings). This strong overlap makes it extremely difficult to interpret ERP effects unambiguously in our task. We now clarify in the results (p. 11) our rationale for conducting EEG analyses in terms of coding precision:

“To identify the neural locus of ketamine effects on cognitive inference, we characterized the neural processing of each stimulus using time-resolved analyses of task-related EEG signals. Due to the rapid sequential presentation of stimuli in our task, standard event-related potentials (ERPs) cannot be interpreted without confounds (Supplementary Fig. 3). We have therefore relied on a neural coding approach which ties EEG signals to the processing of specific stimulus characteristics [Wyart et al., 2012, *Neuron*; Wyart et al., 2015, *J. Neurosci.*; Weiss et al., 2021, *Nat. Comms.*; King and Wyart, 2021, *J. Neurosci.*]. First, we described each stimulus k (about 2,400 per condition and per participant) by two distinct characteristics: 1. its orientation, and 2. the strength of the evidence it provides to the inference process. Because the orientations of category axes varied from trial to trial, these two characteristics were independent of each other. We then applied multivariate pattern analyses to EEG signals aligned to stimulus onset to estimate the neural ‘codes’ associated with these two characteristics (see Methods). Owing to the fine temporal resolution of EEG signals, we could extract the time course of neural information processing within the first hundreds of milliseconds following stimulus onset.”

As indicated in response to the previous point, we have extended our analyses of the effects of ketamine on low-frequency power in the alpha and beta bands. These analyses are now illustrated in Supplementary Fig. 9, and show clear agreement with earlier work (e.g., Dias et al., 2013, *Front. Psychol.*; Grent-'t-Jong et al., 2018, *Brain*).

Supplementary Fig. 5 in the revised manuscript provides relevant information regarding the spectral phase-locked content which contributes to the neural coding precision of different stimulus characteristics. We have used a low-pass filter to determine which frequency bands contribute to the neural coding of stimulus orientation and stimulus evidence. In practice, we have varied the cutoff frequency, from 4 Hz up to 32 Hz, to measure the cutoff frequency for which the neural coding of each stimulus characteristic is maximal - see panel b of Supplementary Fig. 5 to see the neural coding precision of each variable at its peak. We found that the neural coding of stimulus orientation peaks for a frequency cutoff of approximately 16 Hz (the orange curves in the

subpanel), whereas the neural coding of stimulus evidence peaks for a frequency cutoff of approximately 8 Hz (the blue curves in the subpanel):

These results (more visible on the curves shown on Supplementary Fig. 4b than the time-frequency diagrams shown on Supplementary Fig. 4a) indicate that the neural coding of stimulus orientation relies on spectral content up to 16 Hz (above which coding precision starts to decrease), whereas the neural coding of stimulus evidence relies on spectral content up to 8 Hz (above which coding precision starts to decrease). We have now clarified this reasoning in the results (p. 11):

“This neural code overlapped only slightly across successive stimuli (Supplementary Fig. 3c), and was supported by spectral content up to 16 Hz (measured as the frequency cutoff above which coding precision starts to decrease; see Supplementary Fig. 4).”

The link between EEG and behavior is not the only focus of the paper - which rather uses EEG together with behavioral modeling to characterize the effect of NMDA receptor hypofunction on cognitive inference and its neural correlates. However, we fully agree that the link between EEG and behavior is important, and we have devoted several analyses in the study to this aim. For example, the coding imbalance described in Figure 5 and the response preparation activity described in Figure 6 are both based on analyses that relate EEG activity to behavior. Both of these effects are prominent in the abstract, and in the discussion of our findings. Furthermore, we have also related trial-to-trial variability in the neural coding of stimulus evidence to trial-to-trial variability in the weighting of the corresponding stimulus in the subsequent decision. This is the focus of Supplementary Fig. 6 (Supplementary Fig. 5 in the original manuscript):

We mention these analyses explicitly in the manuscript (p. 12):

“This degraded neural processing of stimulus evidence under ketamine mirrors the larger inference errors identified when modeling behavior (Fig. 3). To validate this ‘brain-behavior’ relation, we tested whether the neural coding of stimulus evidence under ketamine correlated with the contribution of the same stimulus to the upcoming decision (Supplementary Fig. 5a, see Methods). We found that stimuli associated with overestimated evidence in EEG signals contributed more strongly to the upcoming decision (Supplementary Fig. 5b,c; $\beta = 0.053 \pm 0.013$, $t_{17} = 4.3$, $p < 0.001$). This relation between neural and behavioral variability indicates that the neural coding of stimulus evidence reflects the noisy representation of momentary evidence being accumulated by participants.”

We focused on frequencies between 8 and 32 Hz for power analyses based on the existing literature on response preparation activity in EEG signals, as indicated in the Results section of the revised manuscript (pp. 15):

“We therefore looked for covert response preparation activity in band-limited EEG power (Donner et al., 2009, *Curr. Biol.*; O’Connell et al., 2012, *Nat. Neurosci.*), a well-validated measure which we could compare between conditions (Fig. 6, see Methods). In agreement with previous work (Dias et al., 2013, *Front. Psychol.*; Grent-’t-Jong et al., 2018, *Brain*), we found that ketamine decreased baseline power in the alpha (10 Hz) and beta (20 Hz) bands and dampened the strong power suppression in these frequency bands during visual stimulation (Supplementary Fig. 9ab). Despite these broad effects of ketamine, the additional power suppression triggered by response execution in the alpha band (10 Hz) at bilateral central channels (Fig. 6a) did not differ between conditions. We could thus use this motor signal to predict the provided response (left- vs. right-handed) in the last few seconds preceding its execution – even before the presentation of the ‘go’ signal which probed participants for a response.”

Note also that we are not ignoring frequencies below 8 Hz, since they are included in the EEG signals used for the neural coding analyses: these signals are high-pass filtered at 1 Hz, and low-pass filtered at frequencies up to 32 Hz. They thus include delta (1-4 Hz) and theta (4-8 Hz) bands. We were not interested in entrainment-related activity for this study which aims at characterizing the processing of individual stimuli in EEG signals.

Gamma-band activity is particularly subtle in scalp EEG signal (in terms of signal-to-noise ratio) and known to be easily contaminated by several artifacts - including line noise and muscular activity. Given that we conducted the study in conditions that did not afford the highest EEG signal quality in this frequency range - i.e., testing in a hospital room without electromagnetic shielding - we did not compute nor aim at studying EEG power in the gamma range for this study.

Some of the terminology needs to be better defined. For example, the term “cognitive inference” seems to subsume processes that might otherwise be called executive processing or working memory both of which have already been studied extensively in ketamine models. For example, in the Wisconsin Card Sorting Task, subjects must accumulate evidence over trials to determine what category of information is relevant to the decision involved. In working memory tasks, information must be maintained over time. It seems like this task mostly taps into evidence accumulation, which would seem to be only one component of cognitive inference. It would be important to know whether the “cognitive inference” term is meant to subsume concepts such as executive processing or working memory that have already been studied in relationship to ketamine or is meant to be a different process. Rather than focusing on schizophrenia, the introduction should focus much more on what sensory and/or cognitive tasks have already been studied with ketamine and how the present study adds to the existing literature.

As stated above in response to an earlier comment, we have rewritten the introduction section of the revised manuscript such that it focuses not on psychosis or schizophrenia, but on the effects of ketamine (and NMDA receptor hypofunction) on cognitive inference (p. 3):

“Theoretical models of neural circuits have identified n-methyl-d-aspartate (NMDA) synaptic receptors as necessary for the accurate integration of noisy input (Wong and Wang, 2006, *J. Neurosci.*; Murray and Wang, 2017, *Comp. Psychiatry*). Indeed, hypofunction of NMDA receptors has been proposed to destabilize the attractor-like dynamics observed in these circuits, by altering the strength of recurrent synaptic connectivity. At the cognitive level, this

synaptic alteration is thought to impair inference in a way that can trigger decision biases (Cavanagh et al., 2020, eLife), ‘jumping to conclusions’ (Adams et al., 2018, *J. Neurosci.*; Strube et al., 2020, *Biol. Psychiatry*), but also deficits in confidence (Vinckier et al., 2016). Together, this theoretical work confers a central role to NMDA receptors in the computational precision of neural circuits implementing cognitive inference. However, and despite the breadth of this work, direct experimental characterization of the effects of NMDA receptor hypofunction on inference and confidence during uncertain decisions is still missing.”

Our computational model of behavior offers an explicit and terminology-free dissection of cognitive inference into sub-components. We have used in the manuscript a terminology which is consistent with earlier work using similar tasks (our own work, see Drugowitsch et al., 2016, *Neuron*; Weiss et al., 2021, *Nat. Comm.*; but also others’ work, see e.g. Tsetsos et al., 2016, *PNAS*; Waskom and Kiani, 2018, *Curr. Biol.*; Shen and Ma, 2019, *Psychol. Rev.*). Indeed, by describing the successive algorithmic steps by which the model solves the task, our work is not tied to a specific terminology and thus offers to clarify which specific processing stage(s) are affected by ketamine. In particular, cognitive inference can be decomposed in two successive processing stages: 1. the estimation of momentary evidence associated with a single stimulus, as a function of the current category axes, and 2. the integration of momentary evidence over time. We have also clarified in the discussion how the different effects of ketamine relate to these two sub-processes.

First, regarding the degraded coding of stimulus evidence under ketamine - corresponding to the estimation of momentary evidence associated with a single stimulus (p. 21):

“Theoretical and empirically motivated models describe the impact of NMDA receptor hypofunction on associative, context-dependent processes required for evidence accumulation (an integral component of cognitive inference). In agreement with these views, we found that ketamine impaired the neural coding of stimulus evidence at frontal and parietal channels at 300 ms following stimulus onset. This impairment indicates that NMDA receptor hypofunction affects the estimation of the momentary evidence provided by each stimulus as a function of category axes which varied randomly across trials – a flexible, context-dependent process known to be supported by neural circuits in parietal and prefrontal cortices. Importantly, this impaired estimation of momentary evidence occurs upstream from the integration of momentary evidence over time – the second processing stage constitutive of cognitive inference.”

Then, regarding the possible effect of ketamine on evidence accumulation leak - corresponding to the integration of momentary evidence over time (p. 23):

“In simulations, premature commitments alter the shape of psychophysical kernels by decreasing the weights of stimuli presented late in each sequence – i.e., after premature commitments have occurred (Supplementary Fig. 10c). This predicted effect stands in apparent contradiction with participants’ psychophysical kernels under ketamine, which did not show such an effect (Supplementary Fig. 2b). However, the shape of psychophysical kernels does not only depend on premature commitments, but also on the ability to accumulate evidence over time without leak (a measure of working memory in our task). Therefore, the similar shapes of participants’ psychophysical kernels under ketamine and placebo – despite the presence of premature commitments under ketamine – suggest that ketamine may impair working memory (as described in previous work, see Adler et al., 1998, *Biol. Psychiatry*) and therefore increase the leak in evidence accumulation (Supplementary Fig. 10d). Future research should further examine the relation between these two cognitive effects of ketamine.”

We hope that these different modifications to the manuscript clarify how the modeling of cognitive inference - in particular the description of its sub-processes - affords the description of obtained findings in a way that is as specific as possible.

The term “belief” also appears to be used promiscuously. In the task subjects make decisions about specific trials, but it trivializes the term somewhat to call their decision a “belief.” From a clinical point of view, it would be more useful to use the term to reflect higher-level concepts. For example, the participant may develop a “belief” that the experimenter is trying to manipulate his/her behavior or that there is a deeper meaning to the interaction than just the simple experiment that they are being asked to perform. These beliefs may interact with the concept of “saliency,” which is extensively discussed in theoretical formulations of delusions and is thought to interrelate with dopamine function. Patients with schizophrenia may also develop “beliefs” that thoughts are being inserted or subtracted from their brains. The present terminology conflates these very different uses. It could be that decision uncertainty is related to higher-level delusions, but that would need to be proven. Even in the present study, it should be possible to examine the relationship between decision uncertainty and unusual thought content (or other symptoms), which I did not see in the manuscript but might have missed. In general, since the CADSS data were collected it would be useful to know whether or not there were correlations between decision-making or EEG results and symptoms. Both positive and negative results should be reported. Positive results would strengthen some of the claims of the manuscript with respect to schizophrenia.

We use the term ‘belief’ in the manuscript in the context of Bayesian theories of reasoning and decision-making. This term is thus not tied either to low- or high-level concepts, as it is the case from a clinical point of view. Based on earlier comments from the reviewer, we have extensively refocused the manuscript in terms of the effects of NMDA receptor hypofunction on cognitive inference and its neural correlates. The discussion of the connection of our findings to the literature on schizophrenia is now restricted to the last paragraph of the discussion, with an explicit statement that future research conducted in patients at different stages of their illness will be required. This conceptual re-focusing of the manuscript makes it clearer that we use belief in the context of Bayesian theories of reasoning and decision-making.

Nevertheless, we have done our best to: 1. reduce the use of the term ‘belief’ to its minimum (we removed it from the title and abstract), because we want our paper to be clear for a broad readership that ranges from cognitive psychologists to clinicians, and 2. clarify explicitly that by ‘belief’ we mean ‘belief about stimulus category’ in the context of our categorization task. By clarifying the revised manuscript in this regard, we believe that it is now clear what we mean by ‘belief’. The first paragraph of the introduction (p. 3) is also much more explicit regarding our definition of the term ‘belief’:

“In uncertain environments where sensory observations are unreliable, making decisions requires the combination of multiple pieces of ambiguous or even conflicting sensory information to form accurate beliefs about their generative cause or their consequences (Vickers, 1979). This form of ‘cognitive inference’ can be described in terms of probabilistic (Bayesian) reasoning, where beliefs correspond to posterior distributions of hidden states of the environment given the available evidence (Oaksford and Chater, 2007). In practice, this inference process has been extensively modeled in terms of a gradual evidence accumulation process (Ratcliff and Smith, 2004) that implements – or approximates – normative Bayesian inference (Wald and Wolfowitz, 1948; Bogacz et al., 2006). Previous research in humans has

shown that the accuracy of this accumulation process is not bounded by the ability to maintain accumulated evidence over time (Brunton et al., 2013; Waskom and Kiani, 2018), but by a limited computational precision – i.e., random variability (noise) during the accumulation of evidence itself. These findings set the precision of inference as an important cognitive ‘bottleneck’ on decision-making under uncertainty (Wyart and Koechlin, 2016; Findling and Wyart, 2021).”

As we have clarified in an earlier response, we did not design our study to perform between-subject correlations between cognitive and neural effects of ketamine observed in the task and psychiatric symptom scales (specific BPRS items or CADSS score). As described in the methods, we developed our study to afford within-subject comparisons between ketamine and placebo sessions by testing each participant in two separate sessions. This within-subject design afforded us to measure the effects of ketamine independently of between-subject variability in task performance that would otherwise decrease statistical power. The testing of N = 20 participants in this protocol is sufficient to identify within-subject effects (through the repeated-measures comparison between task variables in the ketamine and placebo sessions), but is not nearly enough to perform between-subject correlations between psychiatric symptom scores and task-related EEG activity or behavior that are thought to require larger samples to be sufficiently powered.

To summarize, we did not aim to correlate cognitive and neural effects of ketamine in our task with psychiatric symptom scales, and we do not think that these correlations would be interpretable given their low power in the context of our study. We measured these scales to verify that ketamine increased specific symptoms consistent with effects described in the literature. As we state in the discussion section, future research in patients suffering from schizophrenia at different stages of their illness will be required to assess whether the pattern of cognitive impairments observed under ketamine is indeed characteristic of early stages of psychosis.

Overall, this is an intriguing paper in that it combines detailed computational modeling of decision making with high-density EEG and ketamine challenge and thus can directly hypotheses related to the role of NMDAR in decision making as proposed by Wang & Wong, as well as others. As such, it is likely to have a significant effect on the field. The present version addresses the issue only at the level of nonspecific E-I imbalance, which does not take advantage of the richness of the data set. In contrast, there is extensive discussion of potential relevance to delusions in schizophrenia which is highly speculative and not supported by any of the internal data of the study. It should be noted that schizophrenia patients rarely develop delusions about line orientation. There are however syndromes such as Capgras that involve delusional beliefs about faces and identities, but this syndrome is not specifically related to schizophrenia. It is reasonable to speculate about how the present results might be tied to symptoms of schizophrenia such as delusions, but only as a rationale for recommending that similar studies be performed in schizophrenia itself.

Signed:

Daniel C. Javitt

Columbia University

We thank the reviewer for his thoughtful and insightful comments which have stimulated several revisions and additions to the manuscript. We believe that the revised manuscript is much tighter and stronger thanks to these several revisions.

REVIEWER COMMENTS

Reviewer #1 (Remarks to the Author):

The authors have done a great job in revising this manuscript. I was already enthusiastic about the previous version, though puzzled by several interpretational issues. The revised version is now much clearer.

In sum, I strongly support publication of this work (without further revisions) and congratulate the authors on a beautiful paper.

I would like to add one comment that should have no implications for the future editorial process, but is only intended to resolve a *potential* confusion regarding one of my previous set of comments (as Reviewer #1). The authors will know best if it would be useful to react to this with some minor textual adjustment or leave the paper as is.

It is perfectly clear that more premature decision commitment will automatically produce an increase of confidence on specific trials, and all the associated effects that have been identified here. In fact, that was already perfectly clear from the previous version. My point pertained to how exactly the authors *think* about the relationship between the (i) ketamine-induced uncertainty reduction and (ii) of ketamine-induced increase in pre-commitment (with all resulting effects). In my original review, I referred to those as "primary" vs. "secondary" effects, because I had understood that the authors suggest a causal relationship: it is the increase in uncertainty that also pushes observers toward increased pre-commitment. This causal association would NOT be "mechanical"/automatic, but require some active mechanism (e.g., strategic adjustments by the observer). It would, obviously, be reflected in some temporal delay -- at a timescale that is unknown.

Having read the authors' (very careful!) replies, I remain unsure as to what they really think: are those two ketamine effects independent, and just happen to push confidence in opposite directions? Or is the first effect causing the second one?

To illustrate, on the top half of page 3 of the rebuttal, they state "our model hypothesizes that ketamine increases decision uncertainty (effect 1) and triggers premature commitments during inference (effect 2)". The "and" could be interpreted as referring to two independent ketamine effects (?). But the bottom paragraph of the same page reads: "our interpretation is indeed that premature commitments tend to occur because of the elevated decision uncertainty triggered by ketamine", in line with a causal link.

It may help to state clearly, at strategic positions of the paper, how they think about this. It would also be perfectly fine (preferable, from my perspective) to remain agnostic about the nature of this association -- but also this could be stated explicitly. Please apologize if I have missed key statements along this line in the revised version.

Relatedly, while I much like the direct comparison between effects in first vs. second half of the experiment that has now been added, the lack of difference for either of the two effects, of course, does not rule out a temporal delay between them: such a delay could manifest at ANY timescale (though likely slower than a few trials, if it truly reflecting a strategic adjustment).

I truly hope this clarifies my point. As I said above, I am very happy with how the authors have revised this paper and think this will make an important contribution.

Tobias Donner

Reviewer #2 (Remarks to the Author):

The authors have successfully addressed my previous comments, all of which were rather minor.

Reviewer #3 (Remarks to the Author):

This is a much improved version of the manuscript that fully addresses my main concerns regarding how the findings are discussed relative to schizophrenia. I concur with other reviewers about the elegance of the approach. The concepts regarding ketamine effects at the computational level are quite believable and significantly advance our understanding of the role played by NMDAR in ongoing cognition.

I commend the authors for the thoroughness of their revisions and their willingness to provide additional data and analyses. However, there are some issues raised by the new data provided that could be used to further evaluate the role of NMDAR in the task used. Alternatively, at the author's discretion, they could just be added as caveats/limitations.

First, the difference ERP now shown in Supplemental Figure 3 appears to show a relatively classic delta wave entrained to the stimulus presentation rate. The importance of delta entrainment during a rhythmic presentation task has been discussed extensively (e.g. Lakatos et al., *Science*. 2008;320(5872):110-3), as has the breakdown of delta entrainment processes in schizophrenia (e.g. Lakatos *J Neurosci*. 2013;33(28):11692-702. DOI: 10.1523/JNEUROSCI.0010-13.2013). The potential role of NMDAR in delta modulation is discussed by Hong et al., *Neuropsychopharmacology*. 2010;35(3):632-40. DOI: 10.1038/npp.2009.168 and elsewhere. The general underlying construct is that entrainment to ongoing brain rhythms increases the efficiency of brain processing by allowing the relevant cortical region to be in a maximally responsive state at the time of expected stimulus presentation. The predictive modulation of local gamma activity by entrained delta can be detected using intracranial electrodes, although whether or not it projects to the surface in humans will be task and method dependent.

Given the new data presented in the supplement, the authors need to better justify their decision to focus their spectral decomposition analyses only in the 8-32 Hz range. It should be relatively easy to evaluate whether significant delta entrainment occurs during the task and whether it is affected by ketamine. The effects should show up even in a simple FFT, although they could also be detected by extending the window of the spectral decomposition approach currently used in the manuscript. If impairments in delta entrainment correlate with noisy decision making, it would help integrate the neurophysiological and computational data sets. If the analyses do not show a ketamine effect on entrained delta, it would argue against involvement of entrainment in the current task.

However, it is also understandable if the authors do not wish to undertake more analyses given the work that they have already done. In such case, the decision to use EEG band-passed at 8 Hz should simply be listed as a limitation, and it should be explicitly stated that additional processes in the delta and theta frequency ranges might also be of relevance but were not formally evaluated. The caption to Suppl Fig that states that "Ketamine triggers a quasi-periodic ERP difference with the same period at the stimulus presentation rate (ISI = 400 ms). This periodicity makes it impossible to interpret these differences or their latencies" should be changed to simply state that the presence of a quasi-periodic wave at the stimulation frequency raises the possibility that ketamine effects on delta entrainment may also contribute to its behavioral effects (citing the above literature), although these potential effects were not explicitly analyzed.

Second, the authors mention "slow excitation" as the critical feature of NMDA receptors in this task, this is just one distinguishing feature of NMDA receptors and it may or not be relevant to the effects of ketamine. Other properties include 1) non-linear gain due to their requirement for cooperative binding of two glutamate receptors (vs. one for AMPA), 2) voltage-dependent blockade (which enables "Hebbian" integration) or 3) Ca permeability, which allows them to interact with T-type Ca channels in delta generation. I think it is a mistake to pre-judge which of these characteristics contributes to the observed ketamine effects, especially since integration of information over time (which is logically most attributable to prolonged dynamics) appears normal.

Overall, the paper makes an important addition to the literature and illustrates the power of combined pharmacological, computational and neurophysiological approaches.

Signed: Daniel Javitt

Reviewer #1

The authors have done a great job in revising this manuscript. I was already enthusiastic about the previous version, though puzzled by several interpretational issues. The revised version is now much clearer.

In sum, I strongly support publication of this work (without further revisions) and congratulate the authors on a beautiful paper.

We thank the reviewer again for his insightful and positive comments which have clearly improved the manuscript.

*I would like to add one comment that should have no implications for the future editorial process, but is only intended to resolve a *potential* confusion regarding one of my previous set of comments (as Reviewer #1). The authors will know best if it would be useful to react to this with some minor textual adjustment or leave the paper as is.*

We have accounted for this insightful suggestion in the last paragraph of the Discussion.

*It is perfectly clear that more premature decision commitment will automatically produce an increases of confidence on specific trials, and all the associated effects that have been identified here. In fact, that was already perfectly clear from the previous version. My point pertained to how exactly the authors *think* about the relationship between the (i) ketamine-induced uncertainty reduction and (ii) of ketamine-induced increase in pre-commitment (with all resulting effects). In my original review, I referred to those as "primary" vs. "secondary" effects, because I had understood that the authors suggest a causal relationship: it is the increase in uncertainty that also pushes observers toward increased pre-commitment. This causal association would NOT be "mechanical"/automatic, but require some active mechanism (e.g., strategic adjustments by the observer). It would, obviously, be reflected in some temporal delay -- at a timescale that is unknown. Having read the authors' (very careful!) replies, I remain unsure as to what they really think: are those two ketamine effects independent, and just happen to push confidence in opposite directions? Or is the first effect causing the second one?*

To illustrate, on the top half of page 3 of the rebuttal, they state "our model hypothesizes that ketamine increases decision uncertainty (effect 1) and triggers premature commitments during inference (effect 2)". The "and" could be interpreted as referring to two independent ketamine effects (?). But the bottom paragraph of the same page reads: "our interpretation is indeed that premature commitments tend to occur because of the elevated decision uncertainty triggered by ketamine", in line with a causal link.

It may help to state clearly, at strategic positions of the paper, how they think about this. It would also be perfectly fine (preferable, from my perspective) to remain agnostic about the nature of this association -- but also this could be stated explicitly. Please apologize if I have missed key statements along this line in the revised version.

The reviewer is correct that our process model is agnostic to the nature (causal or non-causal) of this relation between elevated uncertainty and premature commitments. This is how the process model is described in the revised (and final) version of the manuscript. We have added an explicit statement in the penultimate paragraph of the Discussion section to clarify this point:

"Note, however, that our computational model remains agnostic about the nature of the relation between these two effects of ketamine."

Nevertheless, the literature cited in the Discussion suggests that, beyond the context of our task and pharmacological intervention, elevated uncertainty may indeed trigger (in a causal sense) strong beliefs. However, in the context of our task and findings, additional clinical work should further investigate this important question. We have therefore added an extra sentence in the last paragraph of the Discussion:

“Confirming this speculative hypothesis will require testing patients diagnosed with schizophrenia in our task at different stages of their illness. Such clinical investigation could clarify whether ‘jumping to conclusions’ (a form of premature commitments) aims at resolving the abnormal uncertainty experienced by patients at early stages of their illness. It may also ultimately help determine whether treatment at early stages of psychosis should aim at increasing tolerance to uncertainty, through psychotherapeutic or pharmacological approaches.”

Relatedly, while I much like the direct comparison between effects in first vs. second half of the experiment that has now been added, the lack of difference for either of the two effects, of course, does not rule out a temporal delay between them: such a delay could manifest at ANY timescale (though likely slower than a few trials, if it truly reflecting a strategic adjustment).

We agree with the reviewer that future work should investigate the issue of a temporal delay between the two effects of ketamine (an issue very much related to the previous point). As for the previous point, we believe that testing patients diagnosed with schizophrenia in our task at different stages of their illness (something which we now mention explicitly in the Discussion) could potentially address this question. Our short-term pharmacological protocol did not aim at addressing this specific question.

We have nevertheless added a sentence to the Discussion section to clarify that our study did not aim at addressing this specific question, and that the comparison between the two halves of the experiment does not rule out a delay between the two effects at other time scales:

“Also, the fact that both effects were sustained across the two halves of the experiment does not rule out a possible delay between their onsets at a different time scale.”

I truly hope this clarifies my point. As I said above, I am very happy with how the authors have revised this paper and think this will make an important contribution.

Tobias Donner

We thank the reviewer again for his insightful review of our work.

Reviewer #2

The authors have successfully addressed my previous comments, all of which were rather minor.

We thank the reviewer again for his insightful comments on our manuscript. His comments have notably led to an important methodological clarification (illustrated in Supplementary Fig. 3) as to why our analyses did not focus on classical event-related analyses of the EEG data – but rather on multivariate decoding of stimulus characteristics.

Reviewer #3

This is a much improved version of the manuscript that fully addresses my main concerns regarding how the findings are discussed relative to schizophrenia. I concur with other reviewers about the elegance of the approach. The concepts regarding ketamine effects at the computational level are quite believable and significantly advance our understanding of the role played by NMDAR in ongoing cognition.

I commend the authors for the thoroughness of their revisions and their willingness to provide additional data and analyses. However, there are some issues raised by the new data provided that could be used to further evaluate the role of NMDAR in the task used. Alternatively, at the author's discretion, they could just be added as caveats/limitations.

We thank the reviewer again for his careful and in-depth comments which have clarified the specific novel contributions of our work to the existing literature. We have accounted for the reviewer's suggestions as limitations and motivations for future work.

First, the difference ERP now shown in Supplemental Figure 3 appears to show a relatively classic delta wave entrained to the stimulus presentation rate. The importance of delta entrainment during a rhythmic presentation task has been discussed extensively (e.g. Lakatos et al., Science. 2008;320(5872):110-3), as has the breakdown of delta entrainment processes in schizophrenia (e.g. Lakatos J Neurosci. 2013;33(28):11692-702. DOI: 10.1523/JNEUROSCI.0010-13.2013). The potential role of NMDAR in delta modulation is discussed by Hong et al., Neuropsychopharmacology. 2010;35(3):632-40. DOI: 10.1038/npp.2009.168 and elsewhere. The general underlying construct is that entrainment to ongoing brain rhythms increases the efficiency of brain processing by allowing the relevant cortical region to be in a maximally responsive state at the time of expected stimulus presentation. The predictive modulation of local gamma activity by entrained delta can be detected using intracranial electrodes, although whether or not it projects to the surface in humans will be task and method dependent.

Given the new data presented in the supplement, the authors need to better justify their decision to focus their spectral decomposition analyses only in the 8-32 Hz range. It should be relatively easy to evaluate whether significant delta entrainment occurs during the task and whether it is affected by ketamine. The effects should show up even in a simple FFT, although they could also be detected by extending the window of the spectral decomposition approach currently used in the manuscript. If impairments in delta entrainment correlate with noisy decision making, it would help integrate the neurophysiological and computational data sets. If the analyses do not show a ketamine effect on entrained delta, it would argue against involvement of entrainment in the current task.

However, it is also understandable if the authors do not wish to undertake more analyses given the work that they have already done. In such case, the decision to use EEG band-passed at 8 Hz should simply be listed as a limitation, and it should be explicitly stated that additional processes in the delta and theta frequency ranges might also be of relevance but were not formally evaluated. The caption to Suppl Fig that states that "Ketamine triggers a quasi-periodic ERP difference with the same period at the stimulus presentation rate (ISI = 400 ms). This periodicity makes it impossible to interpret these differences or their latencies" should be changed to simply state that the presence of a quasi-periodic wave at the stimulation frequency raises the possibility that ketamine effects on delta entrainment

may also contribute to its behavioral effects (citing the above literature), although these potential effects were not explicitly analyzed.

Our study did not aim at studying whether ketamine affects the entrainment of brain activity to the stimulation frequency. Indeed, our paradigm used sequences of visual stimuli that were significantly jittered in time (with an inter-stimulus interval of 400 ms and uniform 80 ms-wide jitter), something which necessarily reduced the strength of steady-state visual evoked potentials in EEG signals. The dampened partial entrainment of EEG data to the stimulation frequency (2.5 Hz), even in the placebo (K-) condition, is visible on Supplementary Fig. 3 (strongly dampened periodicity of ERPs at occipital and parietal channels). Therefore, we believe that future work could indeed, as suggested by the reviewer, investigate the effects of ketamine on the neural entrainment of brain activity, and the relation of these effects with behavioral effects of ketamine.

As suggested by the reviewer, and following his guidelines, we have revised the legend of Supplementary Fig. 3:

“This periodicity makes it impossible to interpret these differences or their latencies, *but raises the possibility that ketamine may also impair the entrainment of phase-locked EEG activity to the stimulation frequency (2.5 Hz) – something which should be explored in future work using less or non-jittered inter-stimulus intervals.*”

Second, the authors mention “slow excitation” as the critical feature of NMDA receptors in this task, this is just one distinguishing feature of NMDA receptors and it may or not be relevant to the effects of ketamine. Other properties include 1) non-linear gain due to their requirement for cooperative binding of two glutamate receptors (vs. one for AMPA), 2) voltage-dependent blockade (which enables “Hebbian” integration) or 3) Ca permeability, which allows them to interact with T-type Ca channels in delta generation. I think it is a mistake to pre-judge which of these characteristics contributes to the observed ketamine effects, especially since integration of information over time (which is logically most attributable to prolonged dynamics) appears normal.

We agree with the reviewer, and thank him for this insightful suggestion. We have therefore updated the Discussion section accordingly to mention these other distinctive features of NMDA receptors:

“NMDA receptors have been proposed to enable balanced computations in associative cortical circuits, *be it through their slow excitation of downstream neurons or other distinguishing features (e.g., their nonlinear gain, their voltage-dependent blockade or their calcium permeability).*”

Overall, the paper makes an important addition to the literature and illustrates the power of combined pharmacological, computational and neurophysiological approaches.

Signed: Daniel Javitt

We want to thank the reviewer again for his insightful review of our work.